# Characterizing Pattern Matching and Its Limits on Compositional Task Structures

**Hoyeon Chang**[1][*]    **Jinho Park**[1][*]    **Hanseul Cho**[1][*]    **Sohee Yang**[2]    **Miyoung Ko**[1]

**Hyeonbin Hwang**[1]    **Seungpil Won**[3]    **Dohaeng Lee**[3]    **Youbin Ahn**[3]    **Minjoon Seo**[1]

[1]KAIST AI        [2]UCL        [3]LG AI Research

{retapurayo, binlepain178, jhs4015, minjoon}@kaist.ac.kr

## Abstract

Despite impressive capabilities, LLMs' successes often rely on pattern-matching behaviors, yet these are also linked to OOD generalization failures in compositional tasks. However, behavioral studies commonly employ task setups that allow multiple generalization sources (e.g., algebraic invariances, structural repetition), obscuring a precise and testable account of how well LLMs perform generalization through pattern matching and their limitations. To address this ambiguity, we first formalize pattern matching as functional equivalence, i.e., identifying pairs of subsequences of inputs that consistently lead to identical results when the rest of the input is held constant. Then, we systematically study how decoder-only Transformer and Mamba behave in controlled tasks with compositional structures that isolate this mechanism. Our formalism yields predictive and quantitative insights: (1) Instance-wise success of pattern matching is well predicted by the number of contexts witnessing the relevant functional equivalence. (2) We prove a tight sample complexity bound of learning a two-hop structure by identifying the exponent of the data scaling law for perfect in-domain generalization. Our empirical results align with the theoretical prediction, under 20× parameter scaling and across architectures. (3) Path ambiguity is a structural barrier: when a variable influences the output via multiple paths, models fail to form unified intermediate state representations, impairing accuracy and interpretability. (4) Chain-of-Thought reduces data requirements yet does not resolve path ambiguity. Hence, we provide a predictive, falsifiable boundary for pattern matching and a foundational diagnostic for disentangling mixed generalization mechanisms.

## 1    Introduction

Despite the remarkable performance of large language models (LLMs) (Brown et al., 2020; Touvron et al., 2023), compositional generalization studies suggest that the core mechanism of generalization might be "pattern matching," i.e., models learning local statistical regularities between input fragments and outputs in some cases (Loula et al., 2018; Johnson et al., 2017; Berglund et al., 2024; Wang et al., 2024a; Mirzadeh et al., 2025; Keysers et al., 2020; Csordás et al., 2022). However, behavioral studies commonly employ task setups that allow multiple generalization sources (e.g., algebraic invariances, structural repetition), discussing pattern matching without a precise definition and diagnosing it post-hoc rather than characterizing it predictively. As a consequence, it remains unclear which behaviors should be counted as pattern matching and which should not, thereby obscuring a constructive and testable account of its boundary.

To make this notion precise, we (1) introduce a data-centered formalism for pattern matching and (2) systematically study how modern architectures like decoder-only Transformers (Vaswani et al., 2017; Radford et al., 2018; 2019) and Mamba (Gu & Dao, 2024) perform generalization through

---

[*]Equal contribution.

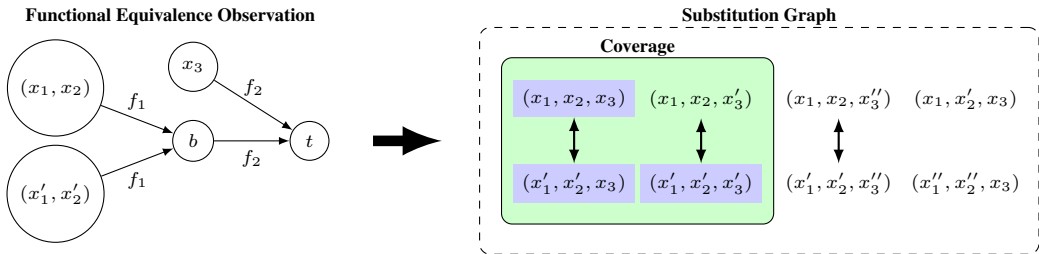

Figure 1: **Illustration of functional equivalence. Left:** In a two-hop task $(x_1, x_2, x_3) \mapsto t$ with $t = f_2(f_1(x_1, x_2), x_3)$, two fragments $(x_1, x_2)$ and $(x_1', x_2')$ satisfying $f_1(x_1, x_2) = f_1(x_1', x_2') = b$ consistently yield the same final output when combined with the same context $x_3$, supporting their **functional equivalence**. **Right:** Among all possible inputs (few shown), we draw an edge between any two inputs that differ only by functionally equivalent fragments to form a **substitution graph**. Then, **coverage** is the set of observed inputs (highlighted as blue) and all inputs connected to them. We define pattern matching as a type of generalization that occurs inside the coverage, harnessing functional equivalence.

pattern matching. Specifically, we first propose a model-agnostic and data-centric definition of pattern matching by formalizing the substitution of input patterns observed to result in identical outputs in shared contexts as **functional-equivalence** (Def. 3.1; henceforth, we use *pattern matching* as equivalent to *functional-equivalence-based generalization*). This induces a **coverage boundary** (Def. 3.2): if learning relies only on such evidence, there is no hope to expect any reliable predictions for the test inputs unreachable by these substitutions.

Moreover, to isolate and study pattern-matching behaviors, we use controlled setups that deliberately remove other sources of generalization and make functional equivalence the *primary available mechanism*. With these setups, our formalism offers *predictive and quantitative insights* about the limitations of pattern matching. To the best of our knowledge, these have not been well characterized yet in prior works:

- **The success in generalization is well predicted by the number of supporting contexts that witness the relevant functional equivalence.** Mechanistically, Transformers seem to implement the functional equivalence by clustering intermediate representations at specific layers/positions. The strength of the clustering aligns with the evidence strength (Sec. 5).

- **We prove a tight sample complexity bound of pattern matching on a two-hop structure by identifying the exponent of the data scaling law (in terms of the token set size) which is necessary and sufficient to achieve perfect in-domain generalization (Theorem 6.1).** We find that the measured power-law exponent agrees with our theoretical bounds, remains stable under roughly $20\times$ parameter increase (from 68M to 1.5B) for GPT-2 (Radford et al., 2019), and also holds for Mamba (Gu & Dao, 2024) architecture (Sec. 6).

- **When the same variable influences the output along multiple computational paths, models fail to form unified intermediate state representations.** Our analysis reveals that they instead develop context-dependent state representations, impairing both generalization and interpretability (Sec. 7).

- **Chain-of-Thought (CoT) supervision (Wei et al., 2022) does *not* completely resolve the path ambiguity without observing nearly exhaustive in-domain combinations**, although it reduces the data requirements to some extent (Sec. 8).

Finally, we situate this characterization of pattern matching within a mechanism-based taxonomy of generalization mechanisms, proposing two additional distinguishable mechanisms of generalization in compositional tasks: property-based and shared-operator generalization (Sec. 9.2 and App. I).

Our formalism opens several research directions with practical implications (e.g., targeted data augmentation to maximize coverage) and motivates expansion to broader tasks and architectures, as well as systematic studies of how pattern matching interacts with other generalization mechanisms. Overall, our study provides a predictive, falsifiable boundary for what can be achieved through pattern matching alone and a foundational diagnostic for disentangling mixed mechanisms in modern neural networks.

## 2   RELATED WORK

**Pattern matching behaviors of LLMs on compositional tasks.**   It is well perceived that pattern matching alone is inadequate for systematic generalization (Fodor & Pylyshyn, 1988), and modern LLMs display generalization abilities that seem to be far beyond what pattern matching alone can do, as measured by their remarkable performance on complex benchmarks (OpenAI, 2023). However, a growing body of work has consistently reported that LLMs still fall short on benchmarks designed to test compositionality (Hupkes et al., 2020), including mathematical reasoning (Mirzadeh et al., 2025), multi-hop reasoning (Yang et al., 2024; Wang et al., 2024a), and more (Lake & Baroni, 2018; Kim & Linzen, 2020; Csordás et al., 2022; Dziri et al., 2023). This gap between their capabilities and pattern-matching behaviors on compositional tasks calls for a principled framework to define what pattern matching is and to what extent a model's behavior can be attributed to pattern matching, but it is mostly discussed with behavioral studies under the context of a specifically designed benchmark. Our work addresses this gap by formally defining pattern matching, and we systematically analyze models' behaviors with controlled tasks that are designed to isolate pattern-matching regimes grounded in our framework.

**Mechanistic interpretability.**   Mechanistic interpretability studies aim to understand how sub-mechanisms implement models' behaviors (Elhage et al., 2021; Olsson et al., 2022; Nanda et al., 2023; Elhage et al., 2022). Recent work analyzes how Transformer components are causally related to certain behaviors (Meng et al., 2022; Hanna et al., 2023; Goldowsky-Dill et al., 2023). In particular, it is reported that in-domain compositional generalization can emerge through grokking, with identifiable intermediate state representations inside Transformers (Wang et al., 2024a). Our framework complements these works by providing mechanistic insights about pattern matching. Our findings also explain why standard interpretability techniques like logit lens (nostalgebraist, 2020; Belrose et al., 2023) may fail to identify state representations in models trained on tasks with path ambiguities.

## 3   FORMALIZING PATTERN MATCHING WITH FUNCTIONAL EQUIVALENCE

We now develop a formal framework for pattern matching. We first provide an intuitive illustration with a two-hop structure, then generalize to arbitrary fixed-length discrete-sequence tasks.

Imagine a learner observing data determined by $f : \mathcal{X}^3 \to \mathcal{X}$. The input $\boldsymbol{x} = (x_1, x_2, x_3) \in \mathcal{X}^3$ is a sequence of three discrete tokens and the output is a single token, where each token is chosen from a finite set $\mathcal{X}$.[1] Suppose (unknown to the learner) that $f$ factorizes as the composition of two primitive functions, $f(\boldsymbol{x}) = f_2(f_1(x_1, x_2), x_3)$, where $f_1 : \mathcal{X}^2 \to \mathcal{X}$ and $f_2 : \mathcal{X}^2 \to \mathcal{X}$, as illustrated in Fig. 2a. How can the learner generalize by only seeing the input-output patterns?

Our key intuition is that **a learner exploits the underlying patterns only when two fragments of inputs are observed to behave identically.** For instance, assume that two fragments $(x_1, x_2), (x_1', x_2') \in \mathcal{X}^2$ give the same implicit intermediate state upon the application of $f_1$, i.e., $f_1(x_1, x_2) = f_1(x_1', x_2') = b$. These fragments behave identically regardless of context, i.e., they are **functionally equivalent**: for all $x_3 \in \mathcal{X}$, $f(x_1, x_2, x_3) = f(x_1', x_2', x_3)$. If observations consistently support their equivalence, i.e., $f(x_1, x_2, x_3) = f(x_1', x_2', x_3)$ for observed $x_3$ values, this equivalence can be supported (Fig. 1 Left). Intuitively, the learner would harness this equivalence pattern to predict $f(x_1', x_2', x_3'')$, provided the training set contains $f(x_1, x_2, x_3'')$.

Equivalently, the learner can utilize the observed functional equivalence to correctly infer the output of an unseen input, if it can reach an observed input by 'safe substitutions' (edges in the substitution graph) supported by observations (Fig. 1 Right), which we define as a pattern matching. **Coverage** is a set of such inputs that are reachable from an observed input through chains of functionally equivalent substitutions. Then, coverage sets a boundary for what can be achieved by solely relying on substituting observed, equivalently behaving patterns. In other words, a learner can only generalize inside the coverage when it relies on functional equivalence, which we will define as pattern matching.

We now formalize these concepts for an arbitrary fixed-length task with an arbitrary set of discrete sequence observations. We restrict our attention to single-token prediction tasks defined as a de-

---

[1] For brevity, we use a shared token set $\mathcal{X}$ in most parts of this paper. A more general notion using position-specific domains (e.g., '$\mathcal{X}_i$') can be used; e.g., see App. F.1.

terministic mapping $f : \mathcal{X}^\ell \to \mathcal{X}$, where $\mathcal{X}$ is a finite set of tokens. We also consider a fixed observation set $D \subset \mathcal{X}^\ell$, a collection of inputs that are allowed to be observed by the learner. Write $\boldsymbol{x} = (x_1, \ldots, x_\ell) \in \mathcal{X}^\ell$ and, for a subset $I \subset [\ell] := \{1, \ldots, \ell\}$, let $\boldsymbol{x}_I := (x_i)_{i \in I}$ be a subsequence of $\boldsymbol{x}$. The first step is to formalize what it means for two subsequences to be **functionally equivalent**.

**Definition 3.1** (Functional $k$-equivalence). Fix a nonempty proper subset $I$ of indices in $[\ell]$. Consider any set $S \subset \mathcal{X}^\ell$ of input sequences.[2] Given a pair of subsequences $\boldsymbol{a}, \boldsymbol{a}' \in \mathcal{X}^{|I|}$, we say a pair of inputs $\{\boldsymbol{x}, \boldsymbol{x}'\}$ to be an $I$-**co-occurrence of** $\boldsymbol{a}$ **and** $\boldsymbol{a}'$ **in** $S$ if it satisfies $\{\boldsymbol{x}, \boldsymbol{x}'\} \subset S$ and $\boldsymbol{x}_I = \boldsymbol{a}$, $\boldsymbol{x}'_I = \boldsymbol{a}'$, and $\boldsymbol{x}_{[\ell] \setminus I} = \boldsymbol{x}'_{[\ell] \setminus I}$. Also, the subsequences $\boldsymbol{a}$ and $\boldsymbol{a}'$ are said to be **functionally $k$-equivalent at** $I$ **in** $S$ and denoted by $\boldsymbol{a} \equiv_I^{(D,k)} \boldsymbol{a}'$, if it satisfies:

1. (**Sufficiency of co-occurrences.**) There are $k$ or more distinct $I$-co-occurrences of $\boldsymbol{a}$ and $\boldsymbol{a}'$ in $S$;
2. (**Consistency.**) Every $I$-co-occurrence $\{\boldsymbol{x}, \boldsymbol{x}'\}$ of $\boldsymbol{a}$ and $\boldsymbol{a}'$ in $S$ satisfies $f(\boldsymbol{x}) = f(\boldsymbol{x}')$.

In other words, two subsequences are functionally $k$-equivalent if they behave identically in the same contexts at least $k$ times. The hyperparameter $k$ represents the strength of evidence required to establish functional equivalence between two subsequences. The minimum value $k = 1$ corresponds to the weakest form of evidence, meaning a single shared context is sufficient to establish equivalence, whereas higher values of $k$ demand more robust evidence.

Next, we ask: which inputs are reachable from observed data utilizing functional equivalence? To formalize this, we define **substitution graph**: Let $\mathcal{G}^{(D,k)} = (V, E)$ be an undirected graph with a vertex set $V = \mathcal{X}^\ell$ of all possible inputs. Two vertices $\boldsymbol{x}, \boldsymbol{x}' \in V$ are connected with an edge in $E$ if and only if there exists an index set $I \subset [\ell]$ such that $\boldsymbol{x}_I \equiv_I^{(D,k)} \{\boldsymbol{x}, \boldsymbol{x}'\}$ is an $I$-co-occurrence (in $V$) of a pair of functionally $k$-equivalent sequences at $I$ in $D$. This process is illustrated on the right side of Fig. 1, as a special case where $k = 1$. With this substitution graph $\mathcal{G}^{(D,k)}$, we formally define the $k$-**coverage**[3] as a set of inputs which are connected to at least one observed input as follows:

**Definition 3.2** ($k$-coverage). The $k$-coverage of $D \subset \mathcal{X}^\ell$, denoted by $\mathrm{Cover}_k(D)$, is the set of all inputs in $\mathcal{X}^\ell$ connected to an $\boldsymbol{x} \in D$ with a path in the substitution graph $\mathcal{G}^{(D,k)}$.

Note that the notion of coverage is a stricter condition of the canonical definition of in-domain (ID), which is obtained by random train/test split (Wang et al., 2024a) or taking combinations of observed internal computations (Dziri et al., 2023). In Sec. 5, we demonstrate that learners may not necessarily generalize on data that are classified as ID in a canonical sense, but coverage can precisely explain when and why this occurs. We also emphasize that coverage is a property of a dataset and is independent of model architectures and learning algorithms, and we demonstrate that the predictions made by our framework are invariant across model architecture and scale in Sec. 6 and 7. Finally, $k$-coverage can be algorithmically determined for any fixed-length discrete sequence tasks (Alg. 1), which we use for the analyses in the following sections.

**Now, we formally define pattern matching as a kind of generalization that is done by substituting functionally $k$-equivalent fragments of inputs, whose boundary is precisely the $k$-coverage defined above.** This formalization enables us to predict, before testing, which inputs will be reliably handled through pattern matching and which require additional mechanisms. In other words, we view pattern matching as possible only within $k$-coverage, and generalization outside the coverage requires generalization mechanisms other than pattern matching, which we discuss in Sec. 10 and App. I. In the following sections, we draw a systematic picture of how task structure, dataset, and model size interact to determine the success and failure of pattern matching through controlled setups, leading us to important and nontrivial insights.

## 4 EXPERIMENTAL SETUP

**Dataset construction.** We construct four synthetic tasks with different structures: 2-HOP, PARALLEL 2-HOP, 3-HOP, and NON-TREE (Fig. 2). To isolate functional-equivalence-based generalizations, we randomly create ground-truth mappings from a product space of token sets to control the generalization sources not attributable to compositional structures (i.e., commutativity). We explain the dataset

---

[2]The set $S$ can be any subset of the whole domain, e.g., $\mathcal{X}^\ell$ itself, the train dataset $D$, or whatever else.
[3]For an undirected graph $\mathcal{G}$, two vertices $u, v$ are connected if $\mathcal{G}$ contains a path between $u$ and $v$.

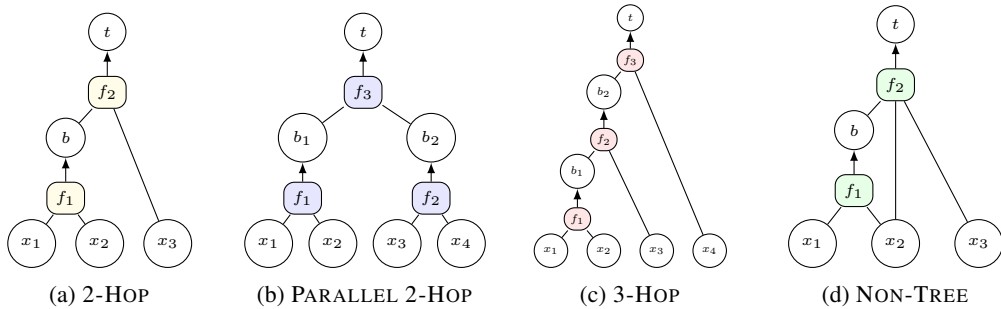

(a) 2-Hop  (b) Parallel 2-Hop  (c) 3-Hop  (d) Non-Tree

Figure 2: Four synthetic task structures we study.

construction process using 2-Hop task (Fig. 2a), $(x_1, x_2, x_3) \mapsto t$ with $t = f_2(f_1(x_1, x_2), x_3)$, as an example. We construct training datasets by defining a token set with size $|\mathcal{X}|$ and randomly defining two mappings for the primitive functions $f_1 : \mathcal{X}^2 \to \mathcal{X}$ and $f_2 : \mathcal{X}^2 \to \mathcal{X}$. We mark a fraction $p_{\text{seen}} = 0.7$ of each function's domain as 'seen', gather all possible combinations where both functions are applied to inputs from their seen domains, and uniformly sample $N$ examples to form a training dataset. See App. B.1 for more details of the dataset construction process.

**Training & evaluation.** Following Wang et al. (2023), we train randomly initialized GPT-2 (Radford et al., 2019) models with 8 layers, 12 heads, and 768 dimensions as a base model (see App. B.2 for details). We construct two evaluation sets, each with 2,000 instances: **(1) ID Test Set** : all primitive function applications (e.g., $f_1(x_1, x_2)$ and $f_2(b, x_3)$ in 2-Hop task) are observed during training, but their specific combination was unseen. **(2) Out-of-coverage (canonical OOD) Test Set**: at least one primitive function application is never observed during training, which is used as a control group.

## 5 Quantitative Analysis of Pattern Matching in Transformers

### 5.1 Evidence Strength is Tightly Aligned to Pattern-Matching Success

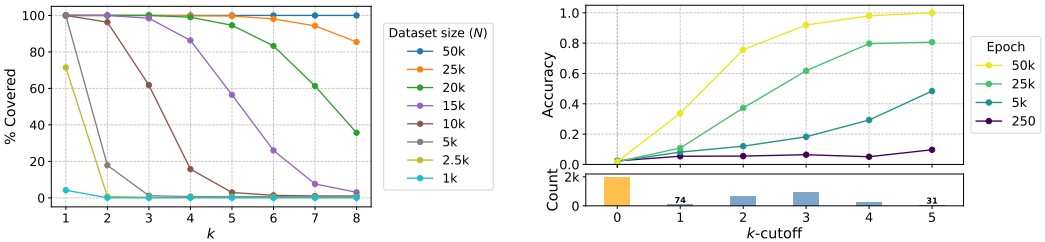

Figure 3: **Left**: Percentage of covered ID data depending on $k$ values and dataset size ($N$), for 2-Hop task ($|\mathcal{X}| = 50$). **Right**: Test accuracy depending on $k$-cutoff values for 2-Hop task ($|\mathcal{X}| = 50$, $N$=10k). Each line represents a different training checkpoint. Note that out-of-coverage ($k = 0$) accuracy remains at chance level ($\approx 1/50$) regardless of training time. The bars below show the number of test data for each $k$-cutoff value.

We first analyze the correlation between $k$-coverage and ID generalization performance of the GPT-2 model. To this end, we implement and release a task-agnostic coverage determination algorithm (see App. C) that can be applied to diverse compositional structures. Then, we analyze what fraction of ID test data of 2-Hop task with $|\mathcal{X}| = 50$ lies inside $k$-coverage, depending on $k$ and dataset size $N$. Fig. 3 (Left) shows that, at $N = 5$k, every ID test example is already covered with minimal evidence ($k = 1$). Hence, in an ideal scenario where a single witness of functional equivalence suffices, training with the dataset as small as $N = 5$k will lead to perfect ID generalization.

However, we demonstrate in our experiments that minimal coverage (i.e., $k = 1$) alone is practically insufficient for ID generalization. To demonstrate this, let us fix a training dataset $D$ and define the $k$-cutoff of each input sequence as the lowest value of $k$ for which an input lies in $k$-coverage, measuring the strength of evidence for functional equivalence. For example, a $k$-cutoff of 3 means that an example is inside coverage with $k = 3$ but not with $k = 4$. For out-of-coverage data, we define $k$-cutoff as 0. Then, for the 2-Hop dataset with $N = 10$k, we classify each ID test instance

according to its $k$-cutoff, and track the accuracy development of the GPT-2 model for each group across 50k training epochs. As shown in Fig. 3 (Right), ID test accuracy shows a strongly positive correlation with $k$-cutoff values. Test data with low $k$-cutoff values show delayed improvement even after extensive training, while examples with stronger evidence generalize much faster.

These results yield two important insights. **First, successful ID generalization in practice requires a *robust* coverage so the model can confidently identify and utilize functional equivalence relationships.** The parameter $k$ effectively quantifies this evidence strength, directly impacting generalization speed and reliability. Second, while our experiments use uniformly sampled datasets, the results can explain why models struggle with generalizing long-tail distributions in imbalanced real-world data (Mallen et al., 2023; Kandpal et al., 2023; Chang et al., 2024). Despite technically being in-distribution, rare combinations naturally receive limited evidence of functional equivalence (low $k$), effectively and practically placing them outside the coverage. We believe that our insights will motivate future research on targeted data augmentation strategies to maximize $k$-coverage.

## 5.2 Latent Representation Clusters Drive Pattern Matching on Coverage

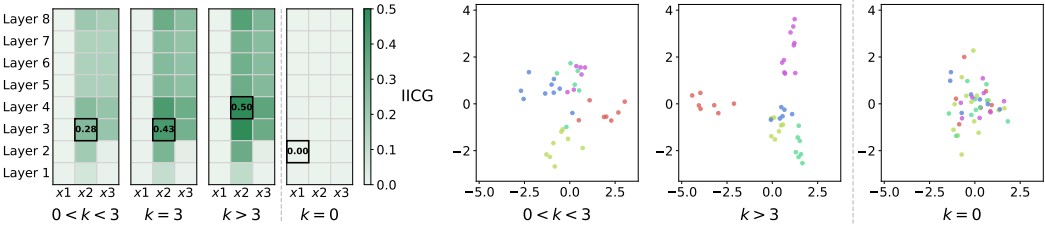

Figure 4: **Left**: Heatmap of Intra-Inter Cosine Gap (IICG) across layers and positions, sliced by $k$-cutoff. Higher IICG values indicate stronger clustering of representations that share the same intermediate state. The positions with the highest IICG values are marked with squares. **Right**: PCA visualization of latent representations at position $x_2$ and layer 3. Datapoints are classified by their intermediate states $b = f_1(x_1, x_2)$.

Next, we investigate how the model internally represents functional equivalence for $k$-covered inputs. Specifically, we inspect a GPT-2 trained on 2-Hop task ($|\mathcal{X}| = 50$, $N = 10k$) for 50k epochs (corresponding to the yellow line in Fig. 3 (Right)).[4] We observe that when a model successfully generalizes to ID test data, it maps functionally equivalent components into tight latent clusters, thereby encoding the equivalence relationships needed for compositional generalization.

To quantify this representation clustering phenomenon, we develop a metric that captures how distinctly the model separates functionally equivalent fragments from others. Specifically, we measure the difference between the average pairwise cosine similarity of latent vectors that share the same intermediate state $b = f_1(x_1, x_2)$ ($\overline{\cos}_{\text{intra}}$), and those that do not ($\overline{\cos}_{\text{inter}}$), for each position and layer of the model. We term this difference the **intra–inter cosine gap** $\text{IICG} = \overline{\cos}_{\text{intra}} - \overline{\cos}_{\text{inter}}$, where higher values indicate stronger within-group clustering relative to between-group separation. Fig. 4 (Left) reveals a positive correlation: higher $k$-cutoff values yield higher IICG scores at certain positions, **indicating that stronger functional equivalence evidence leads to more coherent internal representations.** In contrast, out-of-coverage ($k = 0$) examples exhibit no clustering pattern, as they lack evidence of functional equivalence in the training data. The PCA visualization at position $x_2$ and layer 3 (Right) shows this trend visually. We verify that the representation clusters play a causal role in pattern matching with causal tracing (Goldowsky-Dill et al., 2023; Hanna et al., 2023), a widely used technique to identify Transformer circuits (Fig. 8).

Our findings extend the previous insights from mechanistic interpretability literature (Wang et al., 2024a) in several ways. First, we demonstrate that unified circuit formation is driven by functional equivalence evidence in the training data, not by explicit exposure to intermediate computation steps. Moreover, we find that these clustered representations are not necessarily aligned with vocabulary embeddings, implying that standard interpretability methods like logit lens (nostalgebraist, 2020) may fail to detect these functional equivalence representations despite their presence (see App. E).

---

[4]The analyses for varying factors including task structures, entity set size ($|\mathcal{X}|$), dataset size ($N$), and training steps give consistent results; see App. D.

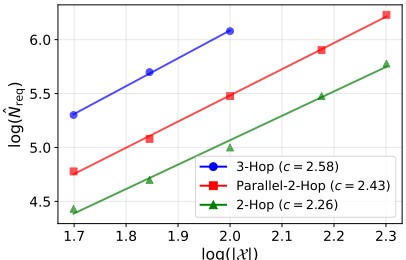 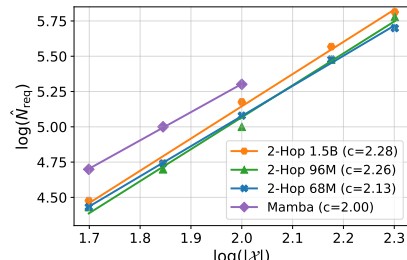

Figure 5: **Left:** Log-log plot of measured $\hat{N}_{\text{req}}$ vs. token set size ($|\mathcal{X}|$) across three compositional tasks. The slope $c$ corresponds to the empirical power-law scaling exponent. Omitted points for 3-HOP are due to prohibitively large dataset requirements. **Right:** Power-law scaling behavior on 2-HOP task across varying GPT-2 model sizes (68M to 1.5B parameters) and Mamba model (For Mamba, we used 4 layers, a hidden dimension of 256, and a learning rate of 0.008, and $\hat{N}_{\text{req}}$ is measured for only $|\mathcal{X}| \leq 100$, since a larger token set size led to training instability). $R^2 > 0.99$ for all linear fitting.

## 6 DATA SCALING LAW OF PATTERN MATCHING BEHAVIORS

Our analysis in the previous section demonstrates that stronger functional equivalence evidence leads to better generalization. A natural follow-up question arises: How large should the training set be to enable full generalization on all ID test data? Intuitively, this requires the training set to support (strongly enough) the functional equivalence of *every* pair of inputs that shares the same intermediate state $b$. Formally, for a 2-HOP task we need $(x_1, x_2) \equiv_{\{1,2\}}^{(D,k)} (x_1', x_2')$ whenever $f_1(x_1, x_2) = f_1(x_1', x_2')$. Assuming that generalization is constrained by the $k$-coverage, how should the train data size scale in the token set size $|\mathcal{X}|$ to achieve it completely? In practical terms, this question seeks to determine the amount of data necessary and/or sufficient to cover all ID combinations with $k$-coverage, which is crucial for understanding the data cost for pattern-matching generalization. To quantify the data cost, we establish and prove the following sample-complexity upper bound for learning a 2-HOP task (the complete statement and proof are provided in App. F):

**Theorem 6.1** (Informal; Corollaries F.9 and F.17). *Consider a* 2-HOP *task with a token set of size* $n$. *For a uniformly randomly sampled train dataset* $D$ *of size* $N$, *consider a learner that generalizes within the* $k$-*coverage of* $D$. *Then, for sufficiently large* $n$, *the learner can achieve perfect ID generalization with high probability if* $N \gtrsim n^c$ *with* $c = 2.5 - \frac{0.5}{k}$.

*In contrast, the learner (with* $k \geq 2$*) cannot achieve perfect ID generalization with high probability for some* 2-HOP *task if* $n^2 \lesssim N \lesssim n^c$. *Here, we ignore the polylogarithmic factors in* $n$.

Theorem 6.1 presents a tight sample complexity bound $\tilde{\Theta}(n^c)$ for pattern-matching-based generalization, showing that the training dataset required to ensure full ID generalization grows polynomially in the token set size $n$, with an exponent $c \in [2, 2.5]$. To empirically confirm this, we define a practical threshold $\hat{N}_{\text{req}}$ to estimate $N_{\text{req}}(|\mathcal{X}|, k)$, as a minimal amount of training data required to exceed ID accuracy of 0.99 within 100 epochs after reaching the same level on training data (see App. G for the measurement details). Fig. 5 (Left) shows the measured power-law exponents for $\hat{N}_{\text{req}}$ vs. $|\mathcal{X}|$ across different task structures. The measured exponent for 2-HOP ($c = 2.26$) aligns well with our theoretical predictions. Although we derive the theoretical bound only for 2-HOP, we observe clear power-law relationships for more complex structures as well. The higher exponents for PARALLEL-2-HOP ($c = 2.43$) and 3-HOP ($c = 2.58$) tasks suggest that extra computational steps essentially add another dimension of relationships that require robust coverage, driving the steeper power-law scaling.

These exponents remain invariant across three different GPT-2 model sizes spanning a 20x range in parameters (from 68M to 1.5B) for all three tasks (Fig. 5 Right and Tab. 2 in App. G). We also show that the exponent measured with a Mamba model (4 layers and a hidden dimension of 256) falls inside the boundary predicted by the theory (same figure). Interestingly, the result in Fig. 6 (Middle) demonstrates that with increasing training dataset size $N$, there is a sharp phase transition from ID generalization failure to complete success near $N = 20$k.

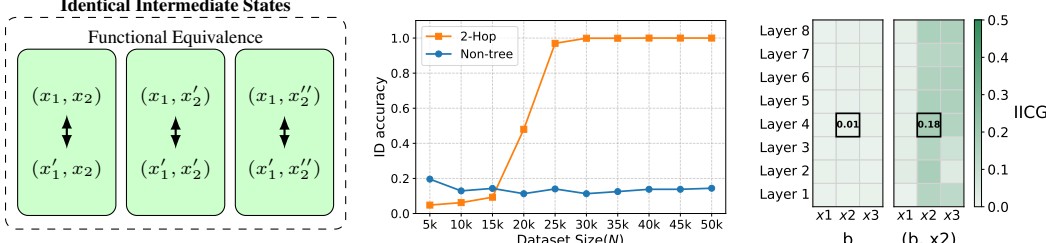

Figure 6: **Left**: In NON-TREE task, the representations of input subsequences with the same intermediate state $b = f_1(x_1, x_2)$ are split into multiple context-dependent state representations, conditioned on $x_2$ value. **Middle**: ID test accuracy after standard training with varying training dataset size ($|\mathcal{X}| = 50$, evaluated 100 epochs after training accuracy reaches 0.99). Observe a sharp transition from ID generalization failure to complete success near $N = 20k$ for 2-HOP, which does not occur in NON-TREE task. **Right**: IICG heatmap from a model that achieved near-perfect ID accuracy (0.96) after extended training (36k epochs, $|\mathcal{X}| = 50$, $N = 50k$).

Overall, the results support that the data scaling law is primarily determined by data properties rather than model capacity or architectures, and additional generalization mechanisms will be required to achieve milder scaling laws on such compositional tasks.[5] We note that our result aligns with the practical observation that parameter scaling does not significantly improve the multi-hop reasoning capability of LLMs (Yang et al., 2024) and the data-hungry nature of compositional tasks (Lake & Baroni, 2018), suggesting that these could be partly attributed to pattern-matching behaviors. We leave further analysis of the connection between scaling behavior and pattern matching as an exciting future research direction.

## 7 PATH AMBIGUITY PROBLEM AS A FAILURE CASE OF PATTERN MATCHING

We identify a *path ambiguity problem* with our framework, a previously uncharacterized failure mode that pattern matching struggles with task structures where a single variable affects the output through multiple paths. In this section, we analyze NON-TREE task (Fig. 2d) as a case study, where $x_2$ affects the output through two paths, as input to $f_1$ and directly to $f_2$. Unlike in the 2-HOP case, one cannot establish the functional equivalence of two subsequences $(x_1, x_2)$ and $(x_1', x_2')$ that produce the same intermediate state $b$, unless they also share the same $x_2$ value ($x_2 = x_2'$). It is because $(x_1, x_2)$ and $(x_1', x_2')$ are not guaranteed to behave identically (i.e., $f(x_1, x_2, x_3)$ is not necessarily equal to $f(x_1', x_2', x_3)$) when $x_1 \neq x_2$. Consequently, we can predict that Transformers trained on the NON-TREE will create context-dependent state representations that are conditioned on $x_2$ values, failing to unify them to represent the true intermediate state $b$ (Fig. 6 Left).

Experiments show that the path ambiguity indeed hinders both generalization on the ID test set and the interpretability of intermediate state representations, as the model now establishes functional equivalence for each $x_2$-conditioned equivalent pair. Fig. 6 (Middle) shows that GPT-2 can fully generalize on the ID test set of 2-HOP task within a reasonable time with increasing data size, but fails with NON-TREE task, even provided with a near-exhaustive amount of possible ID combinations as training data.[6] Notably, scaling to 1.5B parameters does not show significant improvement in the performance (Fig. 18), and the Mamba model used in Sec. 6 shows the same trend of generalization failure (Fig. 19). In addition, extremely prolonged training (36k epochs) with near-exhaustive ID combinations eventually achieves ID accuracy of 0.96; however, IICG analysis reveals no evidence of a unified intermediate state representation formation, with near-zero IICG scores when grouping by the intermediate state value $b$ (Fig. 6 Right). In contrast, grouping by $x_2$-conditioned intermediate state $(b, x_2)$ leads to high IICG scores, showing the formation of context-dependent state representations. This context-dependence due to path ambiguity raises an interpretability concern, as standard linear probing-based techniques like logit lens (nostalgebraist, 2020; Belrose et al., 2023) would not reliably identify intermediate states when a model relies on pattern matching.

Hence, a generalization mechanism other than pattern matching will be required for a robust ID generalization on complex task structures that require the access and update of intermediate states

---

[5]The observed scaling relationships are robust across different hyperparameters (weight decay and learning rate) and empirical decision criteria for $\hat{N}_{req}$ (see App. G).

[6]For $|\mathcal{X}| = 50$ and $p_{seen} = 0.7$, our largest run ($N = 50k$) includes virtually the entire domain ($\approx 0.7^2 \times |\mathcal{X}|^3 \approx 61k$ distinct ID triples).

through multiple paths (e.g., planning tasks (Ruis et al., 2020; Kambhampati et al., 2024; Valmeekam et al., 2023)), where further characterization of this problem remains as an exciting future direction.

## 8 CoT Improves Data Efficiency but Path Ambiguity Persists

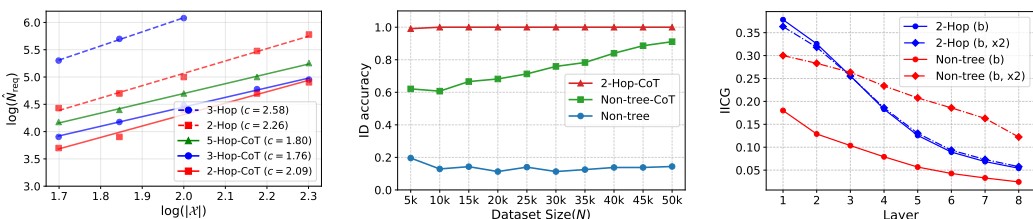

Figure 7: **Left:** Power-law scaling of required dataset size vs. token set size for tasks with CoT supervision. $R^2 > 0.98$ for all linear fits. **Middle:** Comparison of ID test Accuracy of NON-TREE task ($|\mathcal{X}| = 50$) with and without CoT supervision. **Right:** IICG score comparison for NON-TREE and 2-HOP task with CoT supervision ($|\mathcal{X}| = 50$, $N = 10k$). The scores are measured at each layer of intermediate state position $b$, based on two grouping strategies: $b$ and $(b, x_2)$. Models are trained for 100 epochs after reaching training accuracy $> 0.99$.

The CoT supervision (Wei et al., 2022; Kojima et al., 2022) dramatically improves performance on multi-step reasoning tasks. We investigate how it interacts with our framework and whether it can address the challenges observed in Sec. 6 and 7. Specifically, we train models to sequentially generate intermediate states before final outputs, making 2-HOP a two-token prediction task: $(x_1, x_2, x_3) \mapsto (b, t)$, for example. This substantially improves data efficiency (Fig. 7 (Left)), with the power-law exponent dropping from 2.58 to 1.76 in the 3-HOP task, aligning with previous studies on the sample efficiency of CoT (Srivastava et al., 2023; Kim & Suzuki, 2025; Wen et al., 2025). The scaling exponents measured for 2-HOP, 3-HOP, and even 5-HOP tasks become nearly identical with CoT supervision. We interpret this as CoT effectively 'flattening' multi-hop structures into sequences of single-hop tasks, reducing the compounding data requirements of deeper compositional structures.

However, we find the path ambiguity problem persists even with CoT supervision. Despite showing improvements, the models fail to achieve perfect ID generalization under the same training conditions that yield perfect performance in 2-HOP task (Fig. 7 (Middle). IICG analysis (Right) reveals that the model's representations remain partially context-dependent. For the 2-HOP task, the representations cluster purely by intermediate states $b$, as indicated by the result that IICG measurement with $x_2$-conditioned states does not significantly shift the curve. In contrast, the IICG score for NON-TREE task is significantly elevated at every layer with the same conditioning, suggesting the absence of disentangled state representation inside the model. We hypothesize this arises since CoT supervision does not give enough evidence that different $(x_1, x_2)$ pairs sharing the same $b$ should yield identical second-step outputs, as functional equivalence holds only when $x_2 = x_2'$. Hence, while CoT supervision helps with sequential computation by breaking down multi-hop structures, it may partially inherit the limitations on handling tasks with path ambiguities we describe in Sec. 7. Our analysis may explain why LLMs struggle with complex planning tasks even when using CoT techniques and massive training data (Stechly et al., 2024), where we leave further analysis as future work.

## 9 DISCUSSION

### 9.1 PRACTICAL IMPLICATIONS

Although the experiments in this work have focused on synthetic setups to dissect purely pattern-matching behaviors, our results provide valuable practical implications for understanding and improving LLMs on natural language tasks.

**First, it accounts for the data-hungry nature of compositional tasks by demonstrating that robust coverage is required for reliable generalization (Lake et al., 2017).** For instance, our scaling-law analysis (Theorem 6.1) provides a quantitative explanation for the observed behavior that the data demand for generalization on multi-hop natural language data dramatically increases with the number of hops (Yao et al., 2025). Such a data-hungry nature of compositional tasks is also well observed in semantic parsing (Dong & Lapata, 2016), where it has been shown that the diversity of

data (i.e., the same component is shown in diverse contexts) is more important than the sheer size of the dataset for better generalization (Keysers et al., 2020). Based on these insights, we believe our results will motivate strategic data augmentation methods that seek to maximize coverage by ensuring diverse shared contexts for functionally equivalent components (Andreas, 2020).

**Second, our framework helps interpret various real-world failure cases observed in LLMs as stemming from their reliance on pattern matching.** For example, our framework aligns with the reversal curse phenomenon, which is the failure to automatically infer that '$A$ is a child of $B$' by observing that '$B$ is a parent of $A$' (Berglund et al., 2024). This occurs because pattern matching inherently cannot generalize to the reversing relation without clear evidence of functional equivalence in the training dataset. Similarly, our framework can explain the notoriously hard problem of logical negation in LLMs (Truong et al., 2023). A purely pattern-matching learner cannot deduce that the negation rule (i.e., if a statement '$p$' is true, then 'not $p$' is false and vice versa) should apply to any well-formed statement, including the ones that aren't observed, through finite observations. Finally, difficulties in solving complex planning tasks with LLMs (Valmeekam et al., 2023; Stechly et al., 2024; Kambhampati et al., 2024; Wang et al., 2024b) may partly result from path ambiguities, since such tasks likely require the correct tracking of intermediate states, which can be affected by multiple computational paths.

Hence, we believe our framework and analyses can provide important practical insights, which future work can extend to investigate LLMs' pattern-matching behaviors under more realistic setups.

### 9.2 TOWARDS A TAXONOMY OF GENERALIZATION MECHANISMS

Natural language tasks exhibit algebraic and structural properties that distinguish them from the randomly generated mappings we studied. For example, the same knowledge can be used in any hop of the multi-hop reasoning in practice, unlike our 2-HOP task that used $f_1 \neq f_2$. In practice, a learner may reasonably harness such properties of a given task to generalize beyond the boundary of pattern-matching defined by coverage. Therefore, it is natural to ask: **What generalization mechanisms enable generalization beyond the coverage boundary?** While a complete answer requires future work, we outline a mechanism-based taxonomy as a starting point for a constructive categorization of distinct generalization mechanisms beyond pattern matching:

- **Functional equivalence-based generalization**, the main focus of this work.
- **Function property-based generalization** leverages algebraic invariances of individual primitive functions, e.g., commutativity or input irrelevance, where certain arguments never affect the output. This distinguishes it from pattern matching, as it leverages a primitive function's global property that holds across all inputs, not only those observed.
- **Shared-operator generalization** leverages the reuse of the same computation across positions (e.g., when $f_1 = f_2$ in a two-hop task), which may be important in compositional generalization. For example, it is known that Transformers with inductive biases towards computation reuse can improve generalization on compositional tasks (Csordás et al., 2021).

We envision this taxonomy as a foundational diagnostic that quantifies when pattern matching suffices and when other mechanisms are required. See App. I for a complete discussion on the categorization of generalization mechanisms.

## 10 CONCLUSION

In this work, we formalized a framework for characterizing pattern matching. Our theoretical and experimental analyses provided quantitative and predictive insights into modern neural networks' pattern-matching behaviors, moving beyond post-hoc accounts of a model's behavior on compositional tasks: (i) the alignment of instance-wise success with the strength of functional-equivalence evidence (Sec. 5), (ii) the theoretical identification and empirical verification of a sharp sample complexity bound for complete ID generalization on the 2-HOP task through pattern matching (Sec. 6), and (iii) the identification of the path ambiguity problem that impairs accuracy and interpretability even under high coverage and CoT supervision (Sec. 7 and 8). We anticipate that future work will build on this foundation, towards a more complete and constructive understanding of compositional generalization and its failures.

## REPRODUCIBILITY STATEMENT

All codes for dataset generation, training, and analysis can be found at: https://github.com/kaistAI/coverage-principle.

## THE USE OF LARGE LANGUAGE MODELS (LLMs)

This work deployed LLMs to proofread for grammatical errors and improve the quality of writing.

## ACKNOWLEDGEMENTS

We thank Jaewon Oh, Jihoon Ha, Hyowon Cho, Yunjae Won, Seongyun Lee, and Boshi Wang for their invaluable feedback.

This work was supported by Institute for Information & communications Technology Planning & Evaluation(IITP) grant funded by the Korea government (MSIT) (RS-2019-II190075, Artificial Intelligence Graduate School Program(KAIST)). Hoyeon Chang, Jinho Park, Miyoung Ko, Hyeonbin Hwang, & Minjoon Seo acknowledge support by Institute of Information & communications Technology Planning & Evaluation (IITP) grant funded by the Korea government (MSIT) (No.RS-2019-II190075 Artificial Intelligence Graduate School Program (KAIST), 10%; No.RS-2021-II212068, Artificial Intelligence Innovation Hub, 10%; RS-2024-00398115, Research on the reliability and coherence of outcomes produced by Generative AI, 20%; No.2022-0-00113, Developing a Sustainable Collaborative Multi-modal Lifelong Learning Framework, 20%; No.RS-2022-II220264, Comprehensive Video Understanding and Generation with Knowledge-based Deep Logic Neural Network, 20%; RS-2024-00397966, Development of a Cybersecurity Specialized RAG-based sLLM Model for Suppressing Gen-AI Malfunctions and Construction of a Publicly Demonstration Platform) and the InnoCORE program of the Ministry of Science and ICT(N10250156). Hanseul Cho acknowledges support by IITP grant funded by the Korean government (MSIT) (No. RS-2024-00457882, National AI Research Lab Project).

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

# Contents of the Appendix

# A    LIMITATIONS

We deliberately restrict to synthetic tasks to isolate structure-based limits without confounds from lexical or domain priors. We leave extending the coverage analysis to discrete sequence tasks with variable lengths and more natural data as future work. Additionally, our experiments focus on autoregressive architectures, and the applicability of the coverage principle to broader architectures remains to be validated.

# B    DETAILED EXPERIMENTAL SETUP

## B.1    DATASET CONSTRUCTION

We now provide detailed information about our dataset construction process. While we primarily explain this process for the 2-HOP task, we follow similar procedures for the other compositional structures.

**Vocabulary and Token Representation**    For a task with token set size $|\mathcal{X}|$, we create $|\mathcal{X}|$ special tokens of the form `<t_0>`, `<t_1>`, . . ., `<t_(|X| − 1)>`, which we append to the standard GPT-2 vocabulary. We also add special tokens `</a>` to mark the end of sequences. For Chain-of-Thought (CoT) experiments, intermediate computations are represented in the target sequence as the actual intermediate token.

**Function Construction**    For the 2-HOP task, we construct two primitive functions $f_1 : \mathcal{X}^2 \to \mathcal{X}$ and $f_2 : \mathcal{X}^2 \to \mathcal{X}$ by randomly mapping from their respective domains to the codomain $\mathcal{X}$. We create the domain by taking the Cartesian product of the token set with itself. For each function, we randomly designate a fraction $p_{\text{seen}} = 0.7$ of its domain as the "seen" portion, resulting in sets $S_{f_1}$ and $S_{f_2}$.

**Dataset Generation Algorithm**    To generate the training dataset, we first identify all possible combinations where both primitive operations come from their respective "seen" domains. Specifically, we find all valid tuples $(x_1, x_2, x_3, t)$ such that:

$$(x_1, x_2) \in \text{domain}(S_{f_1}) \tag{1}$$

$$(f_1(x_1, x_2), x_3) \in \text{domain}(S_{f_2}) \tag{2}$$

$$t = f_2(f_1(x_1, x_2), x_3) \tag{3}$$

From this set of all possible in-domain combinations, we uniformly sample $N$ examples to form our training dataset. When the number of possible combinations exceeds $N$, this sampling ensures the model sees only a subset of possible in-domain combinations.

**Test Set Construction**    We carefully construct test sets to evaluate the model's generalization capabilities across coverage conditions. Our test sets contain:

- **In-Domain (ID) Test Set**: Combinations unseen during training but where both primitive operations were observed in other contexts. These examples may lie within the coverage as defined by our framework.

- **Out-of-coverage (canonical OOD) Test Set**: Examples where at least one primitive operation was never observed in training. These fall outside the coverage.

**Input-Output Format**    The dataset is formatted for auto-regressive token prediction. For the standard 2-HOP task, inputs comprise three tokens representing $x_1$, $x_2$, and $x_3$, while the target includes these input tokens followed by the prediction $t$ and an end marker. Below are the examples of the dataset format for different settings.

- **Standard Format:**
    - Input: `<t_5><t_12><t_3>`
    - Target Completion: `<t_17></a>`
    - The model must predict the final output token followed by the end marker.
- **Chain-of-Thought Format:**
    - Input: `<t_5><t_12><t_3>`
    - Target Completion: `<t_9><t_17></a>`
    - The model must first predict the intermediate computation result `<t_9>` (where `<t_9>` = $f_1$(`<t_5>`, `<t_12>`)), followed by the final output.
- **Partial Computation Format ($f_1$):**
    - Input: `<t_5><t_12>`
    - Target Completion: `<t_9></a>`
    - These examples represent the primitive function applications used to construct the full compositional task.

For the other compositional tasks, we follow analogous construction procedures, adjusting the number of input tokens and the composition structure based on the specific task's requirements. For example, PARALLEL 2-HOP requires four input tokens, while 3-HOP follows a three-step composition chain requiring appropriate modifications to the function construction and sampling procedures.

## B.2 TRAINING DETAILS

Table 1: Model configurations for different GPT-2 variants used in our experiments

| Configuration | GPT-2-Small | GPT-2 | GPT-2-XL |
|---|---|---|---|
| Number of Attention Heads | 6 | 12 | 25 |
| Number of Layers | 4 | 8 | 48 |
| Hidden Dimension | 768 | 768 | 1600 |
| Total Parameters | 68M | 96M | 1.5B |

For our experiments, we employ three GPT-2 model variants of increasing size: GPT-2-Small (68M parameters), GPT-2 (96M parameters), and GPT-2-XL (1.5B parameters). As shown in Tab. 1, GPT-2-Small consists of 4 layers with 6 attention heads and a hidden dimension of 768. The standard GPT-2 configuration used in most experiments features 8 layers with 12 attention heads while maintaining the same hidden dimension of 768. Our largest model, GPT-2-XL, significantly scales up the architecture with 48 layers, 25 attention heads, and an increased hidden dimension of 1600. The implementation follows the codebase from (Wang et al., 2024a).

We train all models using the AdamW optimizer with beta values of (0.9, 0.999) and epsilon of 1e-8. We set the learning rate to 8e-4 with a weight decay of 0.1. A batch size of 16,384 is used, with full gradient descent applied for datasets smaller than the batch size. All training is conducted with mixed precision (fp16) on 4 NVIDIA A100 GPUs with 80GB memory each. We employ a constant learning rate schedule with a linear warmup period of 2,000 steps. This standardized training configuration is maintained across all experiments to ensure fair comparisons between different task structures and dataset sizes, unless explicitly varied in specific ablation studies.

## C  IMPLEMENTATION DETAILS FOR THE COVERAGE DETERMINATION ALGORITHM

In this section, we explain our algorithm of computing the $k$-coverage $\mathrm{Cover}_k(D)$ (see Def. 3.2), given a train dataset $D \subseteq \mathcal{X}^\ell$ and a threshold $k$ to check functional equivalences (see Def. 3.1). The algorithm works in three main stages, as described with a pseudo-code in Alg. 1.

---

**Algorithm 1:** $k$-Coverage Determination Algorithm

**Input:**
- Training examples: $D \subseteq \mathcal{X}^\ell$, each with known output ($f(\boldsymbol{x}) \in \mathcal{X}$ for each $\boldsymbol{x} \in D$)
- Minimum evidence threshold: $k \geq 1$

**Output:** Coverage set: $\mathrm{Cover}_k(D)$

```
/* STEP 1:  Build behavior maps for each subsequence pattern  */
```
1  Behav ← an empty hash table
2  **foreach** $(I, I^c) : \varnothing \neq I \subsetneq [\ell],\ I^c = [\ell] \setminus I$ **do**
3      Behav$[I]$ ← an empty hash table
4      **foreach** $\boldsymbol{x} \in D$ **do**
5          **if** $\boldsymbol{x}_I \notin$ Behav$[I].keys$ **then**
6              Behav$[I][\boldsymbol{x}_I]$ ← an empty hash table
7          **end**
8          Behav$[I][\boldsymbol{x}_I][\boldsymbol{x}_{I^c}]$ ← $f(\boldsymbol{x})$
9      **end**
10 **end**

```
/* STEP 2:  Build substitution graph using behavior maps    */
```
11 $\mathcal{G}$ ← a graph (vertex set $\mathcal{X}^\ell$; edge set $\varnothing$)
12 **foreach** $(I, I^c) : \varnothing \neq I \subsetneq [\ell],\ I^c = [\ell] \setminus I$ **do**
13     **foreach** $(\boldsymbol{a}, \boldsymbol{a}') : \boldsymbol{a}, \boldsymbol{a}' \in$ Behav$[I].keys$ **and** $\boldsymbol{a} \neq \boldsymbol{a}'$ **do**
14         CoOcc ← Behav$[I][\boldsymbol{a}].keys$ $\cap$ Behav$[I][\boldsymbol{a}'].keys$    // $I$-co-occurrences
15         **if** $|$CoOcc$| \geq k$ **and** $\forall \boldsymbol{b} \in$ CoOcc : Behav$[I][\boldsymbol{a}][\boldsymbol{b}] =$ Behav$[I][\boldsymbol{a}'][\boldsymbol{b}]$ **then**
```
                   /* Statisfies the functional k-equivalence (Def. 3.1) */
```
16             **foreach** $\boldsymbol{b} \in \mathcal{X}^{\ell - |I|}$ **do**
17                 Let $\boldsymbol{x}, \boldsymbol{x}' \in \mathcal{X}^\ell$ such that $\boldsymbol{x}_I = \boldsymbol{a},\ \boldsymbol{x}'_I = \boldsymbol{a}'$, and $\boldsymbol{x}_{I^c} = \boldsymbol{x}'_{I^c} = \boldsymbol{b}$
18                 Add an edge $\{\boldsymbol{x}, \boldsymbol{x}'\}$ to $\mathcal{G}$
19             **end**
20         **end**
21     **end**
22 **end**

```
/* STEP 3:  Build k-coverage                                */
```
23 $\mathrm{Cover}_k(D) \leftarrow \bigcup_{\boldsymbol{x} \in D} \mathrm{ConnectedComponent}(\mathcal{G}; \boldsymbol{x})$
24 **return** $\mathrm{Cover}_k(D)$

---

In the first stage (Alg. 1, Lines 1–10), we first create a mapping of *behaviors* for all subsequences of every training data. To elaborate on the behaviors, for each subset of indices $I$, we record how each subsequence $\boldsymbol{x}_I := (x_i)_{i \in I}$ at $I$ of $\boldsymbol{x} \in D$ behave when being paired with another subsequence $\boldsymbol{x}_{I^c}$ at the complement of indices $I^c := [\ell] \setminus I$. This ends up creating a thrice-nested mapping Behav, as showcased in Line 8. It is particularly useful in the next stage, where we should collect the co-occurrences of each pair of input sequences and verify their consistencies.

The second stage (Alg. 1, Lines 11–22) is for establishing the *substitution graph* $\mathcal{G} = \mathcal{G}^{(D,k)}$ which is essential in the definition of $k$-coverage. To recall, $\mathcal{G}^{(D,k)}$ is an undirected graph with vertex set $\mathcal{X}^\ell$, having edges $\{\boldsymbol{x}, \boldsymbol{x}'\}$ when and only when there exists an index set $I$ such that $\boldsymbol{x}_I \equiv_I^{(D,k)} \boldsymbol{x}'_I$. To verify the presence of an edge via functional $k$-equivalence, we need to figure out at least $k$ distinct $I$-co-occurrences (CoOcc in the pseudocode) for some index set $I$ (Def. 3.1(1)) and check whether all $I$-co-occurrences satisfy the consistency (Def. 3.1(2)). It is done for every nonempty index set

$I \subsetneq [\ell]$ and every distinct pair of subsequences $(\boldsymbol{a}, \boldsymbol{a}') \in \mathcal{X}^{|I|} \times \mathcal{X}^{|I|}$, where we take advantage of the mapping Behav defined in stage 1; see Line 15 in particular.

In the last stage, we generate the $k$-coverage (Alg. 1, Line 23). Here we utilize a subroutine ConnectedComponent($\mathcal{G}; \boldsymbol{x}$) which returns the set of all vertices of $\mathcal{G}$ sitting on the same connected component as the given vertex $\boldsymbol{x}$. Hence, Line 23 generates the set of all input sequences that are connected with a train datapoint by a path in the substitution graph $\mathcal{G}$, which is precisely the definition of $k$-coverage. In other words, this set comprises all inputs that are reachable from the training data through a chain of functionally equivalent subsequence substitutions.

# D    DETAILED ANALYSIS FOR REPRESENTATION UNIFICATION EXPERIMENTS

## D.1    CAUSAL TRACING METHODOLOGY

To analyze the causal role of specific hidden representations in our Transformer model, we employ causal tracing, a technique that measures the effect of intervening on intermediate activations during inference (Goldowsky-Dill et al., 2023; Hanna et al., 2023). Specifically, we measure the causal effect using the *indirect effect* metric defined in (Sharma et al., 2024). This methodology enables us to identify which components and positions in the model most strongly contribute to compositional generalization. We illustrate the measurement with the 2-HOP task.

Our analysis begins by collecting three types of computational traces:

1. **Clean run** ($G$): We run the model on a compositional task with input $(x_1, x_2, x_3)$ where the corresponding output is $t = f_2(f_1(x_1, x_2), x_3)$.

2. **Corrupted run** ($G^*$): We replace the original input with a corrupted version by changing the first two tokens $(x_1, x_2)$ to $(x_1', x_2')$, where $f_1(x_1', x_2') \neq f_1(x_1, x_2)$. This ensures that the model produces a different final output $t^* \neq t$. During this run, we cache all hidden states $h^{*(\ell)}_i$ for each token position $i$ and layer $\ell$.

3. **Patched run** ($G[\leftarrow h^{*(\ell)}_i]$): We run the model on the input from the clean run, but at a specific token position $i$ and layer $\ell$, we replace the hidden state with the corresponding state from the corrupted run.

To quantify the causal effect of a specific hidden state $h^{(\ell)}_i$ on the model's prediction, we measure the *Indirect Effect* (IE):

$$\text{IE}_{h^{(\ell)}_i} = \frac{p[\leftarrow h^{*(\ell)}_i](t^*) - p(t^*)}{p^*(t^*) - p(t^*)}$$

where:

- $p(t^*)$ is the probability assigned to the corrupted output $t^*$ in the clean run $G$;

- $p^*(t^*)$ is the probability assigned to the corrupted output $t^*$ in the corrupted run $G^*$;

- $p[\leftarrow h^{*(\ell)}_i](t^*)$ is the probability assigned to the corrupted output $t^*$ in the patched run $G[\leftarrow h^{*(\ell)}_i]$.

This metric quantifies how much corruption in a particular state affects the overall outcome. An IE value close to 1 indicates that the corruption of the state $h^{(\ell)}_i$ to $h^{*(\ell)}_i$ alone almost completely changes the prediction to that of the corrupted run, suggesting that this state is causally important for the computation. Conversely, an IE value close to 0 indicates that the state has minimal causal impact on the prediction.

In our experiments, we apply causal tracing to analyze different subsets of test data categorized by their $k$-cutoff values, where $k$ represents the minimum evidence threshold required for functional equivalence (as defined in Sec. 3 of the main text). This allows us to correlate the strength of functional equivalence evidence with the formation of unified internal representations.

## D.2    CAUSAL TRACING RESULTS FOR EACH $k$-CUTOFF VALUE IN 2-HOP TASK

Fig. 8 displays the causal tracing results for the 2-HOP task, broken down by different $k$-cutoff values. We observe that the causal patterns are similar across different $k$-cutoff values, with slight differences in where and how strongly the causal effects manifest in the model. This suggests that once an example falls within coverage (even with minimal evidence, $k = 1$), the model forms internal representations that play similar causal roles in prediction.

## D.3    TOKEN SET SIZE ABLATION

We show that the observed patterns of cosine similarity analysis and causal tracing in the 2-HOP task are consistent across different token set sizes $|\mathcal{X}|$. For $|\mathcal{X}| = 70, 100, 150, 200$, we analyze model checkpoints with training dataset size $N = \hat{N}_{\text{req}}(|\mathcal{X}|)$ that achieve training accuracy $> 0.99$. Figure

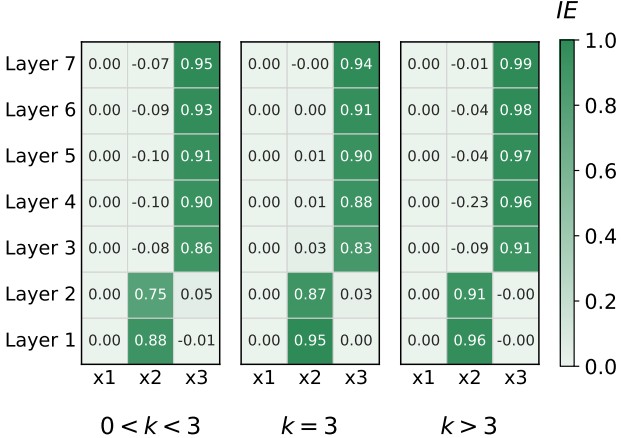

Figure 8: Causal tracing results for the 2-HOP task across different $k$-cutoff values, showing Indirect Effect (IE) scores at each layer and position.

Fig. 9 shows the results, indicating strong representation clustering at the lower layers of position $x_2$ for all cases. The causal tracing results in Fig. 10 show that the clustered functional equivalence representations at the lower layers of position $x_2$ play a causal role in determining the model's final prediction.

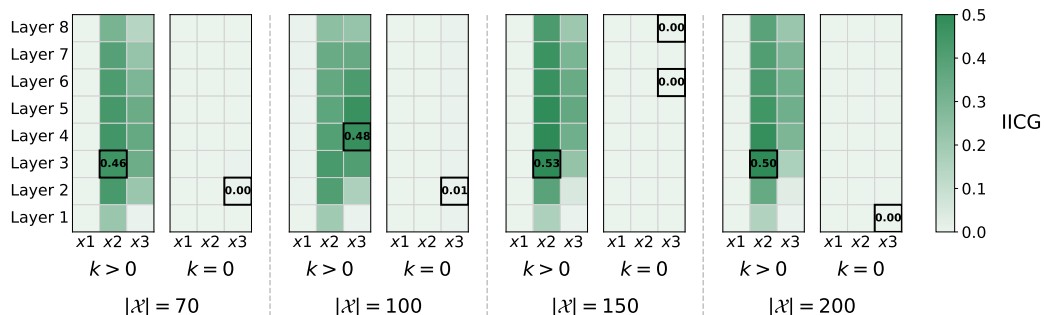

Figure 9: IICG heatmap across different token set sizes, showing consistent representation clustering patterns.

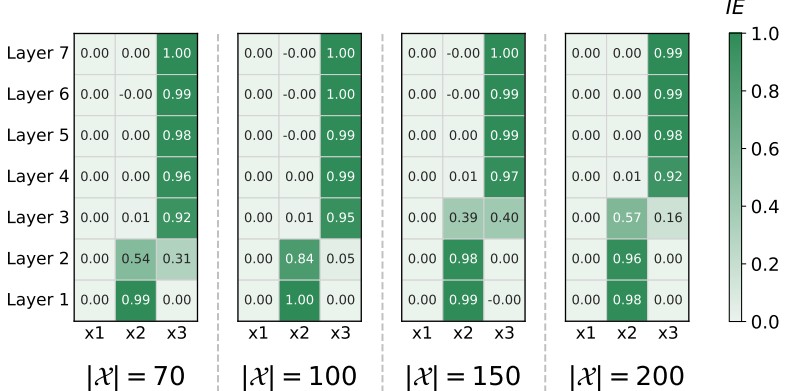

Figure 10: Causal tracing results showing indirect effect heatmaps for different token set sizes $|\mathcal{X}|$.

## D.4 TASK ABLATION

We show that GPT-2 models trained on PARALLEL-2-HOP and 3-HOP tasks exhibit the same patterns: clustered functional equivalence representations of intermediate states at specific layers and positions, confirmed through cosine similarity analysis, with causal tracing analysis verifying their role in model predictions. For both tasks, we analyze with $|\mathcal{X}| = 50$ and examine model checkpoints with training dataset size $N = \hat{N}_{\text{req}}(|\mathcal{X}|)$ that achieve training accuracy $> 0.99$.

Figs. 11 and 12 show the results for the PARALLEL-2-HOP task. The IICG patterns reveal strong representation clustering at mid-layers: at positions $x_2$ and $x_3$ when grouped by $b_1 = f_1(x_1, x_2)$, and at position $x_4$ when grouped by $b_2 = f_2(x_3, x_4)$. Causal tracing confirms the causal role of these clustered representations in the model's predictions.

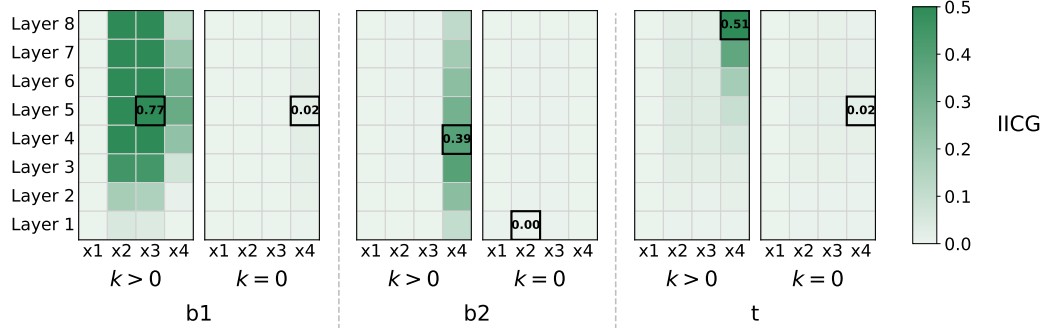

Figure 11: IICG heatmap for PARALLEL-2-HOP task with grouping strategies based on $b_1 = f_1(x_1, x_2)$ (**Left**), $b_2 = f_2(x_3, x_4)$ (**Middle**), and $t = f_3(b_1, b_2)$ (**Right**).

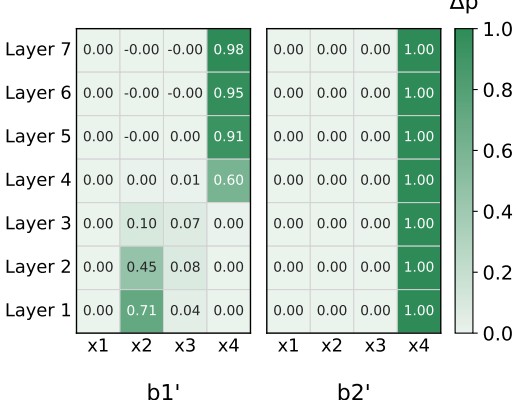

Figure 12: Causal tracing results showing indirect effect heatmap for PARALLEL-2-HOP task. **Left:** perturbation with different $(x_1, x_2)$ pair leading to a different $b_1$ value. **Right:** perturbation with different $(x_3, x_4)$ pair leading to a different $b_2$ value.

Similarly, Figs. 13 and 14 show results for the 3-HOP task. The IICG patterns exhibit strong representation clustering at mid-layers: at position $x_3$ when grouped by $b_1 = f_1(x_1, x_2)$, and at position $x_3$ when grouped by $b_2 = f_2(b_1, x_3)$. Causal tracing again confirms the causal importance of these representations.

These results demonstrate that the formation of clustered intermediate state representations and their causal role in compositional generalization is a consistent phenomenon across different task structures, supporting the generality of our findings beyond the 2-HOP task.

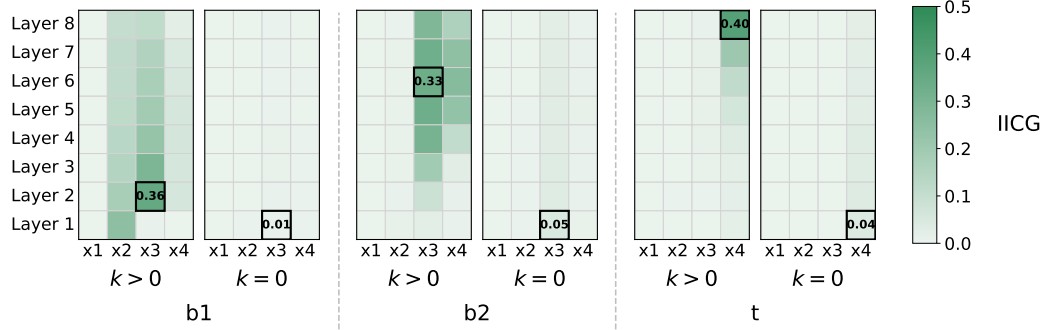

Figure 13: IICG heatmap for 3-HOP task with grouping strategies based on $b_1 = f_1(x_1, x_2)$ (**Left**), $b_2 = f_2(b_1, x_3)$ (**Middle**), and $t = f_3(b_2, x_4)$ (**Right**).

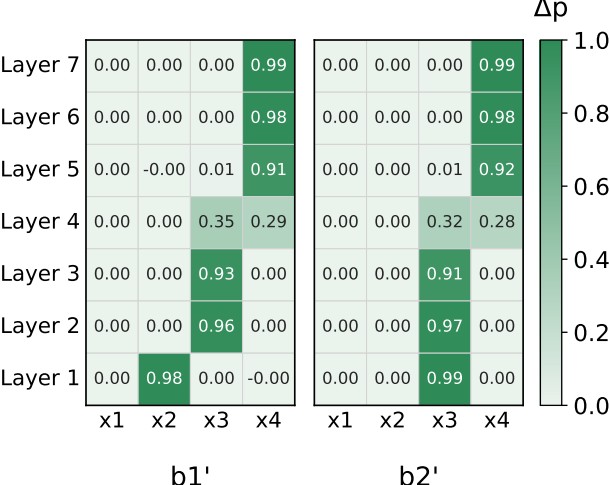

Figure 14: Causal tracing results showing indirect effect heatmap for 3-HOP task. **Left:** perturbation with different $(x_1, x_2)$ pair leading to different $b_1$ value. **Right:** perturbation leading to different $b_2 = f_2(b_1, x_3)$ value.

# E   PARTIAL COMPUTATION OBSERVATION DRIVES THE ALIGNMENT OF FUNCTIONAL EQUIVALENCE REPRESENTATION AND VOCABULARY SPACE

In this section, we investigate how exposure to partial computations affects the interpretability of intermediate state representations through vocabulary space alignment. We compare two training conditions on a modified 2-HOP task with $|\mathcal{X}| = 50$ and $N = 10k$, after 40k epochs of training:

1. **Standard Training**: $f_1 \neq f_2$, model only sees complete two-hop examples $(x_1, x_2, x_3) \mapsto t$.
2. **With Partial Computation**: $f_1 = f_2$, model additionally sees all possible partial computations $(x_1, x_2) \mapsto b$ where $b = f_1(x_1, x_2)$ (2,500 partial examples, not counted toward the $N = 10k$ two-hop training data).

To assess interpretability, we measure the Mean Reciprocal Rank (MRR) of intermediate state representations when projected to vocabulary space using the unembedding matrix. The low MRR indicates that the model's internal representation of intermediate state $b$ aligns with the corresponding vocabulary token.

Fig. 15 shows a striking contrast between the two conditions. Under standard training, the MRR score remains very high throughout training, indicating that intermediate representations are not aligned with vocabulary space despite the model successfully learning the compositional task. However, when partial computations are included, the MRR score becomes very high, demonstrating clear vocabulary alignment.

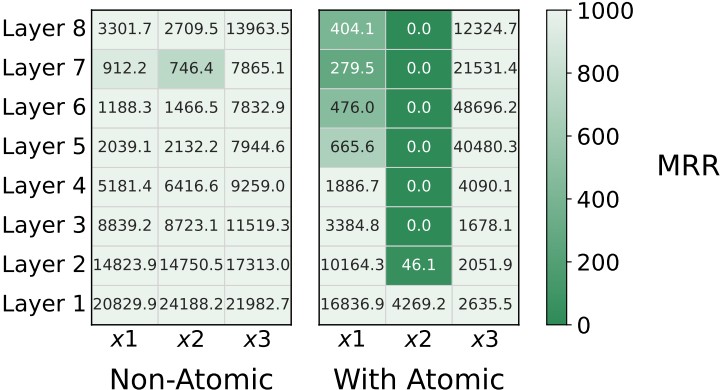

Figure 15: MRR scores for intermediate state representations projected to vocabulary space. **Left:** Standard training ($f_1 \neq f_2$, no partial computation) shows very high MRR regardless of position and layer. **Right:** Training with partial computation ($f_1 = f_2$, with partial examples) shows MRR of 0 in layers 3 to 8 at position $x_2$, indicating strong vocabulary alignment.

This experiment suggests that **logit lens interpretability is orthogonal to functional equivalence representation formation**. A model can develop functionally correct intermediate representations that enable compositional generalization while remaining completely uninterpretable through standard vocabulary projection techniques. Interpretability via logit lens requires explicit vocabulary anchoring through exposure to partial computations that map intermediate states to vocabulary tokens.

This finding has important implications for mechanistic interpretability research: the absence of interpretable representations through the logit lens does not indicate the absence of structured internal computation. Furthermore, it suggests that interpretability techniques may need to account for how training data shapes the alignment between internal representations and vocabulary space, rather than assuming such alignment emerges naturally from task performance.

# F   PROOFS: ASYMPTOTICALLY PERFECT COVERAGE IN 2-HOP TASK

We establish a scaling-law analysis for the 2-HOP task (as illustrated in Fig. 2a) within our formal framework for understanding pattern matching (Sec. 3). While a simplified statement (Theorem 6.1) is already presented in the main text, we provide the complete results of our analyses here in great detail, along with their rigorous and self-contained proofs. This appendix is long; we have organized it into the following structure.

- In App. F.1, we first review and (re)define the problem setting for 2-HOP task. Since the functional ($k$-)equivalence at indices 1 and 2 matters for the 2-HOP task, we argue that it is enough for us to focus on $I = \{1, 2\}$, thereby simplifying some key definitions (e.g., substitution graph, $k$-coverage) accordingly.

- In App. F.2, we state and prove our main theorems that characterize the number of training data sufficient for a learner to achieve a perfect in-domain generalization with high probability. There are two main theorems: one is for $k \geq 2$ (Theorem F.7) and another is for $k = 1$ (Theorem F.8). As a consequence, we further assume a specific regime of set cardinalities—namely, where all token sets have $\Theta(n)$ elements—and provide Corollary F.9, combining both cases for $k \geq 2$ and $k = 1$.

  - We also claim that our results are nearly *tight*, especially for $k \geq 2$. It is done by constructing a worst-case subclass of 2-HOP problems in which, given a dataset slightly smaller than our upper bound, the learner fails in achieving a perfect ID generalization with high probability (Theorem F.16 and Corollary F.17).

  Theorem 6.1 in the main text is indeed a simplified combination of Corollaries F.9 and F.17.

- In App. F.3, we provide (almost) all postponed proofs of lemmas used in App. F.2. We first prove that the *connectivity* of all $b$-evidence graphs (Def. F.5), the intermediate-state-specific induced subgraphs of the substitution graph, is a sufficient condition for the perfect in-domain generalization we want to show: see Lemma F.10.

  - App. F.3.1 is dedicated to explaining the renowned *Poissonization* technique, rigorously proving one of our main lemmas, Lemma F.12. Under a usual fixed-sized dataset sampling scheme, the edge connections of the substitution graph are not guaranteed to be independent. Nevertheless, the Poissonization technique enables us to disentangle these dependencies, thereby simplifying our analysis.

  - App. F.3.2 reviews the binomial random $k$-intersection graphs, a topic in the random graph theory literature. We bring some results on the threshold functions for their connectivity (without bringing their involved proofs). It is crucial because we show that the $b$-evidence graphs can be regarded as binomial random $k$-intersection graphs under the Poissonization.

## F.1   PRELIMINARIES: DEFINITIONS, ASSUMPTIONS, AND FACTS

**Ground-truth mapping.**   Consider a ground-truth mapping $f : \mathbb{X} = \mathcal{X}_1 \times \mathcal{X}_2 \times \mathcal{X}_3 \to Y$ defined as $f(x_1, x_2, x_3) = f_2(f_1(x_1, x_2), x_3)$, a *two-hop* composition of primitive functions $f_1 : \mathcal{X}_1 \times \mathcal{X}_2 \to B$ and $f_2 : B \times \mathcal{X}_3 \to Y$. Here, the set $B := f_1(\mathcal{X}_1 \times \mathcal{X}_2)$ is a collection of *intermediate states*, implicitly assuming surjectivity of $f_1$ without loss of generality.

**Train dataset.**   We take a train dataset $D$ by independently sampling input sequences $N$ times from the uniform distribution over $\mathbb{X} = \mathcal{X}_1 \times \mathcal{X}_2 \times \mathcal{X}_3$. For the sake of simplicity, we allow duplicates/replacements of train samples in $D$, thereby considering $D$ as a multiset.

We find it helpful in our analysis to define a random vector $(Z_{\boldsymbol{x}})_{\boldsymbol{x} \in \mathbb{X}}$ that has a one-to-one correspondence with a dataset constructed by sampling with replacement as described below.

**Proposition F.1.** *Given a train dataset $D$ and for each $\boldsymbol{x} \in \mathbb{X}$, define $Z_{\boldsymbol{x}}$ as the sampled count of $\boldsymbol{x}$ (i.e., the number of identical duplicates equal to $\boldsymbol{x}$) in $D$. In particular, if $D$ is uniformly randomly generated by with-replacement sampling $N$ times from $\mathbb{X}$, the random vector $(Z_{\boldsymbol{x}})_{\boldsymbol{x} \in \mathbb{X}}$ follows a multinomial distribution with parameters $n = N$ and $p_{\boldsymbol{x}} = \frac{1}{|\mathbb{X}|}$ ($\forall \boldsymbol{x} \in \mathbb{X}$). That is, for all collections of integers $z_{\boldsymbol{x}} \geq 0$ ($\forall \boldsymbol{x} \in \mathbb{X}$) such that $\sum_{\boldsymbol{x} \in \mathbb{X}} z_{\boldsymbol{x}} = N$,*

$$\mathbb{P}(Z_{\boldsymbol{x}} = z_{\boldsymbol{x}} \; \forall \boldsymbol{x} \in \mathbb{X}) = \frac{N!}{|\mathbb{X}|^N \cdot \prod_{\boldsymbol{x} \in \mathbb{X}}(z_{\boldsymbol{x}}!)}.$$

The one-to-one correspondence between $D$ and $(Z_{\boldsymbol{x}})_{\boldsymbol{x} \in \mathbb{X}}$ is straightforward: their realizations uniquely determine each other (up to the order of sampling).

**A learner making predictions using functional $k$-equivalence.** Let us consider an artificial learner that is perfectly trained on $D$ (i.e., always predicts correctly on $D$). We suppose that the learner confidently predicts the output for the elements in $k$-coverage of $D$ (defined with functional $k$-equivalence), while producing a random (or unconfident) prediction outside of it. However, the confident prediction is not necessarily correct in general. Luckily, since we suppose a clear two-hop compositional structure of the task, the confident predictions based on functional equivalences at indices $I = \{1, 2\}$ are guaranteed to be correct. Hence, to study the scope of pattern-matching-based in-domain (ID) generalization as confident and correct predictions on unseen inputs, we restrict the scope of our analysis to the index set $\{1, 2\}$ for our analysis in this appendix.

To formalize how far the learner can generalize (Def. F.4) and how far we hope the learner to generalize (Def. F.2), let us (re)define the key terms for our analysis of the two-hop task, based on the restriction with respect to the index set $I = \{1, 2\}$. For brevity, let us write $x_{12} := (x_1, x_2) \in \mathcal{X}_{12} := \mathcal{X}_1 \times \mathcal{X}_2$.

**Definition F.2** (In-domain closure, in terms of $I = \{1, 2\}$). For a train dataset $D$, its **in-domain closure** $\overline{D}$ is defined as

$$\overline{D} = \{(x_{12}, x_3) \in \mathbb{X} : \exists (x'_{12}, x'_3) \in \mathbb{X} \text{ such that } f_1(x_{12}) = f_1(x'_{12}), (x'_{12}, x_3) \in D, \text{ and } (x_{12}, x'_3) \in D\}.$$

Also, its element $(x_{12}, x_3) \in \overline{D}$ is said to be an **in-domain data**.

According to the definition above, the in-domain closure $\overline{D}$ contains all input sequences $(x_{12}, x_3)$ such that (i) its subsequences $x_{12}(= \boldsymbol{x}_I)$ and $x_3(= \boldsymbol{x}_{I^c})$ has already been observed in the train dataset $D$, and (ii) its output can be inferred using functional equivalence at indices $I = \{1, 2\}$, namely, $f(x_{12}, x_3) = f(x'_{12}, x_3)$ for a training data $(x'_{12}, x_3) \in D$. Hence, the set $\overline{D}$ is the largest set of input sequences whose output can be correctly predicted using the exact task structure (e.g., the functional equivalence at $\{1, 2\}$). We also remark that it is obvious to see that $D \subset \overline{D}$. We wish the learner would acquire such prediction capabilities from a training dataset. Nevertheless, the task structure is not directly accessible to our learner since it can only identify the necessary functional relationships through functional $k$-equivalences (Def. 3.1).

On the other hand, the $k$-coverage quantifies how far the assumed learner can actually make correct predictions using the observed functional $k$-equivalences from the training dataset $D$. The definition (Def. F.4) resembles that of the connectivity of a graph with a vertex set $\mathcal{X}_{12}$.

Recall that in our main text, we defined a *substitution graph* with the entire input space $\mathbb{X}$ as its vertex set since we aim to check all functional equivalences across all index sets. However, since we are particularly interested in the functional equivalences at $I = \{1, 2\}$, we can simplify its definition as below.

**Definition F.3** (Substitution graph, in terms of $I = \{1, 2\}$). For a train dataset $D$ and a positive integer $k$, we define a **substitution graph** (in terms of $I = \{1, 2\}$) as $\mathcal{G}_{\bullet}^{(D,k)} = (V_{\bullet}, E_{\bullet}^{(D,k)})$. Here, the vertex set is $V_{\bullet} := \mathcal{X}_{12}$ and the edge set is

$$E_{\bullet}^{(D,k)} := \{\{x_{12}, x'_{12}\} \subset V_{\bullet} : x_{12} \neq x'_{12} \text{ and } |S_f(x_{12}, x'_{12} \mid D)| \geq k\},$$

where the set $S_f(x_{12}, x'_{12} \mid D)$ is defined as

$$S_f(x_{12}, x'_{12} \mid D) := \{\bar{x}_3 \in \mathcal{X}_3 : f(x_{12}, \bar{x}_3) = f(x'_{12}, \bar{x}_3), (x_{12}, \bar{x}_3) \in D, (x'_{12}, \bar{x}_3) \in D\}.$$

In other words, in the substitution graph $\mathcal{G}_{\bullet}^{(D,k)}$, two vertices $x_{12}$ and $x'_{12}$ are adjacent if they have at least $k$ distinct co-occerences in $D$. With this definition, we define the $k$-coverage as follows.

**Definition F.4** ($k$-coverage, in terms of $I = \{1, 2\}$). For a train dataset $D$ and a positive integer $k$, the $k$**-coverage** of $D$ (in terms of $I = \{1, 2\}$) is a subset of $\mathbb{X}$ defined as

$\text{Cover}_k(D)$

$$\stackrel{\text{def}}{=} \Big\{(x_{12}, x_3) \in \mathbb{X} : \exists x'_{12} \in \mathcal{X}_{12} \text{ connected to } x_{12} \text{ with a path in } \mathcal{G}_{\bullet}^{(D,k)}, \text{ and } (x'_{12}, x_3) \in D\Big\}$$

$$= \Big\{(u_0, x_3) \in \mathbb{X} : \exists \ell \geq 0, \exists u_0, u_1, \cdots, u_\ell \in \mathcal{X}_{12} \text{ such that } \{u_{i-1}, u_i\} \in E_{\bullet}^{(D,k)} \ (\forall i \in [\ell]) \ \& \ (u_\ell, x_3) \in D\Big\},$$

where $[\ell] = \{1, 2, 3, \cdots, \ell\}$. Also, an element $(x_{12}, x_3) \in \text{Cover}_k(D)$ is said to be **covered**.

Again, we can easily verify that $D \subset \text{Cover}_k(D)$ by choosing $\ell = 0$ in the definition above. However, the inclusion relation between the in-domain coverage ($\overline{D}$) and the $k$-coverage ($\text{Cover}_k(D)$) cannot be explicitly determined in general. From the next sub-section, we aim to analyze when (or, with how

large $D$) we do have a relation "$\overline{D} \subset \mathrm{Cover}_k(D)$", which we call **perfect coverage** of in-domain data, with high probability (with respect to the randomness of $D$).

We additionally introduce some useful symbols for our later proofs. We denote by $B_D$ the set of all intermediate states observed in $D$: i.e.,

$$B_D := \{b \in B : \exists (x_{12}, x_3) \in D \text{ such that } f_1(x_{12}) = b\}.$$

Moreover, we introduce the *b-evidence graph* $\mathcal{G}_b^{(D,k)}$, defined as a subgraph of $\mathcal{G}_\bullet^{(D,k)}$ induced by the set $V_b = f_1^{-1}(\{b\}) := \{x_{12} \in \mathcal{X}_{12} : f_1(x_{12}) = b\} \subset V_\bullet$ of vertices sharing the same intermediate state $b \in B$.

**Definition F.5** (Evidence graphs). For a train dataset $D$, a positive integer $k$, and an intermediate state $b \in B$, we define a *b*-**evidence graph** (in terms of $I = \{1, 2\}$) as $\mathcal{G}_b^{(D,k)} = (V_b, E_b^{(D,k)})$. Here, the vertex set is $V_b := f_1^{-1}(\{b\})$ and the edge set is

$$E_b^{(D,k)} := \left\{ \{x_{12}, x'_{12}\} \subset V_b : x_{12} \neq x'_{12} \text{ and } |S(x_{12}, x'_{12} \mid D)| \geq k \right\}, \tag{4}$$

where the set $S(x_{12}, x'_{12} \mid D)$ is defined as

$$S(x_{12}, x'_{12} \mid D) := \{\bar{x}_3 \in \mathcal{X}_3 : (x_{12}, \bar{x}_3) \in D, (x'_{12}, \bar{x}_3) \in D\}. \tag{5}$$

Lastly, we introduce an assumption that will be assumed only when we claim the tightness of our sample complexity upper bound results.

**Assumption F.6.** The learner uses the parameter $k \geq k_\star + 1$, where $k_\star \geq 0$ is defined as

$$k_\star = \max_{(b,b') \in B^2} |H(b, b')| \qquad \text{subject to} \quad b \neq b',$$

and $H(b, b') = \{x_3 \in \mathcal{X}_3 : f_2(b, x_3) = f_2(b', x_3)\}$.

According to the assumption above, $f_2(b, x_3) = f_2(b', x_3)$ can hold for more than $k_\star$ elements in $\mathcal{X}_3$ only when $b = b'$. It is quite a natural assumption for general 2-HOP tasks since it naturally eliminates pathological examples that can barely be regarded as 2-HOP tasks (e.g., a constant function $f$ on $\mathbb{X}$).

The assumption above additionally assumes that our learner *pessimistically* recognizes the functional equivalences between $\{1, 2\}$-subsequences, by using $k \geq k_\star + 1$. If $k = k_\star + 1$, the learner is using the precise value of $k$ to recognize functionally equivalent subsequences sharing the same intermediate state; a higher value of $k$ indicates a more pessimistic learner. Because of this pessimism, once the learner recognizes that $x_{12}$ and $x'_{12}$ are functionally $k$-equivalent (i.e., $\{x_{12}, x'_{12}\} \in E_\bullet^{(D,k)}$), it is guaranteed that $x_{12}$ and $x'_{12}$ share the same intermediate state (i.e., $f_1(x_{12}) = f_1(x'_{12})$). More importantly, it implies that there must be no edges (in $E_\bullet^{(D,k)}$) connecting two graphs $\mathcal{G}_b^{(D,k)}$ and $\mathcal{G}_{b'}^{(D,k)}$ for distinct $b, b' \in B$; otherwise, there must be a pair of subsequences $x_{12} \in V_b$ and $x'_{12} \in V_{b'}$ that are functionally $k$-equivalent, contradicting the assumption.

Even though Assumption F.6 appears to pertain to the learning rule of the learner, it can also be interpreted as an assumption about the ground-truth mapping, which necessitates $k_\star \leq k - 1$ for a given $k \geq 1$.

## F.2 MAIN THEOREM: SAMPLE COMPLEXITY BOUND FOR PERFECT COVERAGE WITH HIGH PROBABILITY

The ultimate goal of this appendix is to characterize a sufficient number of training data $|D|$ (i.e., sample complexity upper bound) so that the $k$-coverage includes all in-domain data, i.e., $\overline{D} \subset \mathrm{Cover}_k(D)$, with high probability. Here, we will state and prove our main theorems, Theorem F.7 (for $k \geq 2$) and Theorem F.8 (for $k = 1$), which are followed by a simplified combination of them (Corollary F.9).

Soon after, we will also claim the near-tightness of these theorems by showing that, for $k \geq 2$, the training datasets slightly smaller than the obtained sample complexity are sufficient to guarantee $\overline{D} \not\subset \mathrm{Cover}_k(D)$ with high probability (Theorem F.16 and Corollary F.17). Although we present the proofs of the theorems here, the complete proofs of most lemmas used in the main proofs are postponed to the later part of this appendix (App. F.3).

Here are the statements of our main theorems for the sample complexity upper bounds, which will be proved soon after.

**Theorem F.7** (Sample Complexity Upper Bound, $k \geq 2$, $\downarrow$). *Under the two-hop task setup described in App. F.1, suppose that $k \geq 2$ is a universal constant. Let $\hat{n} := \min_{b \in B_D} |V_b|$ and $\check{n} := \max_{b \in B_D} |V_b|$, and assume that $|\mathcal{X}_3| = \Omega(\check{n})$. Fix any $\delta > 0$. Then, there exists $N_\delta > 0$ satisfying*

$$N_\delta \leq O \left( \max \left\{ \left( \frac{k! \, (\ln \hat{n})}{\hat{n}} \right)^{\frac{1}{2k}} \cdot |\mathcal{X}_1 \times \mathcal{X}_2| \sqrt{|\mathcal{X}_3|}, \, \ln \left( \frac{1}{\delta} \right) \right\} \right)$$

*such that, for any large enough $\hat{n}$ and any integer $N > N_\delta$, we have*

$$\mathbb{P}_N \left( \overline{D} \subset \mathrm{Cover}_k(D) \right) \geq 1 - \delta.$$

*Here, $\mathbb{P}_N$ is the probability measure for the train dataset $D$ constructed by uniformly randomly sampling its elements $N$ times, allowing duplicates, independently from $\mathbb{X} = \mathcal{X}_1 \times \mathcal{X}_2 \times \mathcal{X}_3$.*

**Theorem F.8** (Sample Complexity Upper Bound, $k = 1$, $\downarrow$). *Under the two-hop task setup described in App. F.1, suppose that $k = 1$. Let $\hat{n} := \min_{b \in B_D} |V_b|$ and $\check{n} := \max_{b \in B_D} |V_b|$, and assume that $|\mathcal{X}_3| = O(\hat{n})$. Fix any $\delta > 0$. Then, there exists $N_\delta > 0$ satisfying*

$$N_\delta \leq O \left( \max \left\{ (\ln \check{n}) \cdot |\mathcal{X}_1 \times \mathcal{X}_2|, \, \ln \left( \frac{1}{\delta} \right) \right\} \right)$$

*such that, for any large enough $\hat{n}$ and any $N > N_\delta$, we have*

$$\mathbb{P}_N \left( \overline{D} \subset \mathrm{Cover}_k(D) \right) \geq 1 - \delta.$$

*We use the same definition of $\mathbb{P}_N$ as in Theorem F.7.*

In particular, these two theorems can be easily combined (hence the detailed proof is skipped) in the regime of almost balanced cardinalities as $\Theta(n)$, which is described in the following corollary.

**Corollary F.9** (Power-Law Sample Complexity Upper Bound, Simple). *Under the two-hop task setup described in App. F.1, Assume that $\mathcal{X}_1$, $\mathcal{X}_2$, $\mathcal{X}_3$, and $V_b$ ($\forall b \in B$) are all the sets of size $\Theta(n)$. Fix any $k \geq 1$ and any $\delta > 0$ Then, there exists $N_\delta(n) > 0$ satisfying*

$$N_\delta(n) \leq O \left( \max \left\{ n^{2.5 - \frac{0.5}{k}} \cdot (k! \cdot \ln n)^\gamma, \, \ln \frac{2}{\delta} \right\} \right), \qquad with \; \gamma = \begin{cases} 1, & for \; k = 1; \\ \frac{1}{2k}, & for \; k \geq 2, \end{cases}$$

*such that, for any large enough $n$ and any $N > N_\delta(n)$, the learner with a uniformly randomly sampled training dataset $D$ (with replacements) achieves a perfect coverage of in-domain data, i.e., $\overline{D} \subset \mathrm{Cover}_k(D)$, with probability at least $1 - \delta$.*

Now, we prove our main theorem for $k \geq 2$ (Theorem F.7). After that, we will present the proof of the other case ($k = 1$, Theorem F.8), which follows a similar plot except for a few steps at the end.

*Proof of Theorem F.7.* We begin the proof with the following observation: a sufficient condition for a perfect coverage is that every $b$-evidence graph (Def. F.5) is a connected graph.

**Lemma F.10** ($\downarrow$). *If all $\mathcal{G}_b^{(D,k)}$ are connected graphs ($\forall b \in B_D$), then we have $\overline{D} \subset \mathrm{Cover}_k(D)$.*

Refer to App. F.3 for the proof. Thanks to the lemma above, it suffices to show the following with sufficiently large $N = |D|$:

$$\mathbb{P}_N \left( \overline{D} \not\subset \mathrm{Cover}_k(D) \right) \overset{\text{Lemma F.10}}{\leq} \mathbb{P}_N \left( \exists b \in B_D \text{ such that } \mathcal{G}_b^{(D,k)} \text{ is disconnected} \right) \leq \delta.$$

To this end, consider the multinomial random vector $(Z_{\boldsymbol{x}})_{\boldsymbol{x} \in \mathbb{X}}$ corresponding to a randomly sampled training dataset $D$, as defined in Proposition F.1. Define a property $\mathcal{C}$ as

$$(Z_{\boldsymbol{x}})_{\boldsymbol{x} \in \mathbb{X}} \in \mathcal{C} \iff \mathcal{G}_b^{(D,k)} \text{ is connected } \forall b \in B_D. \tag{6}$$

Then, we want to compute an upper bound of a probability $\mathbb{P}_N \left( (Z_{\boldsymbol{x}})_{\boldsymbol{x} \in \mathbb{X}} \notin \mathcal{C} \right)$. Observe that the property $\mathcal{C}$ is a *non-decreasing* property (Def. F.11): if a vector $(Z_{\boldsymbol{x}})_{\boldsymbol{x} \in \mathbb{X}}$ satisfies $\mathcal{C}$, then it still satisfies $\mathcal{C}$ after increasing in its entries by non-negative amounts. In other words, the negation of $\mathcal{C}$ ("not satisfying $\mathcal{C}$") is a *non-increasing* property. This is because adding a data point to the training dataset $D$ does not remove any edge from all evidence graphs, thereby preserving the connectivity. In general, we define the *monotone* property of vectors as below.

**Definition F.11** (Monotone property). We refer to a property $\mathcal{A}$ of $m$-dimensional vectors as a **non-decreasing property** when, for any vector $(\Delta_1, \cdots, \Delta_m)$ with non-negative entries $\Delta_i \geq 0$ ($\forall i = 1, \cdots, m$),

$$(z_1, \cdots, z_m) \in \mathcal{A} \implies (z_1 + \Delta_1, \cdots, z_m + \Delta_m) \in \mathcal{A}. \qquad \text{(Non-Decreasing Property)}$$

On the other hand, we define **non-increasing property** $\mathcal{A}$ as what satisfies

$$(z_1, \cdots, z_m) \in \mathcal{A} \implies (z_1 - \Delta_1, \cdots, z_m - \Delta_m) \in \mathcal{A}. \qquad \text{(Non-Increasing Property)}$$

Taking advantage of the monotonicity, we can apply the Poissonization technique (App. F.3.1) to obtain an upper bound on the probability $\mathbb{P}_N \left( (Z_{\boldsymbol{x}})_{\boldsymbol{x} \in \mathbb{X}} \notin \mathcal{C} \right)$. The Poissonization technique we use is summarized as the lemma below, which will be proved in App. F.3.1:

**Lemma F.12** (De-Poissonization of Monotone Multinomial Events, ↓). *Fix any $n \geq 1$. Define*

$$P_n := \mathbb{P}_{(Z_1, \cdots, Z_m) \sim \text{Multinomial}(n; \, p_1, \cdots, p_m)} \left( (Z_1, \cdots, Z_m) \in \mathcal{A} \right).$$

*Let $\mathcal{A}$ be a non-decreasing property (Def. F.11) of $m$-dimensional vector. Then, we have an upper bound and a lower bound for $P_n$ as follows:*

$$P_n \leq \mathbb{P}_{\text{Po}(n+\varepsilon)} \left( (Z_1, \cdots, Z_m) \in \mathcal{A} \right) + \exp \left( -\frac{(3-c)\varepsilon^2}{6n} \right), \qquad (\forall c \in (0,3), \; \forall \varepsilon \in (0, cn))$$

$$P_n \geq \mathbb{P}_{\text{Po}(n-\varepsilon)} \left( (Z_1, \cdots, Z_m) \in \mathcal{A} \right) - \exp \left( -\frac{\varepsilon^2}{2n} \right). \qquad (\forall \varepsilon \in (0, n))$$

*If $\mathcal{A}$ is a non-increasing property (Def. F.11), then we have similar upper and lower bounds for $P_n$:*

$$P_n \leq \mathbb{P}_{\text{Po}(n-\varepsilon)} \left( (Z_1, \cdots, Z_m) \in \mathcal{A} \right) + \exp \left( -\frac{\varepsilon^2}{2n} \right), \qquad (\forall \varepsilon \in (0, n))$$

$$P_n \geq \mathbb{P}_{\text{Po}(n+\varepsilon)} \left( (Z_1, \cdots, Z_m) \in \mathcal{A} \right) - \exp \left( -\frac{(3-c)\varepsilon^2}{6n} \right). \qquad (\forall c \in (0,3), \; \forall \varepsilon \in (0, cn))$$

*Here, we denote by $\mathbb{P}_{\text{Po}(\lambda)}$ the probability measure under $Z_i \overset{\text{indep.}}{\sim} \text{Poisson}(\lambda p_i)$ ($\forall i = 1, \cdots, m$).*

In particular, considering the disconnectedness ($\exists b \in B_D$) as a non-increasing property, we have

$$\mathbb{P}_N \left( \exists b \in B_D \text{ such that } \mathcal{G}_b^{(D,k)} \text{ is disconnected} \right)$$

$$\leq \mathbb{P}_{\text{Po}(N-\varepsilon)} \left( \exists b \in B_D \text{ such that } \mathcal{G}_b^{(D,k)} \text{ is disconnected} \right) + \exp \left( -\frac{\varepsilon^2}{2N} \right)$$

$$\leq \sum_{b \in B_D} \mathbb{P}_{\text{Po}(N-\varepsilon)} \left( \mathcal{G}_b^{(D,k)} \text{ is disconnected} \right) + \exp \left( -\frac{\varepsilon^2}{2N} \right) \qquad (\because \text{union bound})$$

for any positive number $\varepsilon < N$, where $\mathbb{P}_{\text{Po}(N-\varepsilon)}$ is the probability measure under i.i.d. Poisson random variables $Z_{\boldsymbol{x}} \sim \text{Poisson} \left( \frac{N-\varepsilon}{|\mathbb{X}|} \right)$ ($\forall \boldsymbol{x} \in \mathbb{X}$). Let us take $\varepsilon = \sqrt{2N \ln \left( \frac{2}{\delta} \right)}$, which is smaller than $N$ by our choice of $N_\delta$ and ensures that

$$\exp \left( -\frac{\varepsilon^2}{2N} \right) = \exp \left( -\ln \left( \frac{2}{\delta} \right) \right) = \frac{\delta}{2}.$$

Now, we claim that each $b$-evidence graph $\mathcal{G}_b^{(D,k)}$ is a binomial random $k$-intersection graph (App. F.3.2) for every $b \in B_D$, under the Poissonization governed by $\mathbb{P}_{\text{Po}(N-\varepsilon)}$.

**Lemma F.13** (Evidence Graphs are Binomial $k$-Intersection Graphs under Poissonization, ↓). *Let any $b \in B_D$. Consider a vector $(Z_{\boldsymbol{x}})_{\boldsymbol{x} \in \mathbb{X}}$ associated with a dataset $D$ (i.e., an input sequence $\boldsymbol{x}$ is sampled $Z_{\boldsymbol{x}}$ times in $D$). Let $\mathbb{P}_{\text{Po}(\lambda)}$ be a probability measure for $Z_{\boldsymbol{x}} \overset{i.i.d.}{\sim} \text{Poisson} \left( \lambda / |\mathbb{X}| \right)$. Then,*

under $\mathbb{P}_{\mathrm{Po}(\lambda)}$, the b-evidence graph $\mathcal{G}_b^{(D,k)}$ (Def. F.5) is an instance of binomial random $k$-intersection graph $\mathcal{G}^{(k)}(n_b, m, p)$ with parameters

$$n_b = |V_b|, \quad m = |\mathcal{X}_3|, \quad p = 1 - \exp\left(-\frac{\lambda}{|\mathbb{X}|}\right).$$

Based on such an observation, we can use the following seminal result about the connectivity of binomial random $k$-intersection graphs with $p = 1 - \exp(-(N-\varepsilon)/|\mathbb{X}|)$. In particular, we will use only the "$k \geq 2$" part (which tightly matches our assumption $m = \Omega(n_b)$ ($\forall b \in B_D$)) below:

**Lemma F.14** (Zhao et al., 2014, Theorem 2; Zhao et al., 2017, Theorem 1 & Remark 1). *Fix any $k \geq 1$. Suppose that*

$$m = \begin{cases} \Omega\left(\min\left\{n\left(\ln n\right)^5, n^\rho\right\}\right), & \text{if } k = 1, \text{ for any } \rho > 1; \\ \Omega\left(n\right), & \text{if } k \geq 2, \end{cases}$$

*and*

$$p = \left(\frac{k!\,(\ln n + \alpha_n)}{n}\right)^{\frac{1}{2k}} \cdot \frac{1}{\sqrt{m}} \tag{7}$$

*for any sequence $\{\alpha_n\}$ which attains a limit $\alpha_\infty \in [-\infty, +\infty]$ as $n \to \infty$. Then,*

$$\lim_{n \to \infty} \mathbb{P}\left(\mathcal{G}^{(k)}(n,m,p) \text{ is connected}\right) = \lim_{n \to \infty} \mathbb{P}\left(\min \deg \mathcal{G}^{(k)}(n,m,p) \geq 1\right) = \exp\left(-e^{-\alpha_\infty}\right),$$

*where we compute $\exp\left(-e^{-(-\infty)}\right) = 0$ and $\exp\left(-e^{-(+\infty)}\right) = 1$.*

Now, we aim to find a sufficient condition for $N$ to have

$$\sum_{b \in B_D} \mathbb{P}_{\mathrm{Po}(N-\varepsilon)}\left(\mathcal{G}_b^{(D,k)} \text{ is disconnected}\right) \leq \frac{\delta}{2}. \tag{8}$$

Let us choose the sequence $\alpha_n = \ln n$ in Lemma F.14, which guarantees that

$$\mathbb{P}(\mathcal{G}^{(k)}(n,m,p) \text{ is connected}) \to 1 \quad \text{as } n \to \infty.$$

From the definition of the limit of a sequence, for any choice of $\delta > 0$, let us define $n_0(\delta) > e > 0$ such that

$$\mathbb{P}(\mathcal{G}^{(k)}(n,m,p) \text{ is disconnected}) \leq \delta, \qquad \forall n > n_0(\delta).$$

Then, choosing all $n_b = |V_b|$ to be greater than $n_0\left(\frac{\delta}{2|B_D|}\right)$, we yield the bound (8). Note that a choice of $p$ larger than the threshold in Eq. (7) will never change that $\mathbb{P}(\mathcal{G}^{(k)}(n,m,p) \text{ is connected}) \to 1$ as $n \to \infty$. Thus, since we choose $\alpha_n = \ln n$, it suffices to have

$$p = 1 - \exp\left(-\frac{N-\varepsilon}{|\mathbb{X}|}\right) \geq \left(\frac{k!\,(2\ln n_b)}{n_b}\right)^{\frac{1}{2k}} \cdot \frac{1}{\sqrt{|\mathcal{X}_3|}}$$

for each $n_b$ ($\forall b \in B_D$). Since $1 - e^{-u} \geq (1 - \frac{1}{e})u$ for $0 < u < 1$ and $\frac{\ln u}{u}$ is a decreasing function for large enough $u > e$, it suffices to have

$$\left(1 - \frac{1}{e}\right)\frac{N-\varepsilon}{|\mathbb{X}|} \geq \left(\frac{k!\,(2\ln \hat{n})}{\hat{n}}\right)^{\frac{1}{2k}} \cdot \frac{1}{\sqrt{|\mathcal{X}_3|}}$$

for $\hat{n} := \min_{b \in B_D} n_b$. Plugging in $\varepsilon = \sqrt{2N \ln\left(\frac{2}{\delta}\right)}$ and $|\mathbb{X}| = |\mathcal{X}_1 \times \mathcal{X}_2|\,|\mathcal{X}_3|$, we have

$$N - \sqrt{2N \ln\left(\frac{2}{\delta}\right)} \geq \left(\frac{e}{e-1}\right) \cdot \left(\frac{k!\,(2\ln \hat{n})}{\hat{n}}\right)^{\frac{1}{2k}} \cdot |\mathcal{X}_1 \times \mathcal{X}_2|\,\sqrt{|\mathcal{X}_3|},$$

which is satisfied by

$$N \geq 4 \cdot \max \left\{ \left( \frac{e}{e-1} \right) \cdot \left( \frac{k! \ (2 \ln \hat{n})}{\hat{n}} \right)^{\frac{1}{2k}} \cdot |\mathcal{X}_1 \times \mathcal{X}_2| \sqrt{|\mathcal{X}_3|}, \ \ln \left( \frac{2}{\delta} \right) \right\}.$$

In conclusion, for large enough $\hat{n} > \max \left\{ n_0 \left( \frac{\delta}{2|B_D|} \right), e \right\}$, provided that the training dataset size $N$ satisfies the inequality above, we finally have

$$\mathbb{P}_N \left( \exists b \in B_D \text{ such that } \mathcal{G}_b^{(D,k)} \text{ is disconnected} \right)$$

$$\leq \sum_{b \in B_D} \mathbb{P}_{\mathrm{Po}(N-\varepsilon)} \left( \mathcal{G}_b^{(D,k)} \text{ is disconnected} \right) + \exp \left( -\frac{\varepsilon^2}{2N} \right)$$

$$\leq \frac{\delta}{2} + \frac{\delta}{2} = \delta.$$

This concludes the proof of the theorem. ∎

*Proof of Theorem F.8.* Even for $k = 1$, we follow almost the same proof as that of Theorem F.7, except for the last few steps. Namely, we again observe that every $\mathcal{G}_b^{(D,1)}$ is a binomial random 1-intersection graph $\mathcal{G}^{(1)}(n_b, m, p)$ with parameters $n_b = |V_b|$, $m = |\mathcal{X}_3|$, and $p = 1 - \exp(-(N - \varepsilon)/|\mathbb{X}|)$ (see the definition of binomial random intersection graph in App. F.3.2). To guarantee the union bound in Eq. (8), however, we use a different lemma: Lemma F.15 (instead of Lemma F.14). In particular, we will use only the "$\rho \leq 1$" part (which tightly matches our assumption $m = O(n_b)$ $(b \in B_D)$) of the lemma below:

**Lemma F.15** (Singer, 1995, Propositions 3.1–2, Theorem 3.3). *Let $k = 1$. Suppose that $m = n^\rho$ for $\rho > 0$ and*

$$p = \begin{cases} \dfrac{\ln n + \alpha_n}{m} & \text{for } \rho \leq 1; \\ \sqrt{\dfrac{\ln n + \alpha_n}{mn}} & \text{for } \rho > 1, \end{cases}$$

*for any sequence $\{\alpha_n\}$ which attains a limit $\alpha_\infty \in \{-\infty, +\infty\}$ as $n \to \infty$. Then,*

$$\lim_{n \to \infty} \mathbb{P}(\mathcal{G}^{(1)}(n, m, p) \text{ is connected}) = \lim_{n \to \infty} \mathbb{P} \left( \min \deg \mathcal{G}^{(1)}(n, m, p) \geq 1 \right) = \begin{cases} 0, & \text{if } \alpha_\infty = -\infty; \\ 1, & \text{if } \alpha_\infty = +\infty. \end{cases}$$

Again, let us choose the sequence $\alpha_n = \ln n$ in Lemma F.15, which guarantees that $\mathbb{P}(\mathcal{G}^{(1)}(n, m, p) \text{ is connected}) \to 1$ as $n \to \infty$. From the definition of the limit of a sequence, for any choice of $\delta > 0$, let us define $n_0(\delta) > 1$ such that

$$\mathbb{P}(\mathcal{G}^{(1)}(n, m, p) \text{ is disconnected}) \leq \delta, \qquad \forall n > n_0(\delta).$$

Then, choosing all $n_b = |V_b|$ (and thus $\hat{n} = \min_{b \in B_D} n_b$) to be greater than $n_0 \left( \frac{\delta}{2|B_D|} \right)$, we yield the same bound (8). Now, by our choice $\alpha_n = \ln n$, it suffices to have

$$p = 1 - \exp \left( -\frac{N - \varepsilon}{|\mathbb{X}|} \right) \geq \frac{2 \ln n_b}{|\mathcal{X}_3|}$$

for each $n_b$ ($\forall b \in B_D$). Since $1 - e^{-u} \geq (1 - \frac{1}{e})u$ for $0 < u < 1$ and $\ln u$ is an increasing function for $u > 1$, it suffices to have

$$\left( 1 - \frac{1}{e} \right) \frac{N - \varepsilon}{|\mathbb{X}|} \geq \frac{2 \ln \check{n}}{|\mathcal{X}_3|}$$

for $\check{n} := \max_{b \in B_D} n_b$. Plugging in $\varepsilon = \sqrt{2N \ln \left( \frac{2}{\delta} \right)}$ and $|\mathbb{X}| = |\mathcal{X}_1 \times \mathcal{X}_2| |\mathcal{X}_3|$, we have

$$N - \sqrt{2N \ln \left( \frac{2}{\delta} \right)} \cdot \geq \left( \frac{e}{e-1} \right) (2 \ln \check{n}) \cdot |\mathcal{X}_1 \times \mathcal{X}_2|,$$

which is satisfied by

$$N \geq 4 \cdot \max \left\{ \left( \frac{e}{e-1} \right) \cdot (2 \ln \check{n}) \cdot |\mathcal{X}_1 \times \mathcal{X}_2|, \; \ln \left( \frac{2}{\delta} \right) \right\}.$$

In conclusion, for large enough $\check{n} \geq \hat{n} > \max \left\{ n_0 \left( \frac{\delta}{2|B_D|} \right), 1 \right\}$, provided that the training dataset size $N$ satisfies the inequality above, we finally have

$$\mathbb{P}_N \left( \exists b \in B_D \text{ such that } \mathcal{G}_b^{(D,1)} \text{ is disconnected} \right)$$

$$\leq \sum_{b \in B_D} \mathbb{P}_{\mathrm{Po}(N-\varepsilon)} \left( \mathcal{G}_b^{(D,1)} \text{ is disconnected} \right) + \exp \left( -\frac{\varepsilon^2}{2N} \right)$$

$$\leq \frac{\delta}{2} + \frac{\delta}{2} = \delta.$$

This concludes the proof. ∎

Subsequently, we argue that the sample complexity upper bound obtained in Theorem F.7 (for $k \geq 2$) is *nearly tight*.[7] To this end, we show that the learner (we assumed in App. F.1) cannot avoid an *incomplete* coverage (i.e., $\overline{D} \not\subset \mathrm{Cover}_k(D)$) with high probability, with a dataset slightly smaller than our upper bound for certain instances of the 2-HOP task. In particular, it is enough to consider a subclass of the 2-HOP task satisfying a mild condition (Assumption F.6) that any two distinct intermediate states in $B$ never share the same output for more than $k_\star (< k)$ tokens in $\mathcal{X}_3$. The formal statement of the tightness result is shown below:

---

**Theorem F.16** (Near-Tightness of the Sample Complexity Upper Bound for $k \geq 2$, ↓). *Under the two-hop task setup described in App. F.1, suppose that $k \geq 2$ is a universal constant. Let $\hat{n} := \min_{b \in B} |V_b|$ and $\check{n} := \max_{b \in B} |V_b|$. Assume that $|\mathcal{X}_3| = \Omega(\hat{n})$ and $|\mathcal{X}_1 \times \mathcal{X}_2| \geq 6$. Let us further assume that the ground-truth mapping $f$ satisfies Assumption F.6. Fix any $\delta > 0$. Then, there exists $\widetilde{N}_\delta > 0$ satisfying*

$$\widetilde{N}_\delta \geq \left( \frac{(k-1)! \, (\ln \hat{n})}{\hat{n}} \right)^{\frac{1}{2k}} \cdot |\mathcal{X}_1 \times \mathcal{X}_2| \sqrt{|\mathcal{X}_3|}$$

*such that, for any large enough $\hat{n}$ and any integer $N$ satisfying the range*

$$\max \left\{ |\mathcal{X}_1 \times \mathcal{X}_2| \ln \frac{4 |\mathcal{X}_1 \times \mathcal{X}_2|}{\delta}, \; \frac{|\mathbb{X}|}{\hat{n}} \ln \frac{4 |B \times \mathcal{X}_3|}{\delta} \right\} \leq N < \widetilde{N}_\delta$$

*we have*

$$\mathbb{P}_N \left( \overline{D} \subset \mathrm{Cover}_k(D) \right) \leq \delta.$$

*We use the same definition of $\mathbb{P}_N$ as in Theorem F.7.*

---

A caveat of Theorem F.16 is that it makes sense only when the range of $N$ is nonempty. Luckily, it is easy to verify its validity in the regime of balanced cardinalities, similar to what we assumed in Corollary F.9. As a result, we establish the *tightness* (for $k \geq 2$, up to a constant factor) of our sample complexity upper bound in Corollary F.9, among the data sizes $N \gtrsim n^2 \ln n$ (Corollary F.17). Note that, as shown in the middle of the proof of Theorem F.16, $O(n^2 \ln n)$ is indeed the data size which is sufficient to guarantee that the learner observes all possible pairs in $\mathcal{X}_1 \times \mathcal{X}_2$ and $B \times \mathcal{X}_3$, ensuring that all possible input sequences in $\mathbb{X}$ are in-domain, with high probability.

---

**Corollary F.17** (Tightness of Sample Complexity Upper Bound for $k \geq 2$, Simple). *Under the two-hop task setup described in App. F.1, suppose that $|\mathcal{X}_1| = |\mathcal{X}_2| = |\mathcal{X}_3| = |B| = |V_b| = n \; (\forall b \in B)$. Also, assume that the ground-truth mapping $f$ satisfies Assumption F.6. Fix any $k \geq 2$ and any $\delta > 0$. Then, there exists $\widetilde{N}_\delta(n) > 0$ satisfying*

$$\widetilde{N}_\delta(n) \geq n^{2.5 - \frac{0.5}{k}} \cdot ((k-1)! \cdot \ln n)^{\frac{1}{2k}}$$

---

[7]We often say a complexity upper bound is tight if we can find a worst-case example whose complexity lower bound is almost identical to the upper bound.

> *such that, for any large enough $n$ (e.g., such that $kn > 256e(\ln n)^3$ and $\sqrt{n}(\ln n)^{\frac{1}{k}} > 4(\ln \frac{4}{\delta})^2$) and any integer $N$ satisfying that*
>
> $$n^2\left(2\ln n + \ln \frac{4}{\delta}\right) \leq N < \tilde{N}_\delta(n),$$
>
> *the learner with a uniformly randomly sampled training dataset $D$ (with replacements) **cannot** achieve a perfect coverage of in-domain data, i.e., $\overline{D} \not\subset \mathrm{Cover}_k(D)$, with probability at least $1 - \delta$.*

From now on, we prove Theorem F.16, the tightness result of the sample complexity upper bound obtained in Theorem F.7.

*Proof of Theorem F.16.* We aim to prove that, even when the dataset $D$ is large enough to ensure that $\overline{D} = \mathbb{X}$, the incomplete coverage may happen with high probability for a certain range of data size $N$. To this end, let us first define two sets $D_{12}$ and $D_{B3}$ as follows:

$$D_{12} := \{x_{12} \in \mathcal{X}_{12} : \exists x_3 \in \mathcal{X}_3 \text{ such that } (x_{12}, x_3) \in D\};$$
$$D_{B3} := \{(b, x_3) \in B \times \mathcal{X}_3 : \exists x_{12} \in V_b \text{ such that } (x_{12}, x_3) \in D\}.$$

Also, we denote by $\min \deg \mathcal{G} := \min_{v \in V} \deg_{\mathcal{G}}(v)$ the minimum degree among all vertices of a graph $\mathcal{G} = (V, E)$. With these definitions in place, we now apply the following lemma, which describes a necessary condition for perfect coverage.

**Lemma F.18 ($\downarrow$).** *Assume that $D_{12} = \mathcal{X}_{12}$ and $D_{B3} = B \times \mathcal{X}_3$ hold. Then, these imply that $\overline{D} = \mathbb{X}$. Furthermore, suppose that Assumption F.6 holds for a given $1 \leq k \leq |\mathcal{X}_3|$. Then, $\overline{D} \subset \mathrm{Cover}_k(D)$ implies that $\min \deg \mathcal{G}_b^{(D,k)} \geq 1$ for all $b \in B$.*

Refer to App. F.3 for its proof. Now, it suffices to show the inequality below:

$$\mathbb{P}_N\left(\overline{D} \subset \mathrm{Cover}_k(D)\right)$$
$$\leq \mathbb{P}_N\left(\overline{D} \subset \mathrm{Cover}_k(D) \text{ or } D_{12} \neq \mathcal{X}_{12} \text{ or } D_{B3} \neq B \times \mathcal{X}_3\right)$$
$$\leq \mathbb{P}_N\left(\min \deg \mathcal{G}_b^{(D,k)} \geq 1 \ (\forall b \in B) \text{ or } D_{12} \neq \mathcal{X}_{12} \text{ or } D_{B3} \neq B \times \mathcal{X}_3\right) \quad (\because \text{Lemma F.18})$$
$$\leq \mathbb{P}_N\left(\min \deg \mathcal{G}_b^{(D,k)} \geq 1 \ (\forall b \in B)\right) + \mathbb{P}_N\left(D_{12} \neq \mathcal{X}_{12}\right) + \mathbb{P}_N\left(D_{B3} \neq B \times \mathcal{X}_3\right) \leq \delta. \quad (9)$$

To this end, consider the multinomial random vector $(Z_{\boldsymbol{x}})_{\boldsymbol{x} \in \mathbb{X}} \sim \mathrm{Multinomial}(N; (p_{\boldsymbol{x}})_{\boldsymbol{x} \in \mathbb{X}})$ corresponding to a randomly sampled training dataset $D$, as defined in Proposition F.1. Define a property $\mathcal{M}_1$ as

$$(Z_{\boldsymbol{x}})_{\boldsymbol{x} \in \mathbb{X}} \in \mathcal{M}_1 \iff \min \deg \mathcal{G}_b^{(D,k)} \geq 1 \ (\forall b \in B). \quad (10)$$

Also, observe that

$$(T_{x_{12}})_{x_{12} \in \mathcal{X}_{12}} \sim \mathrm{Multinomial}(N; (q_{x_{12}})_{x_{12} \in \mathcal{X}_{12}}), \qquad \text{if} \begin{cases} T_{x_{12}} = \sum_{x_3 \in X_3} Z_{(x_{12}, x_3)}, \\ q_{x_{12}} = \sum_{x_3 \in X_3} p_{(x_{12}, x_3)}, \end{cases}$$

$$(U_w)_{w \in B \times \mathcal{X}_3} \sim \mathrm{Multinomial}(N; (r_w)_{w \in B \times \mathcal{X}_{12}}), \qquad \text{if} \begin{cases} U_{(b, x_3)} = \sum_{x_{12} \in V_b} Z_{(x_{12}, x_3)}, \\ r_{(b, x_3)} = \sum_{x_{12} \in V_b} p_{(x_{12}, x_3)}, \end{cases}$$

where we can actually put $p_{(x_{12}, x_3)} = \frac{1}{|\mathbb{X}|}$, $q_{x_{12}} = \frac{|\mathcal{X}_3|}{|\mathbb{X}|} = \frac{1}{|\mathcal{X}_{12}|}$, and $r_{(b, x_3)} = \frac{|V_b|}{|\mathbb{X}|}$ above. Using this notation, we know that

$$D_{12} = \mathcal{X}_{12} \iff T_{x_{12}} \geq 1 \ (\forall x_{12} \in \mathcal{X}_{12});$$
$$D_{B3} = B \times \mathcal{X}_3 \iff U_w \geq 1 \ (\forall w \in B \times \mathcal{X}_3).$$

Hence, to show Eq. (9), it suffices to prove the following three inequalities:

$$\mathbb{P}_{(Z_{\boldsymbol{x}})_{\boldsymbol{x}\in\mathbb{X}}\sim\text{Multinomial}(N;(p_{\boldsymbol{x}})_{\boldsymbol{x}\in\mathbb{X}})}\left((Z_{\boldsymbol{x}})_{\boldsymbol{x}\in\mathbb{X}}\in\mathcal{M}_1\right)\leq\frac{\delta}{2}; \qquad (11)$$

$$\mathbb{P}_{(T_{x_{12}})_{x_{12}\in\mathcal{X}_{12}}\sim\text{Multinomial}(N;(q_{x_{12}})_{x_{12}\in\mathcal{X}_{12}})}\left(\exists x_{12}\in\mathcal{X}_{12}\ \text{such that}\ T_{x_{12}}=0\right)\leq\frac{\delta}{4}; \qquad (12)$$

$$\mathbb{P}_{(U_w)_{w\in B\times\mathcal{X}_3}\sim\text{Multinomial}(N;(r_w)_{w\in B\times\mathcal{X}_{12}})}\left(\exists w\in B\times\mathcal{X}_3\ \text{such that}\ U_w=0\right)\leq\frac{\delta}{4}. \qquad (13)$$

A size $N$ of training dataset $D$ that is sufficient to ensure Eqs. (12) and (13) can be characterized with the following lemma.

**Lemma F.19** (A Tail Bound for Coupon Collector's Problem, ↓). *Consider a multinomial random vector $(Z_1, \cdots, Z_m) \sim \text{Multinomial}(n; p_1, \cdots, p_m)$. Then,*

$$\mathbb{P}(\exists i\ \text{such that}\ Z_i=0)\leq\sum_{i=1}^{m}(1-p_i)^n.$$

*In particular, if $\hat{p} = \min_{1\leq i\leq m} p_i$, then for any $\delta > 0$,*

$$n\geq\frac{1}{\hat{p}}\ln\frac{m}{\delta}\ \implies\ \mathbb{P}(\exists i\ \text{such that}\ Z_i=0)\leq\delta.$$

See App. F.3 for the proof. Applying the lemma and $\hat{n} = \min_{b\in B}|V_b|$, we can figure out that the data size

$$N\geq\max\left\{|\mathcal{X}_{12}|\ln\left(\frac{4|\mathcal{X}_{12}|}{\delta}\right),\ \frac{|\mathbb{X}|}{\hat{n}}\ln\left(\frac{4|B\times\mathcal{X}_3|}{\delta}\right)\right\} \qquad (14)$$

is enough to ensure both Eqs. (12) and (13). The remaining task now is to determine a condition for $N$ that guarantees Eq. (11).

Observe that $\mathcal{M}_1$ (defined as Eq. (10)) is a *non-decreasing* property (Def. F.11) because of a similar reason for the monotonicity of the property $\mathcal{C}$ (defined as Eq. (6); see the proof of Theorem F.7). Thanks to the monotonicity of $\mathcal{M}_1$, we can apply the Poissonization technique (Lemma F.12) once again. That is, taking any fixed $c \in (0, 2)$, we have

$$\mathbb{P}_N\left(\min\deg\mathcal{G}_b^{(D,k)}\geq 1\ (\forall b\in B)\right)$$

$$\leq\mathbb{P}_{\text{Po}(N+\varepsilon)}\left(\min\deg\mathcal{G}_b^{(D,k)}\geq 1\ (\forall b\in B)\right)+\exp\left(-\frac{(3-c)\varepsilon^2}{6N}\right)$$

$$\leq\mathbb{P}_{\text{Po}(N+\varepsilon)}\left(\min\deg\mathcal{G}_b^{(D,k)}\geq 1\right)+\exp\left(-\frac{(3-c)\varepsilon^2}{6N}\right)\qquad(\forall b\in B),$$

for any $\varepsilon\in(0,cN)$, where $\mathbb{P}_{\text{Po}(N+\varepsilon)}$ is the probability measure under i.i.d. Poisson random variables $Z_{\boldsymbol{x}}\sim\text{Poisson}\left(\frac{N+\varepsilon}{|\mathbb{X}|}\right)(\forall\boldsymbol{x}\in\mathbb{X})$. Let us take $\varepsilon=\sqrt{\frac{6N}{3-c}\ln\left(\frac{4}{\delta}\right)}$, which is smaller than $N$ by Eq. (14) (since $|\mathcal{X}_{12}|\geq 6>\frac{6}{3-c}$) and ensures that

$$\exp\left(-\frac{(3-c)\varepsilon^2}{6N}\right)=\exp\left(-\ln\left(\frac{4}{\delta}\right)\right)=\frac{\delta}{4}.$$

Moreover, we again use the fact that $\mathcal{G}_b^{(D,k)}$ is an instance of binomial random $k$-intersection graph $\mathcal{G}^{(k)}(n_b, m, p)$ with parameters $n_b = |V_b|$, $m = |\mathcal{X}_3|$, and $p = 1 - \exp(-(N+\varepsilon)/|\mathbb{X}|)$, under the Poissonization governed by $\mathbb{P}_{\text{Po}(N+\varepsilon)}$. Since we assume $k \geq 2$, we can use the "$k \geq 2$" part of Lemma F.14. In particular, we are to apply Lemma F.14 for an evidence graph $\mathcal{G}_b^{(D,k)}$ with $b \in \arg\min_{b'\in B}|V_b|$ (thus, $n_b = \hat{n}$), which is possible since we assume $m = |\mathcal{X}_3| = \Omega(\hat{n})$. We aim to find a sufficient condition for $N$ to have

$$\mathbb{P}_{\text{Po}(N+\varepsilon)}\left(\min\deg\mathcal{G}_b^{(D,k)}\geq 1\right)\leq\frac{\delta}{4}. \qquad (15)$$

Let us choose the sequence $\alpha_n = -(1 - \frac{1}{k}) \ln n$ in Lemma F.14, which guarantees that

$$\mathbb{P}(\min \deg \mathcal{G}^{(k)}(n, m, p) \geq 1) \to 0 \quad \text{as } n \to \infty.$$

From the definition of the limit of a sequence, for any choice of $\delta > 0$, let us consider $n_0(\delta) > 0$ satisfying that

$$\mathbb{P}\left(\min \deg \mathcal{G}^{(k)}(n, m, p) \geq 1\right) \leq \delta, \qquad n > n_0(\delta).$$

Then, choosing $\hat{n}$ greater than $n_0\left(\frac{\delta}{4}\right)$, we yield the bound (15). Observe that a choice of $p$ smaller than the threshold in Eq. (7) will never change the limit $\mathbb{P}(\min \deg \mathcal{G}^{(k)}(n, m, p) \geq 1) \to 0$. Thus, by our choice $\alpha_n = -(1 - \frac{1}{k}) \ln n$, it is enough to have

$$p = 1 - \exp\left(-\frac{N + \varepsilon}{|\mathbb{X}|}\right) \leq \left(\frac{(k-1)!\, (\ln \hat{n})}{\hat{n}}\right)^{\frac{1}{2k}} \cdot \frac{1}{\sqrt{|\mathcal{X}_3|}}.$$

Applying $1 - e^{-u} \leq u$ for $u \in \mathbb{R}$, it suffices to have

$$\frac{N + \varepsilon}{|\mathbb{X}|} \leq \left(\frac{(k-1)!\, (\ln \hat{n})}{\hat{n}}\right)^{\frac{1}{2k}} \cdot \frac{1}{\sqrt{|\mathcal{X}_3|}}.$$

Plugging $\varepsilon < cN$ and $|\mathbb{X}| = |\mathcal{X}_1 \times \mathcal{X}_2|\, |\mathcal{X}_3|$ in, we eventually have a sufficient condition

$$N \leq \frac{1}{1 + c} \cdot \left(\frac{(k-1)!\, (\ln \hat{n})}{\hat{n}}\right)^{\frac{1}{2k}} \cdot |\mathcal{X}_1 \times \mathcal{X}_2|\, \sqrt{|\mathcal{X}_3|}.$$

To summarize, for large enough $\hat{n} > n_0\left(\frac{\delta}{4}\right)$, provided that the data size $N$ satisfies

$$\max\left\{|\mathcal{X}_{12}| \ln \frac{4\,|\mathcal{X}_{12}|}{\delta}, \frac{|\mathbb{X}|}{\hat{n}} \ln \frac{4\,|B \times \mathcal{X}_3|}{\delta}\right\} \leq N \leq \frac{1}{1+c} \cdot \left(\frac{(k-1)!\, (\ln \hat{n})}{\hat{n}}\right)^{\frac{1}{2k}} \cdot |\mathcal{X}_1 \times \mathcal{X}_2|\, \sqrt{|\mathcal{X}_3|}$$

and we take $\varepsilon = \sqrt{\frac{6N}{3-c} \ln\left(\frac{4}{\delta}\right)}$, we finally have

$$\mathbb{P}_N\left(\overline{D} \subset \mathrm{Cover}_k(D)\right)$$

$$\leq \mathbb{P}_{\mathrm{Po}(N+\varepsilon)}\left(\min \deg \mathcal{G}_b^{(D,k)} \geq 1\right) + \exp\left(-\frac{(3-c)\varepsilon^2}{6N}\right) + \mathbb{P}_N\left(D_{12} \neq \mathcal{X}_{12}\right) + \mathbb{P}_N\left(D_{B3} \neq B \times \mathcal{X}_3\right)$$

$$\leq \frac{\delta}{4} + \frac{\delta}{4} + \frac{\delta}{4} + \frac{\delta}{4} = \delta.$$

Since the choice of $c \in (0, 2)$ is arbitrary, we obtain the same result ($\mathbb{P}_N(\cdots) \leq \delta$) by letting $c \searrow 0$ and choosing $N$ which satisfies that

$$\max\left\{|\mathcal{X}_{12}| \ln \frac{4\,|\mathcal{X}_{12}|}{\delta}, \; \frac{|\mathbb{X}|}{\hat{n}} \ln \frac{4\,|B \times \mathcal{X}_3|}{\delta}\right\} \leq N < \left(\frac{(k-1)!\, (\ln \hat{n})}{\hat{n}}\right)^{\frac{1}{2k}} \cdot |\mathcal{X}_1 \times \mathcal{X}_2|\, \sqrt{|\mathcal{X}_3|}.$$

∎

Before moving on to the postponed proofs of lemmas, we lastly remark that the same proof (of Theorem F.16) can hardly apply to the case of $k = 1$ in general. This is because, for the sake of simplicity in applying the necessary condition for a perfect coverage (Lemma F.18), we first characterize a minimal data size to guarantee $\overline{D} = \mathbb{X}$ with high probability as Eq. (14), using the tail bound of the coupon collector's problem (Lemma F.19). Unfortunately, it already exceeds the sample complexity upper bound to ensure $\overline{D} \subset \mathrm{Cover}_1(D)$ (obtained as Theorem F.8) for $k = 1$ with high probability, especially in the regime of balanced cardinalities assumed in Corollary F.17.

### F.3   BACKGROUNDS, USEFUL FACTS, AND LEMMAS

Now, we delve into the detailed backgrounds to provide a comprehensive understanding of the main theorems' proof, along with the deferred proofs of the lemmas used earlier.

We begin with the proof of a sufficient condition for a perfect coverage (Lemma F.10). For readability, we restate the lemma here.

**Lemma F.10** ($\downarrow$). *If all $\mathcal{G}_b^{(D,k)}$ are connected graphs ($\forall b \in B_D$), then we have $\overline{D} \subset \mathrm{Cover}_k(D)$.*

*Proof of Lemma F.10.* Take any $(x_{12}, x_3) \in \overline{D}$. By definition (Def. F.2), there exists $x'_{12} \in \mathcal{X}_{12}$ such that $f_1(x_{12}) = f_1(x'_{12})$ and $(x'_{12}, x_3) \in D$. Let $b = f_1(x'_{12}) \in B_D$. By assumption, there exists a path in $\mathcal{G}_b^{(D,k)}$ connecting $x_{12}$ and $x'_{12}$, which we write as $u_0(= x_{12}), u_1, \ldots, u_\ell(= x'_{12})$ for some integer $\ell \geq 1$. Then, for each $i \in [\ell]$, we have a set $S(u_{i-1}, u_i \mid D)$ (defined in Def. F.5) of size at least $k$. For each $\bar{x}_3 \in S(u_{i-1}, u_i \mid D)$, both $(u_{i-1}, \bar{x}_3)$ and $(u_i, \bar{x}_3)$ are both in $D$. Also,

$$f(u_{i-1}, \bar{x}_3) = f_2(b, \bar{x}_3) = f(u_i, \bar{x}_3),$$

meaning that the set $S_f(u_{i-1}, u_i \mid D)$ (defined in Def. F.3) is the same as $S(u_{i-1}, u_i \mid D)$, a set of size at least $k$, for each $i \in [\ell]$. Hence, $(x_{12}, x_3)$ satisfies the definition of $k$-coverage with a path $u_0, \ldots, u_\ell$, i.e., $(x_{12}, x_3) \in \mathrm{Cover}_k(D)$.   $\square$

Next, we show a necessary condition for a perfect coverage (Lemma F.18). Before proving the lemma, we recall that we have defined two sets:

$$D_{12} := \{x_{12} \in \mathcal{X}_{12} : \exists x_3 \in \mathcal{X}_3 \text{ such that } (x_{12}, x_3) \in D\};$$
$$D_{B3} := \{(b, x_3) \in B \times \mathcal{X}_3 : \exists x_{12} \in V_b \text{ such that } (x_{12}, x_3) \in D\}.$$

Also, for a graph $\mathcal{G} = (V, E)$, we define $\min \deg \mathcal{G} := \min_{v \in V} \deg_{\mathcal{G}}(v)$ as the minimum degree among all vertices of $\mathcal{G}$.

**Lemma F.18** ($\downarrow$). *Assume that $D_{12} = \mathcal{X}_{12}$ and $D_{B3} = B \times \mathcal{X}_3$ hold. Then, these imply that $\overline{D} = \mathbb{X}$. Furthermore, suppose that Assumption F.6 holds for a given $1 \leq k \leq |\mathcal{X}_3|$. Then, $\overline{D} \subset \mathrm{Cover}_k(D)$ implies that $\min \deg \mathcal{G}_b^{(D,k)} \geq 1$ for all $b \in B$.*

*Proof of Lemma F.18.* We first show that $D_{12} = \mathcal{X}_{12}$ and $D_{B3} = B \times \mathcal{X}_3$ imply that $\overline{D} = \mathbb{X}$. Since we already have $\overline{D} \subset \mathbb{X}$ by definition, it suffices to prove $\mathbb{X} \subset \overline{D}$. Let any $x_{12} \in \mathcal{X}_{12}$ and $x_3 \in \mathcal{X}_3$, and take $b = f_1(x_{12}) \in B$. Let us define:

$$W_{x_{12}} = \{x'_3 \in \mathcal{X}_3 : (x_{12}, x'_3) \in D\};$$
$$\widetilde{W}_{x_3}^{(b)} = \{x'_{12} \in V_b : (x'_{12}, x_3) \in D\}.$$

The conditions $D_{12} = \mathcal{X}_{12}$ and $D_{B3} = B \times \mathcal{X}_3$ imply that $W_{x_{12}} \neq \varnothing$ and $\widetilde{W}_{x_3}^{(b)} \neq \varnothing$, respectively. If we take any $x'_3 \in W_{x_{12}}$ and $x'_{12} \in \widetilde{W}_{x_3}^{(b)}$, they hold that $(x_{12}, x'_3) \in D$, $(x'_{12}, x_3) \in D$, and $f_1(x_{12}) = b = f_1(x'_{12})$. This set of conditions is equivalent to $(x_{12}, x_3) \in \overline{D}$; hence, we have just shown that

$$D_{12} = \mathcal{X}_{12} \text{ and } D_{B3} = B \times \mathcal{X}_3 \implies \overline{D} = \mathbb{X}.$$

Now, we prove the contrapositive of the lemma. Suppose that $\min \deg \mathcal{G}_b^{(D,k)} = 0$ for some $b \in B$; we aim to show that $\mathrm{Cover}_k(D) \neq \mathbb{X} = \overline{D}$, or, $\mathbb{X} \setminus \mathrm{Cover}_k(D) \neq \varnothing$.

Take a $b \in B$ such that $\min \deg \mathcal{G}_b^{(D,k)} = 0$, which implies that such a graph $\mathcal{G}_b^{(D,k)}$ has an isolated vertex $x_{12} \in V_b$ (i.e., $\deg_{\mathcal{G}_b^{(D,k)}}(x_{12}) = 0$). Let us fix such an $x_{12}$. Since we assume Assumption F.6, there must not be any edge connection between $\mathcal{G}_b^{(D,k)}$ and $\mathcal{G}_{b'}^{(D,k)}$ for distinct $b, b' \in B$. Thus, $x_{12}$ must be an isolated vertex in the whole substitution graph $\mathcal{G}_\bullet^{(D,k)}$. The isolation implies that, for any $x_3 \in \mathcal{X}_3$, the input sequence $(x_{12}, x_3)$ cannot be in the $k$-coverage of $D$ unless it is already in $D$.

Observe that $W_{x'_{12}} \neq \varnothing$ for any $x'_{12} \in \mathcal{X}_{12}$; otherwise, it inevitably holds that $(x'_{12}, \tilde{x}_3) \neq \overline{D}$ for every $\tilde{x}_3 \in \mathcal{X}_3$, which contradicts to the fact $\overline{D} = \mathbb{X}$ (under the assumption of the lemma). Thus, we can think of the following two cases:

- **Case 1** ($\varnothing \neq W_{x_{12}} \subsetneq \mathcal{X}_3$). Take any $x_3 \notin W_{x_{12}}$, which implies $(x_{12}, x_3) \notin D$. Thus, we have an element $(x_{12}, x_3) \in \mathbb{X} \setminus \mathrm{Cover}_k(D)$ because $x_{12}$ is an isolated vertex.

- **Case 2** ($W_{x_{12}} = \mathcal{X}_3$). Since $x_{12}$ is an isolated vertex, it is not adjacent to any other vertex $x'_{12}$. Fix any such an $x'_{12} \in \mathcal{X}_{12} \setminus \{x_{12}\}$. By definition of the edges in $E_\bullet^{(D,k)}$, we know that $\left| W_{x_{12}} \cap W_{x'_{12}} \right| < k$. Since $W_{x_{12}} = \mathcal{X}_3$, we also know that $\left| W_{x'_{12}} \right| < k \leq |\mathcal{X}_3|$. This implies that $x'_{12}$ cannot be adjacent to any other vertices in $\mathcal{X}_{12}$, meaning that $x'_{12}$ is an isolated vertex. Moreover, since $W_{x'_{12}} \subsetneq \mathcal{X}_3$ ($\because \left| W_{x'_{12}} \right| < |\mathcal{X}_3|$), we take any $x_3 \notin W_{x'_{12}}$. Then, since $(x'_{12}, x_3) \notin D$ and $x'_{12}$ is isolated, we have an element $(x'_{12}, x_3) \in \mathbb{X} \setminus \mathrm{Cover}_k(D)$.

In both cases above, we obtain the same result that $\mathbb{X} \setminus \mathrm{Cover}_k(D) \neq \varnothing$. It concludes the proof of the lemma. $\square$

We also prove the tail probability bound for the coupon collector's problem (Newman, 1960; Erdős & Rényi, 1961) here. We again restate the lemma here for the sake of readability.

**Lemma F.19** (A Tail Bound for Coupon Collector's Problem, ↓). *Consider a multinomial random vector* $(Z_1, \cdots, Z_m) \sim \mathrm{Multinomial}(n; p_1, \cdots, p_m)$. *Then,*

$$\mathbb{P}(\exists i \text{ such that } Z_i = 0) \leq \sum_{i=1}^{m} (1 - p_i)^n.$$

*In particular, if* $\hat{p} = \min_{1 \leq i \leq m} p_i$, *then for any* $\delta > 0$,

$$n \geq \frac{1}{\hat{p}} \ln \frac{m}{\delta} \implies \mathbb{P}(\exists i \text{ such that } Z_i = 0) \leq \delta.$$

*Proof of Lemma F.19.* Observe that $Z_i \sim \mathrm{Bin}(n, p_i)$ for each $i \in [m]$. It implies that $\mathbb{P}(Z_i = 0) = (1 - p_i)^n$. One can yield the same result by directly summing up the multinomial probability masses and applying the multinomial theorem: for instance,

$$\mathbb{P}(Z_1 = 0) = \sum_{\substack{z_2 + \cdots + z_m = n \\ z_2, \cdots, z_m \geq 0}} \frac{n!}{z_2! \cdots z_m!} \cdot p_2^{z_2} \cdots p_m^{z_m} = (p_2 + \cdots + p_m)^n = (1 - p_1)^n.$$

Thus, by applying a union bound,

$$\mathbb{P}(\exists i \text{ such that } Z_i = 0) \leq \sum_{i=1}^{m} \mathbb{P}(Z_i = 0) = \sum_{i=1}^{m} (1 - p_i)^n \leq m(1 - \hat{p})^n \leq m \, e^{-n\hat{p}},$$

where we apply $\hat{p} = \min_{1 \leq i \leq m} p_i$ and $1 + u \leq e^u$ ($\forall u \in \mathbb{R}$) in the last two inequalities above. Solving $m \, e^{-n\hat{p}} \leq \delta$, we conclude that $n \geq \frac{1}{\hat{p}} \ln \frac{m}{\delta}$ implies $\mathbb{P}(\exists i \text{ such that } Z_i = 0) \leq \delta$. $\square$

### F.3.1 (DE-)POISSONIZATION OF MONOTONE MULTINOMIAL EVENTS

Randomization is a probabilistic technique that provides a convenient way to analyze a sequence by treating the sequence index as a random variable/process. When the sequence $\{a_n\}_{n \geq 0}$ is *monotone* and *bounded*, in particular, it provides us upper/lower bounds of the difference between $a_n$ and the expectation $\mathbb{E}[a_N]$ in terms of a non-negative integral random variable $N$. The following lemma provides a simple conversion between a deterministic monotone bounded sequence and the expectation of the randomized sequence, which we will prove later.

**Lemma F.20** (De-Randomization Lemma, General, ↓). *Consider a non-decreasing real-valued sequence* $\{a_j\}_{j \geq 0}$ *which lies in a closed interval* $[m, M]$, *i.e.,* $m \leq a_0 \leq a_1 \leq a_2 \leq \cdots \leq M$. *Let* $N$ *be a non-negative integer-valued random variable with probability mass* $\mathbb{P}(N = j) = p_j$ *($j = 0, 1, 2, \cdots$). Then, for any* $n \geq 0$, *it holds that*

$$\mathbb{E}[a_N] - (M - m) \cdot \mathbb{P}(N > n) \leq a_n \leq \mathbb{E}[a_N] + (M - m) \cdot \mathbb{P}(N < n). \quad \text{(Non-Decreasing Case)}$$

*On the other hand, if* $\{a_j\}_{j \geq 0}$ *is non-increasing, i.e.,* $M \geq a_0 \geq a_1 \geq a_2 \geq \cdots \geq m$, *a similar result holds that*

$$\mathbb{E}[a_N] - (M - m) \cdot \mathbb{P}(N < n) \leq a_n \leq \mathbb{E}[a_N] + (M - m) \cdot \mathbb{P}(N > n). \quad \text{(Non-Increasing Case)}$$

*Here, all expectations are taken with respect to* $N$.

*Remark* F.21 (On the choice of random index $N$). When obtaining an estimate of a sequence value using the lemma above, it is unnecessary to utilize the same random variable $N$ for both upper and lower bounds. Namely, when the sequence is non-decreasing, a choice of $N$ with $\mathbb{E}[N] > n$ gives a small lower tail $\mathbb{P}(N < n)$ for the upper bound; whereas an $N$ with $0 < \mathbb{E}[N] < n$ gives a small upper tail $\mathbb{P}(N > n)$ for the lower bound. We can think of the opposite relationship for a non-increasing sequence as well. Moreover, to guarantee the tail bounds to be small, we often choose $N$ having a lot of mass concentrated around its mean (e.g., sub-Gaussian); then, we can take advantage of concentration inequalities associated with $N$ to have a better estimate of $a_n \approx \mathbb{E}[a_N]$.

Poissonization (Kac, 1949; Holst, 1986; Aldous, 1989; Jacquet & Szpankowski, 1998; Johansson, 1998; Borodin et al., 2000) can be thought of as a special case of randomization using Poisson random variables. A Poisson random variable $N \sim \text{Poisson}(\lambda)$ with a parameter $\lambda > 0$ is equipped with a probability mass function

$$\mathbb{P}(N = n) = \frac{e^{-\lambda}\lambda^n}{n!}. \qquad (n = 0, 1, 2, \cdots) \tag{Poisson}$$

In this section, our primary goal is to complete the proof of Lemma F.12, asserting that the Poissonization technique is particularly advantageous for analyzing a monotone property (Def. F.11) of multinomial random vectors (e.g., the connectivity of a random graph whose edge connections are sampled with replacements). To see why, let us first review an elementary property of Poisson random variables: the sum of independent Poisson random variables is again a Poisson random variable, whose parameter is the sum of the parameters of individual Poisson variables. We defer the proof for brevity.

**Lemma F.22** ($\downarrow$). *Consider mutually independent Poisson random variables $Z_i \sim \text{Poisson}(\lambda_i)$ ($i = 1, \cdots, m$). Then, $\sum_{i=1}^{m} Z_i \sim \text{Poisson}\left(\sum_{i=1}^{m} \lambda_i\right)$.*

Next, we explore the relationship between multinomial and Poisson distributions by establishing an equivalence between the probability associated with multinomial random vectors and the conditional probability of mutually independent Poisson random variables whose sum is fixed. As a result, we can construct a Poissonization of a sequence of probabilities by regarding the multinomial random vector's parameter $n$ as a Poisson random variable. The detailed formal statement is below, although we again defer its proof.

**Lemma F.23** ($\downarrow$). *Let $\mathcal{A}$ be any property of a (random) $m$-dimensional vector: we write $(z_1, \cdots, z_m) \in \mathcal{A}$ if $(z_1, \cdots, z_m)$ satisfies the property $\mathcal{A}$. Then, for any $\lambda > 0$, and $p_1, \cdots, p_m > 0$ that sums to 1 (i.e., $\sum_{i=1}^{m} p_i = 1$),*

$$\mathbb{P}_{\text{Multi}(n)}\Big((Z_1, \cdots, Z_m) \in \mathcal{A}\Big) = \mathbb{P}_{\text{Po}(\lambda)}\left((Z_1, \cdots, Z_m) \in \mathcal{A} \,\bigg|\, \sum_{i=1}^{m} Z_i = n\right).$$

*Here, $\mathbb{P}_{\text{Multi}(n)}$ is the probability measure under $(Z_1, \ldots, Z_m) \sim \text{Multinomial}(n; p_1, \ldots, p_m)$, whereas $\mathbb{P}_{\text{Po}(\lambda)}$ is the probability measure under $Z_i \sim \text{Poisson}(\lambda p_i)$ ($i = 1, \cdots, m$) which are mutually independent. As a result, it holds that*

$$\mathbb{E}_{N \sim \text{Poisson}(\lambda)}\Big[\mathbb{P}_{\text{Multi}(N)}\Big((Z_1, \cdots, Z_m) \in \mathcal{A}\Big)\Big] = \mathbb{P}_{\text{Po}(\lambda)}\Big((Z_1, \cdots, Z_m) \in \mathcal{A}\Big).$$

We now move our attention to a monotone property of (finite-dimensional) vectors, of which we recall the definition again.

**Definition F.11** (Monotone property). We refer to a property $\mathcal{A}$ of $m$-dimensional vectors as a **non-decreasing property** when, for any vector $(\Delta_1, \cdots, \Delta_m)$ with non-negative entries $\Delta_i \geq 0$ ($\forall i = 1, \cdots, m$),

$$(z_1, \cdots, z_m) \in \mathcal{A} \implies (z_1 + \Delta_1, \cdots, z_m + \Delta_m) \in \mathcal{A}. \qquad \text{(Non-Decreasing Property)}$$

On the other hand, we define **non-increasing property** $\mathcal{A}$ as what satisfies

$$(z_1, \cdots, z_m) \in \mathcal{A} \implies (z_1 - \Delta_1, \cdots, z_m - \Delta_m) \in \mathcal{A}. \qquad \text{(Non-Increasing Property)}$$

A monotone property of multinomial random vectors has an interesting feature: the probability of satisfying the property is also monotone in the parameter $n$ of the multinomial distribution (Lemma F.25). This is roughly because the multinomial distribution is inspired by multiple independent trials of with-replacement sampling: an additional trial corresponds to the increase of the parameter $n$ by 1. This intuitive explanation can be formalized into the following lemma.

**Lemma F.24** ($\downarrow$). *Consider a random vector* $(Z_1, \cdots, Z_m) \sim \text{Multinomial}(n; p_1, \cdots, p_m)$ *for* $n \geq 1$ *and* $p_1, \cdots, p_m > 0$ *such that* $\sum_{i=1}^{m} p_i = 1$. *That is, for any non-negative integers* $z_1, \cdots, z_m$ *whose sum is* $n$,

$$\mathbb{P}(Z_1 = z_1, \cdots, Z_m = z_m) = \frac{n!}{z_1! \cdots z_m!} \cdot p_1^{z_1} \cdots p_m^{z_m}. \qquad \text{(Multinomial)}$$

*Also, consider a categorical (so-called* multinoulli*) random variable* $j \sim \text{Categorical}(p_1, ..., p_m)$ *which can have a value among the integers* $\{1, 2, \cdots, m\}$, *i.e., for each* $i = 1, \cdots, m$,

$$\mathbb{P}(j = i) = p_i. \qquad \text{(Categorical)}$$

*Then, if we let* $\widetilde{Z}_i = Z_i + \mathbb{1}_{\{j=i\}}$ *for* $1 \leq i \leq m$, *we have a new multinomial random vector as follows:*

$$(\widetilde{Z}_1, \cdots, \widetilde{Z}_m) \sim \text{Multinomial}(n + 1; p_1, \cdots, p_m).$$

Using the above as a key lemma, we can prove the following lemma about the monotone property of a multinomial random vector.

**Lemma F.25** ($\downarrow$). *Recall the definition of* $\mathbb{P}_{\text{Multi}(n)}$ *from Lemma F.23. Fix any integers* $1 \leq n \leq n'$. *If a property* $\mathcal{A}$ *is non-decreasing, then*

$$\mathbb{P}_{\text{Multi}(n)} ((Z_1, \cdots, Z_m) \in \mathcal{A}) \leq \mathbb{P}_{\text{Multi}(n')} ((Z_1, \cdots, Z_m) \in \mathcal{A}).$$

*The direction of the inequality should be opposite ("*$\geq$*") if the property is non-increasing.*

Most importantly, thanks to Lemmas F.23 and F.25, we can apply the de-randomization lemma (Lemma F.20) for the sequence $\{a_n\}$ of probabilities defined with a monotone property $\mathcal{A}$ as

$$a_n := \mathbb{P}_{\text{Multi}(n)} ((Z_1, \cdots, Z_m) \in \mathcal{A}).$$

The only things left to obtain a complete (de-)Poissonization lemma for upper/lower-bounding the sequence $a_n$ are the tail probability bounds for a Poisson distribution. The concentration inequalities for the Poisson distribution are already well-known,[8] although we derive slightly different forms of them for our own purpose.

**Lemma F.26** ($\downarrow$). *If* $N \sim \text{Poisson}(n - \varepsilon)$ *for any* $n > 0$ *and* $0 < \varepsilon < n$, *it holds that*

$$\mathbb{P}(N \geq n) \leq \exp\left(-\frac{\varepsilon^2}{2n}\right). \qquad \text{(Upper Tail Bound)}$$

*On the other hand, if* $N \sim \text{Poisson}(n + \varepsilon)$ *for any* $n > 0$ *and* $0 < \varepsilon < cn$ *for some* $c \in (0, 3)$,

$$\mathbb{P}(N \leq n) \leq \exp\left(-\frac{(3-c)\varepsilon^2}{6n}\right). \qquad \text{(Lower Tail Bound)}$$

As a result, we finally summarize the arguments in this section as the following de-Poissonization lemma for a monotone property of multinomial random vectors:

**Lemma F.12** (De-Poissonization of Monotone Multinomial Events, $\downarrow$). *Fix any* $n \geq 1$. *Define*

$$P_n := \mathbb{P}_{(Z_1, \cdots, Z_m) \sim \text{Multinomial}(n; p_1, \cdots, p_m)} ((Z_1, \cdots, Z_m) \in \mathcal{A}).$$

*Let* $\mathcal{A}$ *be a non-decreasing property (Def. F.11) of* $m$-*dimensional vector. Then, we have an upper bound and a lower bound for* $P_n$ *as follows:*

$$P_n \leq \mathbb{P}_{\text{Po}(n+\varepsilon)} ((Z_1, \cdots, Z_m) \in \mathcal{A}) + \exp\left(-\frac{(3-c)\varepsilon^2}{6n}\right), \qquad (\forall c \in (0,3), \ \forall \varepsilon \in (0, cn))$$

$$P_n \geq \mathbb{P}_{\text{Po}(n-\varepsilon)} ((Z_1, \cdots, Z_m) \in \mathcal{A}) - \exp\left(-\frac{\varepsilon^2}{2n}\right). \qquad (\forall \varepsilon \in (0, n))$$

*If* $\mathcal{A}$ *is a non-increasing property (Def. F.11), then we have similar upper and lower bounds for* $P_n$:

$$P_n \leq \mathbb{P}_{\text{Po}(n-\varepsilon)} ((Z_1, \cdots, Z_m) \in \mathcal{A}) + \exp\left(-\frac{\varepsilon^2}{2n}\right), \qquad (\forall \varepsilon \in (0, n))$$

$$P_n \geq \mathbb{P}_{\text{Po}(n+\varepsilon)} ((Z_1, \cdots, Z_m) \in \mathcal{A}) - \exp\left(-\frac{(3-c)\varepsilon^2}{6n}\right). \qquad (\forall c \in (0,3), \ \forall \varepsilon \in (0, cn))$$

*Here, we denote by* $\mathbb{P}_{\text{Po}(\lambda)}$ *the probability measure under* $Z_i \overset{\text{indep.}}{\sim} \text{Poisson}(\lambda p_i)$ $(\forall i = 1, \cdots, m)$.

---

[8]For example, see a document online: "A short note on Poisson tail bounds."

*Proof of Lemma F.12.* This is the combination of Lemmas F.20, F.23, F.25 and F.26. □

We use this de-Poissonization lemma in the proof of our main theorem (Theorems F.7 and F.8).

Lastly, we conclude this section by presenting all the deferred proofs.

*Proof of Lemma F.20.* In the first part of the proof, we deal with the non-decreasing case. Observe that, since $a_N$ has a finite expectation ($m \leq \mathbb{E}[a_N] \leq M$), the infinite sum $\sum_{j \geq 0} p_j a_j$ converges. Now, let us first obtain the upper bound. Since $\sum_{j \geq 0} p_j = 1$,

$$
\begin{aligned}
a_n &= \sum_{0 \leq j < n} p_j a_n + \sum_{j \geq n} p_j a_n \\
&\leq \sum_{0 \leq j < n} p_j a_n + \sum_{j \geq n} p_j a_j && (\because a_j \geq a_n \text{ if } j \geq n) \\
&= \sum_{0 \leq j < n} p_j (a_n - a_j) + \sum_{j \geq 0} p_j a_j \\
&\leq (M - m) \cdot \sum_{0 \leq j < n} p_j + \sum_{j > 0} p_j a_j && (\because a_n - a_j \leq M - m) \\
&= (M - m) \cdot \mathbb{P}(N < n) + \mathbb{E}[a_N].
\end{aligned}
$$

Likewise, we obtain the lower bound.

$$
\begin{aligned}
a_n &= \sum_{0 \leq j \leq n} p_j a_n + \sum_{j > n} p_j a_n \\
&\geq \sum_{0 \leq j \leq n} p_j a_j + \sum_{j > n} p_j a_n && (\because a_j \leq a_n \text{ if } j \leq n) \\
&= \sum_{j \geq 0} p_j a_j - \sum_{j > n} p_j (a_j - a_n) \\
&\geq \sum_{j \geq 0} p_j a_j - (M - m) \cdot \sum_{j > n} p_j && (\because a_n - a_j \geq m - M) \\
&= \mathbb{E}[a_N] - (M - m) \cdot \mathbb{P}(N > n).
\end{aligned}
$$

These imply the inequality (Non-Decreasing Case) and prove the first part of the lemma.

The second part for the non-increasing sequence $\{a_n\}$ directly follows by applying the first part to the non-decreasing sequence $\{b_n\}$ defined as $b_n := M + m - a_n$. Indeed, we have

$$
\mathbb{E}[M + m - a_N] - (M - m) \cdot \mathbb{P}(N > n) \leq M + m - a_n \leq \mathbb{E}[M + m - a_N] + (M - m) \cdot \mathbb{P}(N < n),
$$

which implies the inequality (Non-Increasing Case) and concludes the proof of the lemma. □

*Proof of Lemma F.22.* Denote the sum of random variables by $\bar{Z}_m := \sum_{i=1}^{m} Z_i$. Likewise, let us write $\bar{\lambda}_m := \sum_{i=1}^{m} \lambda_i$. To prove the lemma, we proceed with the induction on $m \geq 1$.

Since the base case is obvious ($Z_1 \sim \text{Poisson}(\lambda_1)$), let us assume $m \geq 2$ and compute the probability mass. The inductive assumption says $\bar{Z}_{m-1} \sim \text{Poisson}(\bar{\lambda}_{m-1})$, which is independent of $Z_m$ due to the mutual independence of $Z_1, \cdots, Z_m$. Then, for any non-negative integer $\bar{z}$,

$$
\begin{aligned}
\mathbb{P}\left(\bar{Z}_m = \bar{z}\right) &= \sum_{z_m=0}^{\bar{z}} \mathbb{P}(\bar{Z}_{m-1} = \bar{z} - z_m, Z_m = z_m) \\
&= 1 \cdot \sum_{z_m=0}^{\bar{z}} \mathbb{P}(\bar{Z}_{m-1} = \bar{z} - z_m) \cdot \mathbb{P}(Z_m = z_m) && (\because \bar{Z}_{m-1} \perp Z_m) \\
&= \frac{\bar{z}!}{\bar{z}!} \cdot \sum_{z_m=0}^{\bar{z}} \frac{\exp\left(-\bar{\lambda}_{m-1}\right) \cdot (\bar{\lambda}_{m-1})^{\bar{z}-z_m}}{(\bar{z} - z_m)!} \cdot \frac{\exp(-\lambda_m) \cdot (\lambda_m)^{z_m}}{z_m!}
\end{aligned}
$$

$$= \frac{\exp\left(-\sum_{i=1}^{m}\lambda_i\right)}{\bar{z}!} \cdot \sum_{z_m=0}^{\bar{z}} \frac{\bar{z}!}{(\bar{z}-z_m)!\cdot z_m!} \left(\bar{\lambda}_{m-1}\right)^{\bar{z}-z_m}\lambda_m^{z_m}$$

$$= \frac{\exp\left(-\sum_{i=1}^{m}\lambda_i\right)}{\bar{z}!} \cdot \left(\bar{\lambda}_{m-1}+\lambda_m\right)^{\bar{z}} \qquad (\because \text{ binomial theorem})$$

$$= \frac{\exp\left(-\sum_{i=1}^{m}\lambda_i\right)}{\bar{z}!} \cdot \left(\bar{\lambda}_m\right)^{\bar{z}}.$$

Since the choice of $\bar{z}$ is arbitrary, by induction, this ends up proving that $\bar{Z}_m \sim \text{Poisson}\left(\bar{\lambda}_m\right)$.

Lastly, we remark that one may prove the same result without induction, by directly applying the multinomial theorem (which is essentially a recurrent application of the binomial theorem). $\qquad \square$

*Proof of Lemma F.23.* By the law of total probability, for any probability measure $\mathbb{P}$ under any distribution of a random vector $(Z_1, \cdots, Z_m)$,

$$\mathbb{P}((Z_1, \cdots, Z_m) \in \mathcal{A}) = \sum_{(z_1, \cdots, z_m) \in \mathcal{A}} \mathbb{P}(Z_1 = z_1, \cdots, Z_m = z_m).$$

Thus, it suffices to compare the multinomial distribution and the conditional distribution of a Poisson random vector given a fixed sum: namely, we aim to show here that

$$\mathbb{P}_{\text{Multi}(n)}(Z_1 = z_1, \cdots, Z_m = z_m) = \mathbb{P}_{\text{Po}(\lambda)}\left(Z_1 = z_1, \cdots, Z_m = z_m \mid \sum_{i=1}^{m} Z_i = n\right). \quad (16)$$

Moreover, it is sufficient for proving the first equation to study the case with non-negative integers $z_1, \cdots, z_m$ such that $\sum_{i=1}^{m} z_i = n$; otherwise, both sides of Eq. (16) are zero. In this case, observe an inclusion between events

$$\{Z_1 = z_1, \cdots, Z_m = z_m\} \subseteq \left\{\sum_{i=1}^{m} Z_i = n\right\}. \quad (17)$$

Because of this, we can compute a conditional probability mass as:

$$\mathbb{P}_{\text{Po}(\lambda)}\left(Z_1 = z_1, \cdots, Z_m = z_m \mid \sum_{i=1}^{m} Z_i = n\right)$$

$$= \frac{\mathbb{P}_{\text{Po}(\lambda)}(Z_1 = z_1, \cdots, Z_m = z_m)}{\mathbb{P}_{\text{Po}(\lambda)}\left(\sum_{i=1}^{m} Z_i = n\right)} \qquad (\because \text{ Eq. (17)})$$

$$= \left(\prod_{i=1}^{m} \frac{e^{-\lambda p_i}(\lambda p_i)^{z_i}}{z_i!}\right) \cdot \left(\frac{e^{-\lambda}\lambda^n}{n!}\right)^{-1} \qquad (\because \sum_{i=1}^{m} Z_i \sim \text{Poisson}(\lambda), \text{ due to Lemma F.22})$$

$$= \frac{n!}{z_1!\cdots z_m!} \cdot p_1^{z_1} \cdots p_m^{z_m} \qquad (\because \sum_{i=1}^{m} z_i = n)$$

$$= \mathbb{P}_{\text{Multi}(n)}(Z_1 = z_1, \cdots, Z_m = z_m).$$

Therefore, we have just proved Eq. (16).

Lastly, we conclude the proof by computing the expectation in terms of $N \sim \text{Poisson}(\lambda)$: since $\mathbb{P}(N = j) = \mathbb{P}_{\text{Po}(\lambda)}\left(\sum_{i=1}^{m} Z_i = j\right)$ ($\because$ Lemma F.22), by law of total probability,

$$\mathbb{E}_{N \sim \text{Poisson}(\lambda)}\left[\mathbb{P}_{\text{Multi}(N)}\left((Z_1, \cdots, Z_m) \in \mathcal{A}\right)\right]$$

$$= \sum_{j \geq 0} \mathbb{P}_{\text{Multi}(j)}\left((Z_1, \cdots, Z_m) \in \mathcal{A}\right) \cdot \mathbb{P}(N = j)$$

$$= \sum_{j \geq 0} \mathbb{P}_{\text{Po}(\lambda)}\left((Z_1, \cdots, Z_m) \in \mathcal{A} \mid \sum_{i=1}^{m} Z_i = j\right) \cdot \mathbb{P}(N = j)$$

$$= \sum_{j \geq 0} \mathbb{P}_{\text{Po}(\lambda)}\left((Z_1, \cdots, Z_m) \in \mathcal{A} \text{ and } \sum_{i=1}^{m} Z_i = j\right)$$

$$= \mathbb{P}_{\text{Po}(\lambda)}\left((Z_1, \cdots, Z_m) \in \mathcal{A}\right).$$

$\qquad \square$

*Proof of Lemma F.24.* Take any integers $z_1, \cdots, z_m$ such that $z_1 + \cdots + z_m = n+1$. Let us calculate the probability mass. Applying the law of total probability,

$$
\mathbb{P}\left(\widetilde{Z}_1 = z_1, \cdots, \widetilde{Z}_m = z_m\right)
$$

$$
= \sum_{i=1}^{m} \mathbb{P}\left(Z_1 + \mathbb{1}_{\{j=1\}} = z_1, \cdots, Z_m + \mathbb{1}_{\{j=m\}} = z_m \mid j = i\right) \cdot \mathbb{P}(j = i)
$$

$$
= \sum_{i=1}^{m} \mathbb{P}\left(Z_i = z_i - 1, Z_r = z_r \ (\forall r \neq i)\right) \cdot \mathbb{P}(j = i) \qquad (\because j \perp (Z_1, \cdots, Z_m))
$$

$$
= \sum_{i=1}^{m} \left(\frac{n! \cdot z_i}{z_1! \cdots z_m!} \cdot \frac{p_1^{z_1} \cdots p_m^{z_m}}{p_i}\right) \cdot p_i
$$

$$
= \frac{n!}{z_1! \cdots z_m!} \cdot p_1^{z_1} \cdots p_m^{z_m} \cdot \sum_{i=1}^{m} z_i
$$

$$
= \frac{(n+1)!}{z_1! \cdots z_m!} \cdot p_1^{z_1} \cdots p_m^{z_m}.
$$

Since the choice of $z_1, \cdots, z_m$ is arbitrary (given their fixed sum), it proves that $(\widetilde{Z}_1, \cdots, \widetilde{Z}_m)$ follows the distribution $\mathrm{Multinomial}(n + 1; p_1, \cdots, p_m)$, as desired. $\square$

*Proof of Lemma F.25.* We only consider the non-decreasing property $\mathcal{A}$ since the proof of the non-increasing case can be done symmetrically. Then, by induction, it suffices to prove

$$
\mathbb{P}_{\mathrm{Multi}(n)}\left((Z_1, \cdots, Z_m) \in \mathcal{A}\right) \leq \mathbb{P}_{\mathrm{Multi}(n+1)}\left((Z_1, \cdots, Z_m) \in \mathcal{A}\right). \tag{18}
$$

Fix any $n \geq 1$ and consider $(Z_1, \cdots, Z_m) \sim \mathrm{Multinomial}(n; p_1, \cdots, p_m)$. Then, by a property of multinomial random variables (Lemma F.24), we have another multinomial vector $(\widetilde{Z}_1, \cdots, \widetilde{Z}_m) \sim \mathrm{Multinomial}(n + 1; p_1, \cdots, p_m)$ by $\widetilde{Z}_i = Z_i + \mathbb{1}_{\{j=i\}}$, where $j \sim \mathrm{Categorical}(p_1, \cdots, p_m)$. Observe that

$$
\mathbb{P}\left((\widetilde{Z}_1, \cdots, \widetilde{Z}_m) \in \mathcal{A} \,\Big|\, (Z_1, \cdots, Z_m) \in \mathcal{A}\right) = 1,
$$

since $Z_i \leq \widetilde{Z}_i \ (\forall i)$ and $\mathcal{A}$ is a non-decreasing property.

Hence, we can derive the following by applying the law of total probability:

$$
\mathbb{P}\left((\widetilde{Z}_1, \cdots, \widetilde{Z}_m) \in \mathcal{A}\right)
$$

$$
= \mathbb{P}\left((\widetilde{Z}_1, \cdots, \widetilde{Z}_m) \in \mathcal{A} \,\Big|\, (Z_1, \cdots, Z_m) \notin \mathcal{A}\right) \cdot \mathbb{P}\left((Z_1, \cdots, Z_m) \notin \mathcal{A}\right)
$$

$$
\quad + \mathbb{P}\left((\widetilde{Z}_1, \cdots, \widetilde{Z}_m) \in \mathcal{A} \,\Big|\, (Z_1, \cdots, Z_m) \in \mathcal{A}\right) \cdot \mathbb{P}\left((Z_1, \cdots, Z_m) \in \mathcal{A}\right)
$$

$$
\geq 0 + 1 \cdot \mathbb{P}\left((Z_1, \cdots, Z_m) \in \mathcal{A}\right)
$$

$$
= \mathbb{P}_{\mathrm{Multi}(n)}\left((Z_1, \cdots, Z_m) \in \mathcal{A}\right).
$$

This proves Eq. (18), concluding the proof. $\square$

*Proof of Lemma F.26.* Let us recall the moment generating function (MGF) of a Poisson random variable $N \sim \mathrm{Poisson}(\lambda)$ is $\mathbb{E}[e^{tN}] = \exp\left(\lambda(e^t - 1)\right)$ ($\forall t \in \mathbb{R}$):

$$
\mathbb{E}[e^{tN}] = \sum_{j \geq 0} e^{tj} \cdot \frac{e^{-\lambda} \lambda^j}{j!} = e^{-\lambda} \cdot \sum_{j \geq 0} \frac{(\lambda e^t)^j}{j!} = \exp\left(-\lambda\right) \exp\left(\lambda e^t\right) = \exp\left(\lambda(e^t - 1)\right).
$$

For the upper tail (Chernoff) bound, we use $\lambda = n - \varepsilon$ and apply Markov inequality on $e^{tN}$: for any $n > \varepsilon > 0$ and $t > 0$,

$$
\mathbb{P}(N \geq n) = \mathbb{P}\left(e^{tN} \geq e^{tn}\right) \leq \mathbb{E}\left[e^{tN}\right] \cdot e^{-tn} \leq \exp\left((n - \varepsilon)(e^t - 1) - tn\right).
$$

Since $t$ is arbitrary, we take the infimum of both sides over $t > 0$, obtaining

$$\mathbb{P}(N > n) \le \inf_{t>0} \exp\left((n - \varepsilon)(e^t - 1) - tn\right)$$

$$= \exp\left(n\left(\ln\left(1 - \frac{\varepsilon}{n}\right) + \frac{\varepsilon}{n}\right)\right)$$

$$\le \exp\left(-\frac{\varepsilon^2}{2n}\right). \qquad\qquad \left(\because \ln(1 - u) + u \le -\frac{u^2}{2} \quad \text{if } u > 0\right)$$

This proves Eq. (Upper Tail Bound). On the other hand, for the lower tail (Chernoff) bound, we use $\lambda = n + \varepsilon$ and apply Markov inequality on $e^{-tN}$: for any $n > 0$, $\varepsilon > 0$, and $t > 0$,

$$\mathbb{P}(N \le n) = \mathbb{P}\left(e^{-tN} \ge e^{-tn}\right) \le \mathbb{E}\left[e^{-tN}\right] \cdot e^{tn} \le \exp\left((n + \varepsilon)(e^{-t} - 1) + tn\right).$$

Similarly as before, taking the infimum over $t > 0$,

$$\mathbb{P}(N < n) \le \inf_{t>0} \exp\left((n + \varepsilon)(e^{-t} - 1) + tn\right)$$

$$= \exp\left(n\left(\ln\left(1 + \frac{\varepsilon}{n}\right) - \frac{\varepsilon}{n}\right)\right)$$

$$\le \exp\left(-\frac{\varepsilon^2}{2n} + \frac{\varepsilon^3}{6n^2}\right). \qquad \left(\because \ln(1 + u) - u \le -\frac{u^2}{2} + \frac{u^3}{6} \quad \text{if } u > 0\right)$$

Observe that, if $0 < \varepsilon < cn$ for some $c \in (0, 3)$,

$$-\frac{\varepsilon^2}{2n} + \frac{\varepsilon^3}{6n^2} \le -\frac{\varepsilon^2}{2n} + \frac{\varepsilon^2}{6n} \cdot c = -\frac{(3 - c)\varepsilon^2}{6n}.$$

This proves Eq. (Lower Tail Bound). $\qquad\qquad\qquad\qquad\qquad\qquad\qquad\qquad\qquad\qquad\square$

### F.3.2 BINOMIAL RANDOM INTERSECTION GRAPHS

Let us explain the random $k$-intersection graph (Singer, 1995; Karoński et al., 1999; Godehardt & Jaworski, 2003). Consider a set $V$ of vertices and another set $W$ of items. Each vertex $v \in V$ is randomly assigned a subset of items $W_v \subset W$. Then, a pair of vertices $u$ and $v$ are adjacent (i.e., connected with an edge) in a random $k$-intersection graph if and only if at least $k$ items are shared between $u$ and $v$, i.e., $|W_u \cap W_v| \ge k$. We remark that $k$ is a universal constant throughout our paper.

One of the most well-studied models of random intersection graphs is *binomial random $k$-intersection graphs* $\mathcal{G}^{(k)}(n, m, p)$, with $n = |V|$ and $m = |W|$. In the $\mathcal{G}^{(k)}(n, m, p)$ model, for every vertex $v \in V$, each item $w \in W$ is assigned to $v$ independently with the same probability $p \in (0, 1)$. That is, the indicator variables $\mathbb{1}_{\{w \text{ is assigned to } v\}}$ are i.i.d. Bernoulli random variables with the same parameter $p$, for all vertices $v \in V$ and items $w \in W$. The word *binomial* is attached to the name because the number of items assigned to each vertex is a random variable following $\text{Bin}(m, p)$.

We are particularly interested in the $\mathcal{G}^{(k)}(n, m, p)$ model because of the following observation.

**Lemma F.13** (Evidence Graphs are Binomial $k$-Intersection Graphs under Poissonization, $\downarrow$)**.** *Let any $b \in B_D$. Consider a vector $(Z_{\boldsymbol{x}})_{\boldsymbol{x} \in \mathbb{X}}$ associated with a dataset $D$ (i.e., an input sequence $\boldsymbol{x}$ is sampled $Z_{\boldsymbol{x}}$ times in $D$). Let $\mathbb{P}_{\text{Po}(\lambda)}$ be a probability measure for $Z_{\boldsymbol{x}} \overset{i.i.d.}{\sim} \text{Poisson}(\lambda/|\mathbb{X}|)$. Then, under $\mathbb{P}_{\text{Po}(\lambda)}$, the b-evidence graph $\mathcal{G}_b^{(D,k)}$ (Def. F.5) is an instance of binomial random $k$-intersection graph $\mathcal{G}^{(k)}(n_b, m, p)$ with parameters*

$$n_b = |V_b|, \quad m = |\mathcal{X}_3|, \quad p = 1 - \exp\left(-\frac{\lambda}{|\mathbb{X}|}\right).$$

*Proof of Lemma F.13.* Recall that the vertex set of $\mathcal{G}_b^{(D,k)}$ is $V_b = \{x_{12} \in \mathcal{X}_{12} : f_1(x_{12}) = b\} \subset \mathcal{X}_{12}$. Also, consider $W := \mathcal{X}_3$ as the set of all 'items.' Define a set of items assigned to each vertex $x_{12} \in V_b$ as

$$W_{x_{12}} = \{x_3 \in W : (x_{12}, x_3) \in D\} = \left\{x_3 \in W : Z_{(x_{12}, x_3)} \ge 1\right\}.$$

Observe that all vertex-item assignments are i.i.d. Bernoulli random variables with the same probability parameter $p$: for any $x_{12} \in V_b$ and $x_3 \in \mathcal{X}_3$,

$$p = \mathbb{P}_{\mathrm{Po}(\lambda)}(Z_{(x_{12},x_3)} \geq 1) = 1 - \mathbb{P}_{\mathrm{Po}(\lambda)}(Z_{(x_{12},x_3)} = 0) = 1 - \exp\left(-\frac{\lambda}{|\mathbb{X}|}\right).$$

In addition, observe that the set $S(x_{12}, x'_{12} \mid D)$ defined in Eq. (5) is identical to the intersection $W_{x_{12}} \cap W_{x'_{12}}$. Hence, by the definition of the edge set $E_b^{(D,k)}$ in Eq. (4), two distinct vertices $x_{12}$ and $x'_{12}$ are adjacent if and only if $\left| W_{x_{12}} \cap W_{x'_{12}} \right| \geq k$. In summary, every $b$-evidence graph $\mathcal{G}_b^{(D,k)}$ is a binomial random $k$-intersection graph $\mathcal{G}^{(k)}(n_b, m, p)$ with parameters $n_b = |V_b|$, $m = |\mathcal{X}_3|$, and $p = 1 - \exp(-\lambda/|\mathbb{X}|)$. □

Note that $n$, $m$, and $p$ are not necessarily independent of each other; the parameters $m = m_n$ and $p = p_{n,m}$ are often regarded as functions of $n$. Indeed, it is often studied in the literature on random graphs that a sufficient condition for the parameters (in terms of $n \to \infty$) to guarantee certain graph properties, at least asymptotically.

Here, we review a few of the seminal results on the connectivity (as well as the disappearance of isolated vertices) of binomial random $k$-intersection graphs (Singer, 1995; Zhao et al., 2014; 2017; Rybarczyk, 2011; 2017).[9] The proofs are involved; thus, we omit them.

**Lemma F.14** (Zhao et al., 2014, Theorem 2; Zhao et al., 2017, Theorem 1 & Remark 1). *Fix any $k \geq 1$. Suppose that*

$$m = \begin{cases} \Omega\left(\min\left\{n(\ln n)^5, n^\rho\right\}\right), & \text{if } k = 1, \text{ for any } \rho > 1; \\ \Omega(n), & \text{if } k \geq 2, \end{cases}$$

*and*

$$p = \left(\frac{k! (\ln n + \alpha_n)}{n}\right)^{\frac{1}{2k}} \cdot \frac{1}{\sqrt{m}} \tag{7}$$

*for any sequence $\{\alpha_n\}$ which attains a limit $\alpha_\infty \in [-\infty, +\infty]$ as $n \to \infty$. Then,*

$$\lim_{n\to\infty} \mathbb{P}\left(\mathcal{G}^{(k)}(n, m, p) \text{ is connected}\right) = \lim_{n\to\infty} \mathbb{P}\left(\min \deg \mathcal{G}^{(k)}(n, m, p) \geq 1\right) = \exp\left(-e^{-\alpha_\infty}\right),$$

*where we compute $\exp\left(-e^{-(-\infty)}\right) = 0$ and $\exp\left(-e^{-(+\infty)}\right) = 1$.*

**Lemma F.15** (Singer, 1995, Propositions 3.1–2, Theorem 3.3). *Let $k = 1$. Suppose that $m = n^\rho$ for $\rho > 0$ and*

$$p = \begin{cases} \dfrac{\ln n + \alpha_n}{m} & \text{for } \rho \leq 1; \\ \sqrt{\dfrac{\ln n + \alpha_n}{mn}} & \text{for } \rho > 1, \end{cases}$$

*for any sequence $\{\alpha_n\}$ which attains a limit $\alpha_\infty \in \{-\infty, +\infty\}$ as $n \to \infty$. Then,*

$$\lim_{n\to\infty} \mathbb{P}(\mathcal{G}^{(1)}(n, m, p) \text{ is connected}) = \lim_{n\to\infty} \mathbb{P}\left(\min \deg \mathcal{G}^{(1)}(n, m, p) \geq 1\right) = \begin{cases} 0, & \text{if } \alpha_\infty = -\infty; \\ 1, & \text{if } \alpha_\infty = +\infty. \end{cases}$$

---

[9]Caveat: in the literature of (random) graph theory, the letter '$k$' is often used for $k$-connectivity (i.e., being connected after removing less than $k$ vertices/edges). On the other hand, they study 'random $s$-intersection graphs,' using $s$—instead of $k$ as this paper—as the intersection constraint parameter.

# G  ADDITIONAL RESULTS FOR POWER-LAW SCALING ANALYSIS

## G.1  MEASUREMENT PROTOCOL FOR $N_{\mathrm{req}}$

To empirically determine the minimum dataset size required for reliable compositional generalization ($N_{\mathrm{req}}$), we develop a measurement protocol that accounts for practical computational constraints while ensuring robustness. For each token set size $|\mathcal{X}|$ and task structure, we test multiple dataset sizes until we identify the threshold point where the model successfully generalizes to the ID test set.

Specifically, our criterion for the "reliable generalization" on ID is defined as: *The model must reach ID test accuracy of 0.99 within 100 epochs after achieving training accuracy > 0.99.* This protocol balances several considerations:

1. **Training-to-generalization delay**: Larger datasets naturally require more iterations to fit training data. By measuring epochs after reaching training accuracy > 0.99, we focus on the generalization gap rather than conflating it with initial training difficulty.

2. **Epoch-based measurement**: Using epochs rather than raw training steps ensures that the model sees each functional equivalence evidence approximately the same number of times, regardless of dataset size. This provides a fairer comparison across different dataset sizes.

3. **Practical time constraints**: While indefinite training might eventually yield generalization with smaller datasets, we established a reasonable upper bound (100 epochs post-training convergence) to reflect practical limitations.

4. **Measurement precision**: For each identified $N_{\mathrm{req}}$, we verified that 75% of this dataset size consistently failed to meet our generalization criterion. This establishes that our measurement error is at most $-\log(0.75) \approx 0.125$ in log scale, providing confidence in the derived power-law exponents.

## G.2  MEASURED POWER-LAW SCALING EXPONENTS ACROSS TASK STRUCTURES AND MODEL SIZES

Using our measurement protocol, we measure the required dataset size $N_{\mathrm{req}}$ across three different compositional structures (2-HOP, PARALLEL-2-HOP, and 3-HOP) and three model scales (68M, 96M, and 1.5B parameters). For each task structure, we vary the token set size $|\mathcal{X}|$ from 50 to 200, allowing us to observe the scaling relationship.

Tab. 2 presents the power-law exponents obtained by linear fitting $\log(|\mathcal{X}|)$ vs. $\log(N_{\mathrm{req}})$ plots, all with $R^2 > 0.99$. The consistency of exponents across model sizes suggests that the observed power-law scaling relates to properties of the compositional tasks themselves, rather than model capacity. This observation aligns with our theoretical derivation in Section 5.1, which predicts that the required dataset size scales at least quadratically with token set size.

Table 2: The power-law exponents for different tasks and GPT-2 sizes, obtained by linear fitting $\log(|\mathcal{X}|)$ vs. $\log(N_{\mathrm{req}})$ plots. $R^2 > 0.99$ for all linear fitting.

| Model Size | 2-HOP | PARALLEL-2-HOP | 3-HOP |
|---|---|---|---|
| 68M | 2.13 | 2.47 | 2.61 |
| 96M | 2.26 | 2.35 | 2.50 |
| 1.5B | 2.28 | 2.17 | 2.60 |

## G.3  ROBUSTNESS TO HYPERPARAMETER VARIATIONS

To verify that the power-law scaling relationship we observed is not an artifact of specific hyperparameter choices, we conduct ablation studies with modified training configurations. Fig. 16 demonstrates that for the 2-HOP task with $|\mathcal{X}| = 50$, the following changes did not significantly affect the measured $N_{\mathrm{req}}$ or the derived power-law exponent:

1. **Learning rate reduction**: Halving the learning rate from 8e-4 to 4e-4;

2. **Weight decay reduction**: Decreasing weight decay by a factor of 10 (from 0.1 to 0.01);

3. **Generalization criteria modification**: Requiring test accuracy > 0.95 within 10 epochs after training accuracy > 0.95.

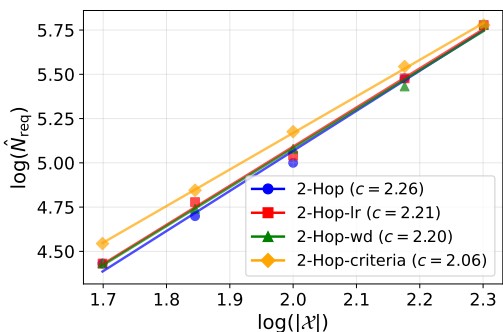

Figure 16: Robustness of power-law scaling relationship to hyperparameter variations in the 2-HOP task with $|\mathcal{X}| = 50$. Each line shows the training and test accuracy curves for a different configuration: (1) baseline, (2) reduced learning rate (4e-4, half of baseline), (3) reduced weight decay (0.01, one-tenth of baseline), and (4) changed generalization criteria (test accuracy > 0.95 within 10 epochs after training accuracy > 0.95). $R^2 > 0.99$ for all linear fitting.

This robustness to hyperparameter variations suggests that the power-law relationship between token set size and required dataset size is primarily a property of the compositional generalization process, rather than an artifact of specific optimization settings.

# H   DETAILED ANALYSIS FOR NON-TREE TASK

This section provides additional analyses that support our findings in Sec. 7 regarding the challenges of path ambiguity in the NON-TREE task.

## H.1   COVERAGE ANALYSIS

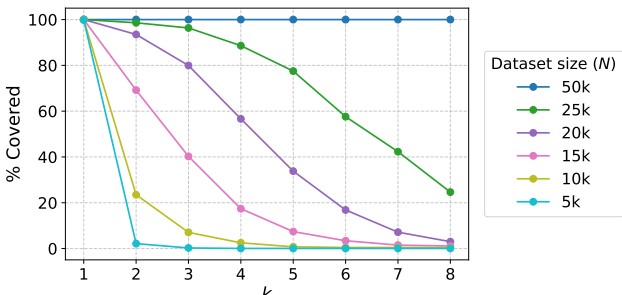

Figure 17: Coverage analysis for NON-TREE task with $|\mathcal{X}| = 50$. The graph shows the percentage of ID test data covered at different $k$ values across various dataset sizes ($N$). Compared to the 2-HOP task (Fig. 3, left), NON-TREE has significantly lower coverage at equivalent dataset sizes, indicating that path ambiguity impedes the formation of functional equivalence relationships.

Fig. 17 demonstrates that with equivalent training dataset sizes, a smaller percentage of ID test examples fall inside $k$-coverage for the NON-TREE task compared to the 2-HOP task shown in Fig. 3 (Left). This aligns with our theoretical analysis in Sec. 7, which predicts that path ambiguity limits the establishment of functional equivalence relationships between input subsequences, as the model cannot generalize across different $x_2$ values in the NON-TREE structure even when they produce the same intermediate state $b = f_1(x_1, x_2)$.

## H.2   EFFECT OF MODEL SCALING

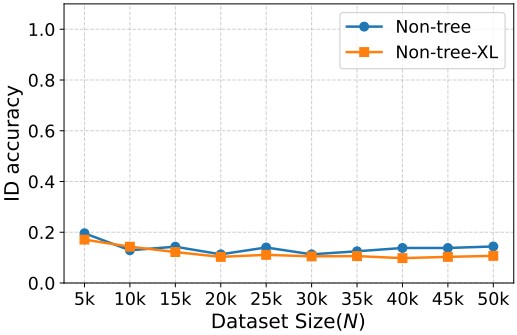

Figure 18: ID test accuracy comparison between GPT-2 (96M parameters) and GPT-2-XL (1.5B parameters) on the NON-TREE task with $|\mathcal{X}| = 50$, measured 100 epochs after training accuracy exceeds 0.99. Despite the 15x increase in parameter count, the accuracy does not increase.

Fig. 18 shows that scaling up the model size to GPT-2-XL (1.5B parameters) does not significantly improve generalization performance on the NON-TREE task, even when measured 100 epochs after reaching training accuracy > 0.99. This suggests that the challenges posed by path ambiguity cannot be overcome simply by increasing model capacity, supporting our claim that the limitation is structural rather than related to model capacity.

## H.3   COMPARISON BETWEEN MAMBA AND GPT-2 ON NON-TREE TASK

Fig. 19 shows that the Mamba model (4 layers, hidden dimension of 256, trained with learning rate of 0.008) shows a similar trend of ID test accuracy on NON-TREE task compared to GPT-2, suggesting that the generalization failure is more likely due to the task structure itself, rather than a specific model architecture.

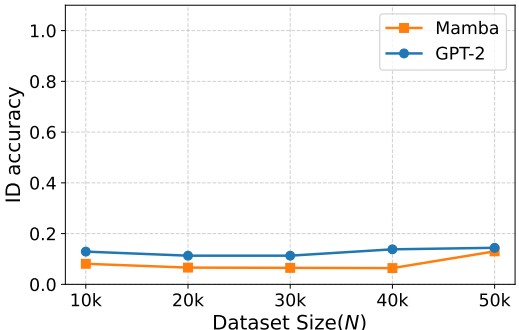

Figure 19: ID test accuracy comparison between GPT-2 and Mamba on the NON-TREE task with $|\mathcal{X}| = 50$, measured 100 epochs after training accuracy exceeds 0.99.

## H.4 REPRESENTATION ANALYSIS IN SUCCESSFUL GENERALIZATION

For a model that eventually achieved near-perfect ID accuracy (0.96) after extended training (36k epochs, $|\mathcal{X}| = 50$, $N = 50$k), we conduct causal tracing analysis to understand how it achieves generalization despite path ambiguity.

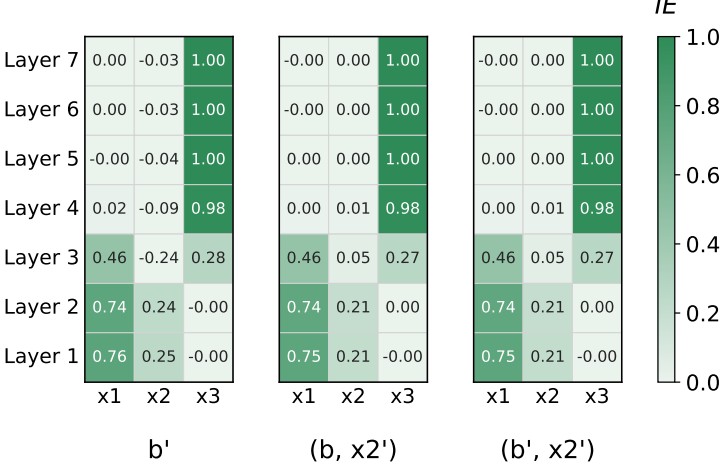

Figure 20: Causal tracing analysis for the NON-TREE model after extended training. The heatmap shows indirect effect values across different layer-token positions. **Left**: perturbation leading to different intermediate state $b = f_1(x_1, x_2)$. **Middle**: same $b$ but different $x_2$. **Right**: different $b$ and $x_2$.

The causal tracing results in Fig. 20 reveal how the model achieves generalization in the presence of path ambiguity. Across all perturbation strategies, the model's predictions show strong causal dependence on representations at both the $x_1$ and $x_2$ positions, indicating reliance on direct access to both input tokens rather than an abstracted intermediate computation. This pattern contrasts sharply with the 2-HOP task, where causal effects concentrate primarily at positions corresponding to clustered functional equivalence representations.

This analysis demonstrates that even models achieving high accuracy on NON-TREE tasks do so by developing context-dependent representations rather than unified abstractions of intermediate states. The model forms separate computational pathways conditioned on the $x_2$ value, rather than learning a single unified representation of the intermediate state $b = f_1(x_1, x_2)$. This represents a fundamentally different solution strategy compared to the 2-HOP task, with implications for both generalization capability and interpretability.

# I    Detailed Discussion on the Taxonomy for Understanding Generalization Mechanisms

In this section, we initiate a discussion to disambiguate the mixed mechanisms of generalization into isolated testable parts by sketching a preliminary taxonomy that distinguishes three complementary mechanisms of generalization. We note that we do not view our categorization as a complete one.

**Type-I: Functional equivalence-based generalization (pattern matching).**  This is precisely what we formalized through this work: models learn that different input fragments yield identical results in shared contexts, enabling generalization to new fragment combinations. Crucially, this generalization remains bounded by coverage, and reliable generalization fails without sufficient functional equivalence evidence. In other words, it describes the ceiling of pattern matching.

**Type-II: Function property-based generalization.** This mechanism exploits intrinsic properties of individual primitive functions, e.g., algebraic invariances such as commutativity or *input irrelevance*, where certain arguments never affect the output (e.g., $f(x_1, x_2) = f(x_1)$ even when distractor $x_2$ is present (Wen et al., 2025)). Unlike the previous type, this mechanism explains the generalization beyond the coverage by leveraging 'global' properties that hold across all possible inputs of a primitive, beyond what is actually observed. We interpret the Reversal Curse phenomenon (Berglund et al., 2024) as an example of the layered nature of challenges across multiple generalization types. Our framework predicts the failure of pattern matching on this problem, since training on "$A$ is $B$" provides no functional equivalence evidence for "$B$ is $A$". An architectural modification to learn inverse mappings from the same training data to handle this problem (Lv et al., 2024) can be interpreted as a utilization of Type-II generalization to enable generalization beyond coverage.

**Type-III: Shared-operator generalization.** This mechanism emerges through the reuse of identical primitive functions across computational positions (e.g., when $f_1 = f_2$). Recurrent architectures (Hochreiter & Schmidhuber, 1997) exemplify the utilization of this through weight sharing across time steps, enabling processing of variable-length sequences (Graves et al., 2014). Likewise, it has been reported in Transformers with inductive biases towards reuse of the same computation through parameter sharing (Dehghani et al., 2019; Csordás et al., 2021; Wang et al., 2024a) can improve generalization on complex compositional tasks where the same primitive function can be reused in various contexts. With a similar insight, the generalization capability of Transformers in terms of sequence length (i.e., *length generalization*) and its limitation has recently been studied a lot, especially in terms of arithmetic/algorithmic tasks (Anil et al., 2022; Zhou et al., 2024a; Shen et al., 2023; Zhou et al., 2024b; Cho et al., 2024; 2025). We interpret this mechanism as exploiting structural repetition.

**Distinguishing mechanisms from phenomena.**  Compared to prior categorizations of generalization, which focus on observed phenomena (Lake & Baroni, 2018; Hupkes et al., 2020), we categorize the underlying mechanisms. As noted in Sec. 1, many behavioral studies have examined tasks mixing functional equivalence, primitives' intrinsic properties, and operator reuse within the same benchmark, making it difficult to pinpoint the true source of success or failure. We therefore advocate clearer experimental control and community discussion around this mechanistic distinction to sharpen future analyses of neural generalization.

**Implications and future directions.**  Real compositional tasks typically involve combinations of all three types (and possibly more). While preliminary, we believe this taxonomy guides future research design on constructive characterization of neural networks' generalization behaviors on discrete sequence tasks. In this broader context, this work can be understood as a characterization and formalization of pattern-matching generalization to clarify its specific boundaries. When models succeed beyond our coverage predictions, we view these as exploiting other generalization mechanisms, i.e., beyond pattern matching. Our focused study suggests that challenges to reliable generalization remain as long as models rely primarily on pattern matching, requiring methodological innovations that harness non-pattern-matching mechanisms, e.g., variable binding. We hope this preliminary taxonomy serves as a research program towards our better understanding of generalization, and confirming or refuting its utility is an empirical matter that we invite the community to explore.

