# OpenReview forum: "Characterizing Pattern Matching and Its Limits on Compositional Task Structures"
_ICLR.cc/2026/Conference — ICLR 2026 Poster_

### Official Review · Reviewer_ZpTF · 2025-10-25

**Soundness:** 4
**Presentation:** 3
**Contribution:** 4
**Rating:** 8
**Confidence:** 3

**Summary:**

This work presents a formalization of pattern matching by quantifying evidence for functional equivalence. The authors systematically study how strength of functional equivalence in the data leads to different kinds of success and failures in compositional generalization tasks. The formalization is accompanied by empirical evidence with both Transformers and mamba architectures on data scaling, learned internal representation of functional equivalence, and the role of CoT on a set of synthetic compositional generalization tasks. The results yield important insights on the limitations of compositional generalization supported by pattern-matching behavior, including generalizing to in-frequent data and multi-hop problems.

**Strengths:**

- This work makes a significant contribution in providing a formalization to the boundary of pattern matching and compositional generalization.
- The paper is dense but I find it a good read. I appreciate that the authors studied a range of important issues beyond the base setting within the task suite, including testing two architectures, data/model scaling, interpretability, tasks with different compositional structures, and CoT.
- The results make several interesting implications for the capabilities of larger models.

**Weaknesses:**

- I think it would be good if the implications for specific LLM failures are discussed in more detail, as well as more discussions of whether certain aspects of the formalization would/would not apply given how natural language data may differ in properties studied in the synthetic settings. Though I understand this could be in part due to the space limit.

**Questions:**

- For clarification, how is k computed if there is functional equivalence between more than two subsequences?
- I'm curious what might happen if the models are trained on a mixture of different compositional tasks, or a mixture of compositional/non-compositional tasks? IMO this potentially captures the nuanced structures in natural languages, in which certain aspects are strongly compositional and others less so. Without explicit task cues, how would this change the learned strategy? Would functional equivalence be learned earlier or later? Would models still develop some form of context/task-dependent functional equivalence?

---

> ### Author Response · Authors · 2025-11-21
>
> We thank Reviewer ZpTF for their thoughtful and insightful questions/comments on our work. We appreciate the reviewer’s recognition of the significance of our work’s contribution and rigor in systematic analyses. We would like to respond to each weakness and question in order.
>
> ## (W1) Implications for specific LLM failures & How natural language data may differ
>
> Yes, these are very important points, but we could not fully discuss this in the main text due to the space limit, and we thank the reviewer for recognizing this!
>
> For the first point, our work has several important implications for specific LLM failures. To list a few of them:
>
> - It helps to quantifiably explain the data-hungry nature of LLMs on compositional tasks. [1] observed that data demand for generalization on k-hop natural-language data dramatically increases with increasing the number of hops. Such a data-hungry nature of compositional tasks is also well observed in semantic parsing, the task in which the model should convert natural language instructions into a structured SQL query. [2] demonstrated that the diversity of data, in that the same component is shown in diverse contexts, is more important than the sheer size of the dataset for good generalization.
> We note that this insight is not completely new (e.g., [3]), but our work steps further to provide a rigorous quantitative analysis of the data scaling law required for generalization on compositional task structures through pattern matching.
> - This also helps explain why strategic data augmentation methods [4] of substituting components in natural language data (and hence can contribute to a robust functional equivalence) greatly boost a model’s performance on a given natural-language compositional generalization task.
> - Our framework aligns with the reversal curse phenomenon (e.g., failure to automatically infer ‘A is child of B’ by observing ‘B is parent of A’) observed in recent LLMs [5], as pattern-matching models fundamentally cannot generalize to reversed relationships without explicit functional equivalence evidence in training data.
> - Similarly, our framework well explains the notoriously hard problem of logical negation in LLMs [6], as a purely pattern-matching learner cannot deduce that the negation rule (if a statement ‘p’ is true, then ‘not p’ is false and vice versa) should apply to *any* well-formed statement (including the ones that aren’t observed) through finite observations.
>
> For the second point, yes, several distinguishing properties of natural language are not captured in our synthetic settings, which can supply additional generalization mechanisms:
>
> - **Natural language data contain inherent semantic properties.** Unlike the random mapping we constructed, there are systematic rules, including the reversal relationship and logical negation we discussed above. If we have an explicit inductive bias to harness such algebraic properties of mappings, we may improve generalization, as in recent work that mitigates the reversal curse via symmetry-aware training schemes [7].
> - **Same computations can be recurrently used in different places in natural language.**
> When learning natural language tasks, a model may exploit the reuse of identical primitive functions across computational positions (e.g., $f_1 = f_2$ in our two-hop task). Recurrent architectures exemplify the utilization of this property through weight sharing across time steps, enabling processing of variable-length sequences [8]. Similarly, it has been observed that Transformers with inductive biases towards reuse of the same computation through parameter sharing [9] can improve generalization on complex compositional tasks where the same primitive function can be reused in various contexts [10]. We interpret this mechanism as exploiting structural repetition.
> - **Natural language has a dynamic structure.** This is covered in our response to Q2.
>
> We finally note that the first and second points are identified and discussed further in App.H as ‘function property-based generalization’ and ‘shared-operator generalization’ respectively. We will add this discussion to the main text in the final version, provided with additional space.

---

> ### Author Response · Authors · 2025-11-21
>
> ## (Q1) How is k computed if there is functional equivalence between more than two subsequences?
>
> In our framework, **k is always defined pairwise for a fixed index set** $I$. For two subsequences $x_I$ and $x^\prime_I$, $k$ is the number of *distinct* shared contexts $x_{I^c}$ in the training set where both $(x_I, x_{I^c})$ and $(x^\prime_I, x^\prime_{I^c})$ appear and yield the same output. We say they are functionally $k$-equivalent if this count is at least $k$.
> If one pair of examples provides evidence at multiple index sets (say $I$ and $J$), that example contributes separately to $k_I$ and $k_J$. In short, we never pool evidence across different index sets.
> The description of how $k$ is algorithmically computed in our analysis can be found in App.C.
>
>
> ## (Q2) What might happen if the models are trained on a mixture of different tasks?
>
> This is a great question, and we believe this is indeed one of the most important next steps to investigate.
> We do expect there will be synergy or interference between different task structures. Although not included in the paper, for the experiment used in App.I, we trained a model with a mixture of one-hop triplets with two-hop instances. In that experiment, we observed a synergistic behavior: the generalization speed is boosted, and the number of two-hop instances required for complete ID generalization ($N_{\rm req}$) is slightly lower, compared to when we train the model only with two-hop data. Such a synergistic effect is also observed in [1], which demonstrated that training with mixed data or a curriculum learning approach of showing data of gradually increasing hops can improve data efficiency. Although we did not observe such interference explicitly, it might arise if we simultaneously train a model with two unrelated compositional tasks (e.g., multitask learning), which remains as an exciting future direction.
>
> ---
>
> **As a final remark, we have significantly strengthened the theoretical contribution in Sec.6 by replacing the heuristic derivation of Result 6.1 with a rigorous formal proof (now Theorem 6.1), which yields a sample-complexity upper bound for perfect ID generalization.** We kindly refer the reviewer to our global comment (“Major Theoretical Upgrade”) for a summary of this result and the corresponding changes in the updated revision.
>
> We sincerely thank the reviewer again, and we hope our response addresses all points raised in the review.
>
> ---
>
> ## References
>
> [1] Yao, Y., Du, Y., Zhu, D., Hahn, M., & Koller, A. (2025). Language models can learn implicit multi-hop reasoning, but only if they have lots of training data. In C. Christodoulopoulos, T. Chakraborty, C. Rose, & V. Peng (Eds), *Proceedings of the 2025 Conference on Empirical Methods in Natural Language Processing* (pp. 9695–9713).
>
> [2] Keysers, D., Schärli, N., Scales, N., Buisman, H., Furrer, D., Kashubin, S., … Bousquet, O. (2020). Measuring Compositional Generalization: A Comprehensive Method on Realistic Data. *International Conference on Learning Representations*.
>
> [3] Lake, B., & Baroni, M. (2018). Generalization without systematicity: On the compositional skills of sequence-to-sequence recurrent networks. *In International conference on machine learning* (pp. 2873-2882). PMLR.
>
> [4] Andreas, J. (2020). Good-enough compositional data augmentation. *In Proceedings of the 58th annual meeting of the association for computational linguistics* (pp. 7556-7566).
>
> [5] Berglund, L., Tong, M., Kaufmann, M., Balesni, M., Stickland, A. C., Korbak, T., & Evans, O. (2024). The Reversal Curse: LLMs trained on “A is B” fail to learn “B is A”. *The Twelfth International Conference on Learning Representations*.
>
> [6] Truong, T. H., Baldwin, T., Verspoor, K., & Cohn, T. (2023). Language models are not naysayers: an analysis of language models on negation benchmarks. In A. Palmer & J. Camacho-collados (Eds), *Proceedings of the 12th Joint Conference on Lexical and Computational Semantics (SEM 2023)* (pp. 101–114).
>
> [7] Golovneva, O., Allen-Zhu, Z., Weston, J. E., & Sukhbaatar, S. (2024). Reverse Training to Nurse the Reversal Curse. *First Conference on Language Modeling*.
>
> [8] Graves, A., Wayne, G., & Danihelka, I. (2014). Neural Turing machines. *arXiv preprint arXiv:1410.5401*.
>
> [9] Dehghani, M., Gouws, S., Vinyals, O., Uszkoreit, J., & Kaiser, L. (2019). Universal Transformers. *International Conference on Learning Representations*.
>
> [10] Csordás, R., Irie, K., & Schmidhuber, J. (2021, November). The Devil is in the Detail: Simple Tricks Improve Systematic Generalization of Transformers. In M.-F. Moens, X. Huang, L. Specia, & S. W.-T. Yih (Eds), *Proceedings of the 2021 Conference on Empirical Methods in Natural Language Processing* (pp. 619–634).

---

> > ### Comment · Reviewer_ZpTF · 2025-11-27
> >
> > I thank the authors for following up on these questions. I'm glad to see these additional discussions/results help enrich the paper. I will remain my positive recommendation.

---

> > > ### Author Response · Authors · 2025-11-27
> > >
> > > We thank the reviewer for their time and effort in providing constructive feedback, which has significantly helped improve our work!

---

### Official Review · Reviewer_cspU · 2025-10-30

**Soundness:** 3
**Presentation:** 2
**Contribution:** 3
**Rating:** 6
**Confidence:** 3

**Summary:**

This paper characterizes “pattern matching” in sequence-to-sequence tasks as the identification of particular equivalences between pairs of certain input variables in sequences (subsequences of the input sequence), when all else is held equal. They investigate the ability of models to learn this kind of information by creating a synthetic domain where no other information exists, specifically by framing the problem as the composition of binary functions of the input, each of which is a random lookup table. In general, they find that transformers appear to be using this kind of pattern recognition when trained on this kind of data; but only when these equivalences appear several times. In general, in situations where such an equivalence is found, evidence can be found in intermediate layers for clustering. They develop a scaling law for the data needed to perform this kind of pattern matching. Additionally, they discover that having a non-tree-structured computation graph (where the same input variable can affect the output via multiple paths), causes pattern recognition to become much harder.

**Strengths:**

The domain setup seems to eliminate other potential sources of information cleanly. The definition of functional equivalence and specifically k-equivalence are simple and naturalistic definitions.

The large sweep over a variety of dataset sizes is also helpful for determining the role of data access.

**Weaknesses:**

The abstract and first paragraph of the introduction do not make it clear enough that “pattern matching" is undesirable. The first sentence could be read as “pattern matching" performed by LLMs as being too surface level. This reading recontextualizes later uses of the term to be neutral rather than negative, confusing such a reader. It should be made more clear that “pattern matching" specifically is being used to exclusively refer to undesirably syntactic/surface level heuristics.

"Functional equivalence, i.e., substituting input fragments observed to result in identical outputs in shared contexts” (used in the abstract and introduction lede) should be rephrased as, by itself, this is too vague to communicate what it means to someone who has not read the paper.. Perhaps something like “functional equivalence, i.e., identifying pairs of subsequences of inputs that consistently lead to identical results when the rest of the input is held constant.”

“Tightly ordered by” is used several times, but as far as I can tell is not a standard term. I think a term like “well predicted by” would be more standard and thus easier to read.

**Questions:**

Pattern matching is generally defined in terms of substitution rules (i.e., $a * (b + c) \to a * b + a * c$. Is the notion of pattern matching you define equivalent to unbounded substitution rules (with an unlimited number of variables on each side of the $\to$)? Is it a generalization? Is it a subset? A discussion of this would help ground your definition in the context of existing notions of pattern matching.

What is the purpose of holding out 30% of inputs and having a test condition where you add a new one in? In my understanding, this task is impossible to perform above chance on, as the primitives are defined as random tables.

---

> ### Author Response · Authors · 2025-11-21
>
> We thank Reviewer cspU for insightful and valuable feedback on our paper.
> We are glad that the reviewer has recognized our work’s strength in clear conceptual formulation and experimental design, with rigorous analyses.
> We would like to respond to each weakness and question in order.
>
> ## (W1) Is Pattern Matching Desirable or Undesirable?
>
> We thank the reviewer for pointing out this subtle but important point.
> We first clarify our view of pattern matching, and then summarize how the changes are made in the updated revision to avoid the potential confusion pointed out by the reviewer.
>
> In general, we intended a neutral position on whether pattern matching is desirable or not, **since it depends on how we define the meaning of ‘generalization'**.
> That said, as motivated in the abstract, pattern matching is often regarded as an undesirable behavior when we observe the absence of systematic generalization behaviors (thereby being ‘surface level’), as the success of pattern matching is bounded in the in-domain region by definition.
> However, when one views generalization in a broader sense as ‘making correct predictions on data that is unobserved during training’, pattern matching can be considered as a desirable one. Indeed, for some cases like in our experimental setup, generalization is completely impossible without pattern matching.
> In short, pattern matching supports in-domain generalization but, by itself, does not guarantee systematic compositional generalization.
>
> Therefore, we take a neutral stance on viewing pattern matching, although it is very reasonable to believe that another mechanism is necessary for systematic compositional generalization. We would like to highlight that our focus is not to advocate nor oppose pattern matching, but to define what pattern matching is and to characterize its limitations.
>
> That said, as the reviewer mentioned, we agree that we initially framed pattern matching as an undesirable behavior in the abstract, which might inadvertently confuse the readers.
> In the revision, we have updated Abstract and Introduction to keep a neutral tone on pattern matching and avoid confusion:
>
> - **In the abstract:** “Despite impressive capabilities, LLMs often exhibit surface-level pattern-matching behaviors, evidenced by OOD generalization failures in compositional tasks.” $\rightarrow$ “Despite impressive capabilities, LLMs' successes often rely on pattern-matching behaviors, yet these are also linked to OOD generalization failures in compositional tasks.”
> - **In the introduction:** "...compositional generalization studies report their "pattern-matching" behaviors, i.e., models exploiting local statistical regularities between input fragments and outputs in some cases. ...discussing pattern matching without a precise definition and diagnosing it post-hoc via benchmark failures." $\rightarrow$
> "...compositional generalization studies report that a core mechanism is "pattern matching", i.e., models learning local statistical regularities between input fragments and outputs. ...discussing pattern matching without a precise definition, often diagnosing it post-hoc rather than characterizing it predictively."
>
> ## (W2 & W3) Suggestions for Improving Presentation
>
> Thank you for your thoughtful comments and concrete suggestions for the improvement of the presentation. We agree with both points, and we applied the changes to the updated revision.

---

> ### Author Response · Authors · 2025-11-21
>
> ## (Q1) Is pattern matching equivalent to unbounded substitution rules?
>
> We thank the reviewer for asking this, and it is a very important point!
>
> No, our formulation is **not equivalent** to unbounded substitution rules.
> It is best viewed as a strictly weaker, evidence-based subset of what unbounded substitution rules describe. Still, it can be seen as a generalization mechanism restricted to the in-domain region.
> Unbounded substitution rules (which imply true **variable binding**) would represent the learning of an abstract, universal rule. If the model had learned such a rule, it should be able to generalize perfectly to all in-domain cases after seeing minimal evidence (e.g., $k=1$) to deduce the rule.
> Our results in **Figure 3 (Right)** show this does *not* happen. Generalization is not a single leap but is **well predicted by $k$**, the amount of statistical evidence. This strongly suggests the model is failing to learn the abstract substitution rule and is instead relying on our more bounded, evidence-based mechanism of functional equivalence.
>
> ## (Q2) Purpose of holding out inputs
>
> Yes, the held-out portion of data is intended to construct OOD data as a control group, and it is trivially impossible for any learning algorithms to perform above chance by design. However, they were helpful to understand the ID generalization or clustering behaviors discussed in Sec.5.1-5.2 (Fig.3 and Fig.4), to compare with in-coverage data as a limit case.
>
> ---
>
> **As a final remark, we have significantly strengthened the theoretical contribution in Sec.6 by replacing the heuristic derivation of Result 6.1 with a rigorous formal proof (now Theorem 6.1), which yields a sample-complexity upper bound for perfect ID generalization.** We kindly refer the reviewer to our global comment (“Major Theoretical Upgrade”) for a summary of this result and the corresponding changes in the updated revision.
>
> We sincerely thank the reviewer again, and we hope our response addresses all points raised by them.

---

> > ### Comment · Reviewer_cspU · 2025-11-24
> >
> > Thank you for your comments, which address my concerns. I have increased my score.

---

> > > ### Author Response · Authors · 2025-11-25
> > >
> > > We are delighted that our response has addressed the concerns raised by the reviewer. Thank you very much for your time and effort to provide helpful feedback, which greatly helped improve our work.

---

### Official Review · Reviewer_ntJX · 2025-10-31

**Soundness:** 4
**Presentation:** 2
**Contribution:** 3
**Rating:** 8
**Confidence:** 4

**Summary:**

This paper is about when LLMs can learn many-to-one (or, equivalently, non-injective) maps. A many-to-one map is a function such that two different inputs yield the same output. All maps are deterministic. In their experiments, an LLM is trained on many examples of such functions (sometimes composed with each other) and is asked to produce outputs on unseen inputs. For example, if we see $f_1(a) = f_1(b) = c$ and $f_2(f_1(b), d) = e$, we can infer that $f_2(f_1(a), d) = e$ as well, without having seen this particular example before.

They use many-to-one maps as a formalization of what pattern matching means in LLMs. They then create quantifiable tests for when pattern matching is or is not happening. They also explore how data requirements scale with the complexity of the task, and how the amount of computation necessary to learn pattern matching scales with the size of the input vocabulary. Each section of the paper is summarized as follows:

- Their first experiments show how the number of times an invariance is observed affects the test accuracy on the many-to-one maps. Despite the fact that just two observations of an invariance is sufficient to implement an algorithm that solves the task perfectly, the LLM only begins to generalize well until after more examples of the invariance are seen (3+, depending on the amount of training).
- They next develop a metric for measuring how well internal representations capture shared states. In particular, one might hope that if two function inputs lead to the same output, the internal representations would be similar after a few layers. Similarly, if two function inputs lead to different outputs, they internal representations should be different at all points. They capture this by measuring cosine similarity between inputs that induce the same output and cosine between inputs that induce different outputs and then taking the difference between these two quantities. They then show that the models trained on their many-to-one task exhibit high values on the metric.
- They next consider how much data should be needed to learn the task, for a given input vocabulary. They show that, depending on how many examples are needed to confirm an invariance, the number of examples needed scales as polynomial in the size of the domain, with exponent between 2 and 2.5. By scaling the input size, they verify empirically that these scaling laws hold.
- When the function composition structure is more complex (the computation graph is not a tree, and is instead some general directed graph), they show the LLMs do not perform well. This is because it is harder to generalize when the same token is used for multiple different arguments to a function.
- They also train the models to output intermediate computations and find that it leads to more data-efficient generalization. However, when learning with general graphs rather than trees there is still no generalization.

**Strengths:**

- At a high level, thinking about generalization in terms of many-to-one functions seems like it clearly captures a kind of task-level generalization. Completing the task correctly requires non-trivial logical reasoning. The task has the nice properties that (1) it is possible to get 100% accuracy when correctly applying logical reasoning / a graph algorithm and (2) the LLM never sees the exact problem instance it is evaluated on.
- The empirical results in the paper strongly support the narrative presented. The setups of the experiments are thoughtful, and the analysis is suitable for the questions addressed in the paper. I would say the first results (that more examples of an invariance are helpful) is not very surprising, but I learned a lot from the second through fourth analyses (scaling laws, non-tree tasks, and CoT).

**Weaknesses:**

- I found the exposition introducing the problem to be a bit confusing. It wasn’t clear to me whether pattern matching is a desirable or undesirable property of transformers (is it capturing overfitting or generalizing?) The abstract suggests that surface-level pattern matching is bad, but perhaps that deeper pattern matching (which survives multiple logical steps) is a good thing.
- I am also confused about why it is interesting to understand pattern matching in LLMs. I’m not sure how the toy problem is supposed to generalize to other tasks, or shed insight into broad phenomena around LLM reasoning. The introduction did not really have examples of pattern matching in real-world LLM uses, so this added to the confusion. There were many citations supposedly about pattern matching, but not any explanation of how we are supposed to imagine pattern matching works beyond the toy problems in this paper itself. I believe there could be a good motivation for studying this problem, but I don’t get it from this paper, and I’d like the authors to explain.
- I think there could be more discussion of what makes the non-tree case hard. Is there not a way to reduce the non-tree task to the tree task? Or is generalization impossible because of the same token is used for multiple arguments.
- Another minor critique I have is about terminology: “functional equivalence” to me suggests that two functions are the same. I think “functional invariance” is a clearer term?

**Questions:**

What are the class of real-world problems or tasks that being good at pattern matching helps solve? Is it supposed to capture logical reasoning? Why this particular logical reasoning task versus another (like syllogisms or SAT problems)?

---

> ### Author Response · Authors · 2025-11-21
>
> We thank Reviewer ntJX for their time reviewing our paper, providing valuable and rich feedback on our work. We are glad that the reviewer has recognized the clarity of our framework and the significance of our experimental analyses. We would like to respond to each weakness and question in order.
>
> ## (W1) Is Pattern Matching Desirable or Undesirable?
>
> This is a subtle point, and we take a neutral position: **the desirability of pattern matching depends on how we define 'generalization’.**
>
> As motivated in the abstract, pattern matching is often regarded as an undesirable behavior when we observe the absence of systematic generalization behaviors (often used interchangeably with ‘extrapolation’), as the success of pattern matching is bounded inside the in-domain region by definition.
> However, when we view generalization as ‘making correct predictions on data that is unobserved during training’, pattern matching can be considered as a desirable one. Indeed, we could not observe any hope for generalization without pattern matching, at least in our experimental setup.
> In short, pattern matching supports in-domain generalization but, by itself, does not guarantee systematic compositional generalization. Our focus is not to advocate nor oppose pattern matching upon a specific definition of generalization, but to define what pattern matching is, and to characterize its limitations.
>
> That said, as reviewer cspU mentioned, we agree that we initially framed pattern matching as an undesirable behavior in the abstract, which might inadvertently confuse the readers.
> In the revision, we have updated the abstract and Sec.1 to keep a neutral tone on pattern matching and avoid such confusion:
>
> - **In the abstract**: “Despite impressive capabilities, LLMs often exhibit surface-level pattern-matching behaviors, evidenced by OOD generalization failures in compositional tasks.” $\rightarrow$ “Despite impressive capabilities, LLMs' successes often rely on pattern-matching behaviors, yet these are also linked to OOD generalization failures in compositional tasks.”
> - **In the introduction (Sec.1)**: "...compositional generalization studies report their "pattern-matching" behaviors, i.e., models exploiting local statistical regularities between input fragments and outputs in some cases. ...discussing pattern matching without a precise definition and diagnosing it post-hoc via benchmark failures." $\rightarrow$
> "...compositional generalization studies report that a core mechanism is "pattern matching", i.e., models learning local statistical regularities between input fragments and outputs. ...discussing pattern matching without a precise definition, often diagnosing it post-hoc rather than characterizing it predictively."

---

> ### Author Response · Authors · 2025-11-21
>
> ## (W2) Why is it interesting to understand pattern matching in LLMs?
>
> This is a great point, and we are glad to have the opportunity to discuss this further. We believe understanding pattern matching is crucial for two reasons: **(i) it provides a predictive lens on modern LLM behavior (including their failures)**, and **(ii) it connects empirical ML results to foundational debates on systematic compositionality.**
>
> At a practical level, pattern matching is interesting because it plausibly accounts for LLMs’ success on **in-distribution** examples, while systematically failing on **compositional extrapolation**. Our framework moves beyond post-hoc diagnosis and makes this mechanism predictive: it tells us when a purely pattern-matching learner *should* generalize and when it *should not*, given the amount of functional equivalence evidence in the training data. This leads to several practical implications:
>
> - Recent work on the Reversal Curse [1] shows that models trained on “A is B” often fail to infer “B is A”. In our framework, this is exactly what one would expect from a purely pattern-matching learner, as observing “A $\rightarrow$ B” provides no functional equivalence evidence for the reversed pattern, so generalization to “B $\rightarrow$ A” is not supported.
> - Our experimental and theoretical analysis of the scaling law for the two-hop task shows that the amount of data needed for in-domain generalization grows roughly polynomially in the size of the domain. This matches the empirically observed “data explosion” for multi-hop reasoning in natural language tasks [2], and interprets such a data requirement explosion as a fundamental *property* of pattern-matching-based generalization.
> - Because our notion of coverage is defined combinatorially, it provides a theoretical basis for designing data augmentation strategies (e.g., which substitutions actually increase functional evidence), complementing heuristic approaches such as good-enough compositional augmentation [3].
>
> Beyond immediate practical implications, our motivation is also connected to a decades-old debate in cognitive science and philosophy of mind on what computational mechanisms are necessary for **systematic compositionality**. In 1988, Fodor and Pylyshyn argued that connectionist networks cannot explain systematic behavior (e.g., if one understands ‘Mary loves John’, then one also understands ‘John loves Mary’) without an explicit role–filler binding mechanism [4]. They claimed that binding roles (e.g., subject vs. object) and fillers (e.g., ‘John’, ‘Mary’) are crucial for such systematicity, and that standard connectionist architectures lack this capacity, which is still debated.
>
> Meanwhile, accumulating studies show that modern LLMs struggle with precisely these kinds of systematic behaviors as discussed above. Taken together, this makes it very plausible that **pattern matching alone** is insufficient to explain systematic generalization on compositional tasks.
> However, there has been no formal framework of pattern matching as a common ground to understand pattern-matching behaviors in LLMs, making it difficult to apply the conclusion in one task to another.
> For instance, Dziri et al. [5] demonstrate failures on simple compositional tasks such as integer multiplication beyond seen combinations. However, it is unclear whether the limited generalization stems purely from matching previously seen patterns or from leveraging task-specific algebraic properties such as commutativity.
> Our contribution is to provide a formal framework for pure pattern matching that future work can constructively expand to more realistic scenarios, ultimately helping to diagnose and characterize which additional mechanisms are required for compositional systematicity.
>
> We will make these motivations explicit in the introduction and related work in the final version, provided with additional space.

---

> ### Author Response · Authors · 2025-11-21
>
> ## (W3) What makes the non-tree case hard?
>
> The difficulty of generalizing the non-tree case originates from the fact that the problem is inherently underdetermined, unlike the tree cases, from a purely pattern-matching learner’s perspective.
> As illustrated in Fig. 6 (Left) and described in L392-398, since $x_2$ affects the output both directly and through the intermediate state, functional equivalence for the subsequences of the first two inputs $(x_1, x_2)$ and $(x^\prime_1, x^\prime_2)$ can only be established when $x_2 = x^\prime_2$.
> Hence, there are (at least) two valid hypotheses explaining the structure:
>
> - Hypothesis (1): The ground-truth structure that treats $(x_1, x_2)$ and $(x^\prime_1, x^\prime_2)$ are equivalent as far as $f_1(x_1,x_2) = f_1(x^\prime_1,x^\prime_2)$.
> - Hypothesis (2): Treat $(x_1,x_2)$ and $(x^\prime_1,x^\prime_2)$ are not equivalent as far as $x_1 \neq x'_1$.
>
> While both can explain the observed data consistently, functional equivalence alone does not invalidate any of them, nor support that the model should prefer (1) over (2).
> To put it another way, if a learner has no *a priori* structural information (or inductive bias) such that there is an intermediate state $b$ determined by $b = f_1(x_1,x_2)$, there is no good reason to capture the ground-truth structure (Fig. 2d) we intended.
> Therefore, we do not believe that such non-tree cases can be completely reduced to the tree case (except for the trivial reduction to a lookup table that maps a triplet directly to the output), unless provided with additional inductive bias (which will make the learner not ‘purely pattern-matching’ anymore).
>
> Another interesting consequence observed in the non-tree case is that more data is needed to cover all in-domain data inside the k-coverage. As demonstrated in Fig.16 (in App. G.1), all ID data lies inside  $k$-coverage with $k=2$ only when data size is 50k, where the two-hop task achieves this at the size of 15k (Fig. 3 Left). This makes it harder for the learner to generalize on ID test data of the non-tree case with the same amount of training data compared to the two-hop case.
>
> ## (W4) Terminology on functional equivalence
>
> We appreciate the reviewer’s thoughtful comment on the terminology.
> We intended to use the term ‘functional’ not by the mathematical meaning of function as mappings, but in that two given subsequences are ‘functioning’ as if they are the same.
> The suggested term ‘functional invariance’ is a strong candidate for the term, but we are cautious since it can be interpreted as an invariance as a characteristic of a single primitive function, and confused with what we characterized as ‘function property-based generalization’ as a separate generalization mechanism in Discussion (L474--478) and App.H.
> Hence, we decided to improve the description for clarity while keeping the original term in our revision, by improving the explanation of the term as suggested by reviewer cspU.

---

> ### Author Response · Authors · 2025-11-21
>
> ## (Q1) Response to Question 1
>
> ### 1. What real-world tasks does this help solve?
> Our work suggests pattern matching is not a solution to a specific class of tasks, but rather may serve as a primary mechanism that models use to solve the **'in-domain' (ID) portion of many real-world tasks.** This may include tasks like syllogisms or arithmetic, where success relies on exploiting vast statistical regularities observed in training.
>
> Our framework explains why models often succeed on in-distribution examples but fail on long-tail or out-of-domain (OOD) instances of those very same tasks. In summary, the main “real-world problem” that pattern matching helps solve is achieving **high in-domain performance**.
>
> ### 2. Is it supposed to capture logical reasoning? (If so) Why this particular task?
> This might depend on how we define the term ‘logical,’ but we claim that pattern matching, as we formalize it, is **not** intended to capture logical reasoning. Although the notion of logicality is debated in modern philosophy of logic, we ground our argument in one influential perspective due to Tarski and Sher (see [6]): on this view, a logical operation is one that is invariant under all permutations of the domain. For example, the meaning of a logical constant such as ‘and’ or ‘for all’ does not depend on which particular entities or predicates it is applied to.
>
> In contrast, our notion of functional equivalence is **evidence-based**: two fragments are treated as equivalent only after they have appeared in at least k shared contexts with identical outputs, and nothing in our setup guarantees that this equivalence will hold under arbitrary permutations of entities. If the model were learning a permutation-invariant ‘logical structure’ of the task, it should generalize to all in-domain cases with minimal evidence (e.g., k=1), which our results in Fig. 3 (Right) show is not the case. We therefore take these results as evidence that the pattern matching we study is distinct from logical reasoning in Tarski and Sher’s sense of logicality.
>
> ---
>
> **As a final remark, we have significantly strengthened the theoretical contribution in Sec.6 by replacing the heuristic derivation of Result 6.1 with a rigorous formal proof (now Theorem 6.1), which yields a sample-complexity upper bound for perfect ID generalization.** We kindly refer the reviewer to our global comment (“Major Theoretical Upgrade”) for a summary of this result and the corresponding changes in the updated revision.
>
> We sincerely thank the reviewer again, and we hope our response addresses all points raised by the reviewer.
>
> ---
>
> ## References
>
> [1] Berglund, L., Tong, M., Kaufmann, M., Balesni, M., Stickland, A. C., Korbak, T., & Evans, O. (2024). The Reversal Curse: LLMs trained on “A is B” fail to learn “B is A”. *The Twelfth International Conference on Learning Representations*.
>
> [2] Yao, Y., Du, Y., Zhu, D., Hahn, M., & Koller, A. (2025). Language models can learn implicit multi-hop reasoning, but only if they have lots of training data. In C. Christodoulopoulos, T. Chakraborty, C. Rose, & V. Peng (Eds), *Proceedings of the 2025 Conference on Empirical Methods in Natural Language Processing* (pp. 9695–9713).
>
> [3] Andreas, J. (2020). Good-enough compositional data augmentation. *In Proceedings of the 58th annual meeting of the association for computational linguistics* (pp. 7556-7566).
>
> [4] Fodor, J. A., & Pylyshyn, Z. W. (1988). Connectionism and cognitive architecture: A critical analysis. *Cognition*, *28*(1-2), 3-71.
>
> [5] Dziri, N., Lu, X., Sclar, M., Li, X. L., Jiang, L., Lin, B. Y., ... & Choi, Y. (2023). Faith and fate: Limits of transformers on compositionality. *Advances in Neural Information Processing Systems*, 36, 70293-70332.
>
> [6] Sher, G., (1991). The Bounds of Logic: A Generalized Viewpoint. MIT Press.

---

### Official Review · Reviewer_5m5w · 2025-11-01

**Soundness:** 3
**Presentation:** 2
**Contribution:** 2
**Rating:** 6
**Confidence:** 3

**Summary:**

This paper formulates functional equivalence as the condition where two sets of tokens produce the same output in the same context. The experimental results demonstrate that the model's performance and internal representation of these equivalences get stronger as more of these functional equivalences occur in the training data. The paper also analyzes the scaling law (required training data size depends on the vocabulary size), the effect of ambiguous composition, and the same phenomenon in CoT training.

**Strengths:**

1. This paper studies an important problem of the compositional generalization of language models.

2. The experiments include various settings of practical relevance.

**Weaknesses:**

1. The results are limited to small synthetic task structures.

2. The settings require a deterministic function and strict functional equivalence, which may be too restrictive in a real-world NLP dataset.

**Questions:**

1. If the task structure is more complicated, would observing robust functional equivalence still be enough for compositional generalization? Wouldn't task complexity challenge the model's understanding of the task, resulting in training failure?

2. Is there any equivalent empirical observation where compositional generalization requires robust observation of functional equivalence in a real-world NLP dataset?

3. Would the formulation in the paper generalize to stochastic functions and approximate functional equivalence?

---

> ### Author Response · Authors · 2025-11-21
>
> We thank Reviewer 5m5w for the time and effort in reviewing our work. We are glad that the reviewer has recognized our work’s significance and rigorous experimental analyses. We will respond to each weakness and question raised by the reviewer.
>
> ## (W1 & W2) Motivation for the experimental design with synthetic structures
>
> While our scope is restricted to a synthetic setup, we would like to emphasize that this is deliberately adopted to dissect and understand pattern-matching behaviors. Previous studies on compositional generalization have observed models’ pattern-matching behaviors, yet they rarely defined them with precise language or tested pattern-matching in isolation. Our work therefore aims to construct a formal language to characterize what pure pattern matching is, which future studies can use to ground their hypotheses on and describe their observations. Although experimenting with more realistic setups was not covered in the scope of this work, we believe our framework and insights provide clear next steps to expand towards more complicated setups, providing a constructive roadmap to understand models’ compositional generalization.
>
> In our response below to the reviewer’s questions, we discuss concrete practical implications of our framework and its potential extensions to more complex tasks, including the nondeterministic cases.
>
> ## (Q1) On more complicated task structures
>
> The term ‘complexity' is a multifaceted concept, and the question can be rephrased in various ways, which we answer in order:
>
> - What if the task structure is a **deeper tree**?
>     - As we observed in Sec.6 and Fig.5, the data requirement for ID generalization relying on pattern matching will explode with a steeper scaling law. In such a regime, the capacity of a given model might be a bottleneck for generalization, even though the data supply is sufficient for establishing functional equivalence.
> - What if the task structure is a **general graph with more path ambiguities**?
>     - We empirically validated in Sec.7 that path ambiguity makes functional equivalence observation practically insufficient (albeit being helpful) for the model to capture and utilize the underlying task structure, resulting in training failure. We expect adding more path ambiguities will further make the functional equivalence observation insufficient for full generalization, but we highlight that this can be predicted and explained under our framework, as discussed in Sec.7.
> - What if the task structure is **non-deterministic and noisy**?
>     - This is discussed in our response to the second question below.
> - What if the task structure is **dynamic (=mixed)**?
>     - We do expect there will be synergy or interference between different task structures. Although not included in the paper, for the experiment used in App.I, we trained a model with a mixture of one-hop triplets with two-hop instances. In that experiment, we observed a synergistic behavior: the generalization speed is boosted, and the number of two-hop instances required for complete ID generalization ($N_{\rm req}$) is slightly lower, compared to when we train the model only with two-hop data.
>     Such a synergistic effect is also observed in [1], which demonstrated that training with mixed data or a curriculum learning approach of showing data of gradually increasing hops can improve data efficiency.
>
> Although we did not cover some of these in the scope of this work, each of them points to a straightforward and important future direction that can be tested with the language of our framework. This is exactly the aim of our work, providing a formal framework to understand pattern matching in complex task structures in future studies. Some of the above might reveal that functional equivalence as a necessary condition alone is insufficient to establish generalization in complex setups, as the reviewer mentioned. We would view this as a further characterization of the limitation of pattern-matching behaviors, synergistically contributing to our understanding of compositional generalization.

---

> ### Author Response · Authors · 2025-11-21
>
> ## (Q2) Connection to previous empirical observation in real-world NLP dataset
>
> Yes, there has been consistent evidence that a robust functional equivalence observation is crucial for compositional generalization in real-world NLP tasks:
>
> - **It helps to quantifiably explain the data-hungry nature of LLMs on compositional tasks.** [1] observed that data demand for generalization on k-hop natural-language data dramatically increases with increasing the number of hops. Such a data-hungry nature of compositional tasks is also well observed in semantic parsing, the task in which the model should convert natural language instructions into a structured SQL query. [2] demonstrated that the diversity of data, in that the same component is shown in diverse contexts, is more important than the sheer size of the dataset for good generalization.
> We note that this insight is not completely new (e.g., [3]), but our work steps further to provide a rigorous quantitative analysis of the data scaling law required for generalization on compositional task structures through pattern matching.
> - It is widely known that the strategic data augmentation method [4] of substituting components in natural language data (and hence can contribute to a robust functional equivalence) greatly boosts a model’s performance on a given natural-language compositional generalization task.
> - Our framework aligns with the reversal curse phenomenon (e.g., failure to automatically infer ‘A is child of B’ by observing ‘B is parent of A’) observed in recent LLMs [5], as pattern-matching models fundamentally cannot generalize to reversed relationships without explicit functional equivalence evidence in training data.
> - Similarly, our framework well explains the notoriously hard problem of logical negation in LLMs [6], as a purely pattern-matching learner cannot deduce that the negation rule (if a statement ‘p’ is true, then ‘not p’ is false and vice versa) should apply to *any* well-formed statement (including the ones that aren’t observed) through finite observations.
>
> Overall, these results strongly support the view that a robust functional equivalence observation is crucial, which motivated our work to formalize those conditions.
>
> ## (Q3) On the extension of our framework to non-deterministic functions and noisy functional equivalence
>
> This is a great question. Although it is nontrivial, we believe our framework can be extended to handle stochasticity or approximate functional equivalences.
>
> The requirement of functional $k$-equivalence can be relaxed by introducing a tolerance parameter. Instead of requiring perfect agreement across shared contexts, we can treat two fragments as approximately equivalent when the fraction of the agreement under shared contexts exceeds a certain tolerance threshold. Then, the rest of our machinery (e.g., substitution graphs and $k$-coverage) can be defined in the same spirit, which enables the further investigation of qualitative phenomena we observed in deterministic setups (e.g., rapidly growing data requirements and sensitivity to path ambiguity).
>
> However, a full treatment of non-deterministic settings would require specifying a concrete noise model or deriving formal learnability guarantees under this extension, which is beyond the scope of this work. Hence, we leave it as an important and interesting future research direction, building on our framework.
>
> ---
>
> **As a final remark, we have significantly strengthened the theoretical contribution in Sec. 6 by replacing the heuristic derivation of Result 6.1 with a rigorous formal proof (now Theorem 6.1), which yields a sample-complexity upper bound for perfect ID generalization.** We kindly refer the reviewer to our global comment (“Major Theoretical Upgrade”) for a summary of this result and the corresponding changes in the updated revision.
>
> We sincerely thank the reviewer again for the time and effort to improve this work.

---

> > ### Author Response · Authors · 2025-11-21
> >
> > ## References
> >
> > [1] Yao, Y., Du, Y., Zhu, D., Hahn, M., & Koller, A. (2025). Language models can learn implicit multi-hop reasoning, but only if they have lots of training data. In C. Christodoulopoulos, T. Chakraborty, C. Rose, & V. Peng (Eds), *Proceedings of the 2025 Conference on Empirical Methods in Natural Language Processing* (pp. 9695–9713).
> >
> > [2] Keysers, D., Schärli, N., Scales, N., Buisman, H., Furrer, D., Kashubin, S., … Bousquet, O. (2020). Measuring Compositional Generalization: A Comprehensive Method on Realistic Data. *International Conference on Learning Representations*.
> >
> > [3] Lake, B., & Baroni, M. (2018). Generalization without systematicity: On the compositional skills of sequence-to-sequence recurrent networks. *In International conference on machine learning* (pp. 2873-2882). PMLR.
> >
> > [4] Andreas, J. (2020). Good-enough compositional data augmentation. *In Proceedings of the 58th annual meeting of the association for computational linguistics* (pp. 7556-7566).
> >
> > [5] Berglund, L., Tong, M., Kaufmann, M., Balesni, M., Stickland, A. C., Korbak, T., & Evans, O. (2024). The Reversal Curse: LLMs trained on “A is B” fail to learn “B is A”. *The Twelfth International Conference on Learning Representations*.
> >
> > [6] Truong, T. H., Baldwin, T., Verspoor, K., & Cohn, T. (2023). Language models are not naysayers: an analysis of language models on negation benchmarks. In A. Palmer & J. Camacho-collados (Eds), *Proceedings of the 12th Joint Conference on Lexical and Computational Semantics (SEM 2023)* (pp. 101–114).

---

> > > ### Comment · Reviewer_5m5w · 2025-11-25
> > >
> > > Thank you for your detailed answers to my review. I acknowledge that the rebuttal addresses all of my concerns. I updated my score accordingly.

---

> > > > ### Author Response · Authors · 2025-11-26
> > > >
> > > > We are delighted that our response has addressed all concerns raised by the reviewer. We sincerely appreciate your constructive feedback, which has significantly helped improve our work.

---

### Author Response · Authors · 2025-11-21
**Global comment (1/2)**

We genuinely appreciate all reviewers for their thoughtful and helpful feedback, which has significantly improved the paper. In this global comment, we would like (i) to recall the strengths of our work raised by reviewers, (ii) to summarize our updates and clarifications in our revised manuscripts as per the reviewers’ comments, and (iii) to highlight our major updates in our theoretical analysis established upon our formal framework.

## Strengths

We are encouraged that the reviewers recognized several core strengths of the work:

- **Problem and framework:** tackling compositional generalization via a “simple and naturalistic” (cspU) and “significant” (ZpTF) formalization of pattern matching (5m5w, ntJX).
- **Experimental design:** using a “thoughtful” (ntJX) and “clean” (cspU) synthetic domain that isolates pattern-matching behaviors.
- **Analysis:** providing “systematic” (ZpTF) analyses that “strongly support the narrative” (ntJX) and yield “important insights” (ZpTF).

## Common Questions & Clarifications

The reviewers raised valid points regarding the framing of pattern matching and the work's connection to real-world implications. We summarize our updates and clarifications below:

- **Stance on 'Pattern Matching' (ntJX, cspU):** We clarified that our view of pattern matching is **neutral** and its desirability depends on the definition of generalization (desirable for in-domain/interpolation, and insufficient for OOD/extrapolation). We have revised the abstract and the introduction to remove any inadvertently negative framing.
- **Motivation for Synthetic/Controlled Setup (5m5w, ntJX):** We discussed why a controlled setup is methodologically important. It allows us:
    1. to isolate the specific mechanism of functional equivalence from other confounds (like algebraic properties),
    2. to derive precise quantitative boundaries (such as our scaling laws) that help explain LLM failures on real-world tasks, and
    3. to provide a clean setting that can inform ongoing debates about which neural mechanisms may be required for systematic compositionality [1].
- **Practical Implications (ZpTF, 5m5w, ntJX):** We elaborated on the practical implications, including:
    1. quantitatively helping to explain the data-hungry nature of compositional tasks,
    2. providing a theoretical basis for targeted data augmentation [2], and
    3. offering a mechanistic hypothesis for well-known failure cases of LLMs, including the reversal curse [3] and negation failures [4]. We will expand this discussion in the final version, provided with additional space.

## **Major Upgrade in Theoretical Analysis: Rigorous Proof of Scaling Law (Theorem E.6)**

We have significantly strengthened the theoretical foundation of our work by replacing the heuristic derivation in Section 6 with a **rigorous formal proof** (now **Theorem 6.1**). For a large enough vocabulary size $n$, the theorem provides a **sample complexity upper bound** $\tilde{O} \left(n^{2.5-\frac{0.5}{k}}\right)$ (ignoring polylogarithmic factors in $n$) to achieve perfect in-domain generalization with high probability, which is formally stated and proved in **Appendix E** in the updated revision.

The proof proceeds in three key steps to rigorously handle the dependencies in the data sampling process:

1. **Reduction to Graph Connectivity:** We first prove (in Lemma E.9) a sufficient condition for perfect coverage of all ID data: it suffices to prove the connectivity of all evidence graphs (Definition E.5, whose vertices are certain input subsequences sharing the intermediate states, and edges represent functional $k$-equivalence between two subsequences) with high probability.
2. **Poissonization:** To disentangle the dependencies inherent in fixed-size dataset sampling (characterized with a multinomial distribution), we apply Poissonization techniques. This allows us to model the edge formation in evidence graphs as independent events under a Poisson process, yielding upper bounds on the failure probability (Lemma E.11).
3. **Random Graph Theory:** We identify the evidence graphs as instances of random $k$-intersection graphs under Poissonization. We then leverage the threshold results from the random graph theory literature to derive the sufficient sample size $N$.

In Appendix E, we actually provide more general sample complexity upper bound results under mild constraints about the cardinalities of the token sets (e.g., $\mathcal{X}_1, \mathcal{X}_2, \mathcal{X}_3$) in Theorems E.6 (for $k\ge 2$) and E.7 (for $k=1$). Combining these two and further assuming the balanced token set cardinalities at a rate of $\Theta(n)$, we yield a simpler form of sample complexity as in Theorem 6.1 (or, Corollary E.8 for a more general form).
We have reflected this in the revision, with the updated sentences annotated in green.

We sincerely thank the reviewers again, and we hope our response addresses all points raised by the reviewers.

---

> ### Author Response · Authors · 2025-11-21
> **Global comment (2/2)**
>
> ## References
>
> [1] Fodor, J. A., & Pylyshyn, Z. W. (1988). Connectionism and cognitive architecture: A critical analysis. *Cognition*, *28*(1-2), 3-71.
>
> [2] Andreas, J. (2020). Good-enough compositional data augmentation. *In Proceedings of the 58th annual meeting of the association for computational linguistics* (pp. 7556-7566).
>
> [3] Berglund, L., Tong, M., Kaufmann, M., Balesni, M., Stickland, A. C., Korbak, T., & Evans, O. (2024). The Reversal Curse: LLMs trained on “A is B” fail to learn “B is A”. *The Twelfth International Conference on Learning Representations*.
>
> [4] Truong, T. H., Baldwin, T., Verspoor, K., & Cohn, T. (2023). Language models are not naysayers: an analysis of language models on negation benchmarks. In A. Palmer & J. Camacho-collados (Eds), *Proceedings of the 12th Joint Conference on Lexical and Computational Semantics (SEM 2023)* (pp. 101–114).

---

> > ### Author Response · Authors · 2025-11-25
> > **Revision Update: Sharp Sample Complexity Bound & Practical Implications**
> >
> > To AC and Reviewers,
> >
> > We have uploaded a new revision incorporating two major changes:
> >
> > 1. **We strengthened Theorem 6.1 by showing that our sample complexity bound is indeed sharp for** $k\geq 2$**.** Specifically, we proved that, for some instances of two-hop task, the learner does not achieve perfect ID generalization with high probability if the data scaling law grows slightly slower than the specified rate. We accordingly updated the main text and appendix to incorporate the updated statement and proof.
> > 2. We noticed that the ICLR 2026 author guide allows one additional page for revisions during the rebuttal period. **As we promised to reviewers ntJX and ZpTF, we added a detailed discussion of practical implications to the main text (Sec.9.1).**
> >
> >
> >
> > We thank the reviewers again for the time and effort to review our work.
> >
> > Sincerely,
> >
> > Authors

---

### Meta-Review · Area_Chair_MQdQ · 2026-01-05

**Summary:**

This paper provides a formal characterization of pattern matching in compositional task structures, defining it via functional equivalence and studying its strengths and limits in carefully controlled synthetic settings. The work combines a clean experimental design with theoretical analysis, including scaling laws and failure modes such as path ambiguity. Reviewers generally found the framework clear, the experiments thoughtful, and the results informative for understanding when pattern matching supports in-domain generalization and when it breaks down.

**Reviewer Concerns:**

Main concerns :
- reliance on synthetic task structures and questions about how directly the results transfer to real-world language tasks
-  initial ambiguity in the framing of “pattern matching” (desirable vs undesirable) and motivation in the introduction
-  requests for stronger theoretical grounding of the scaling claims and clearer discussion of practical implications
- questions about extending the framework to more complex, non-tree, stochastic, or mixed task settings

The AC found the rebuttal to be strong and to be addressing the significant concerns. The authors did a good job at clarifying the framing of pattern matching, adopting a neutral stance, and improving the introduction and abstract. They significantly strengthened the theoretical contribution by replacing heuristic arguments with rigorous proofs of the scaling laws. The rebuttal also expanded the discussion of practical implications and connections to known LLM failures (e.g., reversal curse, negation), and multiple reviewers explicitly stated that their concerns were addressed. They indicated their eagerness to maintain positive scores. While questions about broader real-world applicability remain, these are acknowledged as scope limitations rather than flaws.

**Reviewer Scores:**

Pre-rebuttal scores were mostly positive, with some reviewers explicitly above threshold and others borderline but open to acceptance. After rebuttal, sentiment clearly improved.
* reviewer ntJX: 8, unchanged (strongly positive)
* reviewer ZpTF: 8, unchanged (strongly positive)
* reviewer 5m5w: 6 --> increased after rebuttal
* reviewer cspU: 6 --> increased after rebuttal

---

### Decision · Program_Chairs · 2026-01-26

Accept (Poster)